# CALIBRATING GENERATIVE MODELS TO DISTRIBUTIONAL CONSTRAINTS

## ABSTRACT

Generative models frequently suffer miscalibration, wherein class probabilities and other statistics of the sampling distribution deviate from desired values. We frame calibration as a constrained optimization problem and seek the closest model in Kullback-Leibler divergence satisfying calibration constraints. To address the intractability of imposing these constraints exactly, we introduce two surrogate objectives for fine-tuning: (1) the relax loss, which replaces the constraint with a miscalibration penalty, and (2) the reward loss, which converts calibration into a reward fine-tuning problem. We demonstrate that these approaches substantially reduce calibration error across hundreds of simultaneous constraints and models with up to one billion parameters, spanning applications in protein design, image generation, and language modeling.

## 1 INTRODUCTION

Generative models commonly produce samples whose statistics deviate systematically from desired values. Such *miscalibration* occurs in many domains. Image models, such as GANs and diffusion models, exhibit mode collapse, producing images that cover only a subset of the training distribution (Arora & Zhang, 2017; Qin et al., 2023). Language models represent gender, race, religion, and age in ways that reinforce societal biases (Gallegos et al., 2024). In synthetic biology applications, protein structure models produce samples that have alpha-helical and beta-strand substructures at frequencies atypical of proteins found in nature (Lu et al., 2025), and DNA models generate samples that contain subsequences at frequencies that differ from those in human DNA (Sarkar et al., 2024). These calibration errors arise from many sources including dataset imbalances, suboptimal training dynamics, and post-hoc adjustments such as low-temperature sampling or preference fine-tuning.

We frame calibration as a constrained optimization problem: find the distribution closest in Kullback-Leibler (KL) divergence to the base model that satisfies a set of expectation constraints. We introduce two fine-tuning algorithms—**CGM-relax** and **CGM-reward** ("calibrating generative models")—that approximately solve the calibration problem by stochastic optimization. We demonstrate across three applications that CGM effectively calibrates high-dimensional generative models to meet hundreds of simultaneous constraints.

**Problem statement.** Consider a trained "base" generative model $p_{\theta_{\text{base}}}(\boldsymbol{x})$ with parameters $\theta_{\text{base}}$, a statistic $\boldsymbol{h}(\boldsymbol{x})$, and an expectation value desired for the statistic $\boldsymbol{h}^*$. We say $p_{\theta_{\text{base}}}$ is *calibrated* if $\mathbb{E}_{p_{\theta_{\text{base}}}}[\boldsymbol{h}(\boldsymbol{x})] = \boldsymbol{h}^*$ and *miscalibrated* if $\mathbb{E}_{p_{\theta_{\text{base}}}}[\boldsymbol{h}(\boldsymbol{x})] \neq \boldsymbol{h}^*$. In the case that $p_{\theta_{\text{base}}}$ is miscalibrated, our goal is to fine-tune its parameters $\theta_{\text{base}}$ to some $\theta$ such that $p_\theta$ is calibrated.

For example, if $\boldsymbol{h}(\boldsymbol{x}) = \mathbb{1}\{\boldsymbol{x} \in C\}$ is the 0-1 function indicating whether $\boldsymbol{x}$ belongs to class $C$, then $\mathbb{E}_{p_{\theta_{\text{base}}}}[\boldsymbol{h}(\boldsymbol{x})] = p_{\theta_{\text{base}}}(\boldsymbol{x} \in C)$ is the probability that $p_{\theta_{\text{base}}}$ generates a member of class $C$. When $\boldsymbol{h}^* > \mathbb{E}_{p_{\theta_{\text{base}}}}[\boldsymbol{h}(\boldsymbol{x})]$, calibration corresponds to increasing the probability of class $C$.

For a given $\boldsymbol{h}(\cdot)$ and $\boldsymbol{h}^*$, many calibrated models may exist. Provided a calibrated model exists, we seek the one that is closest to the base model in KL divergence,

$$p_{\theta^*} := \arg\min_{p_\theta} \mathrm{D}_{\mathrm{KL}}\left(p_\theta \parallel p_{\theta_{\text{base}}}\right) \quad \text{such that } \mathbb{E}_{p_\theta}[\boldsymbol{h}(\boldsymbol{x})] = \boldsymbol{h}^*, \tag{1}$$

where $\mathrm{D}_{\mathrm{KL}}\left(p' \parallel p\right) = \mathbb{E}_{p'}[\log p'(\boldsymbol{x})/p(\boldsymbol{x})]$ for $p'$ with a probability density with respect to $p$. Out of many possible notions of distance we choose $\mathrm{D}_{\mathrm{KL}}$ because it is simple and, as we will see, is tractable for several classes of generative models.

**Related work.** Within the generative modeling community, there are a wealth of fine-tuning methods that incorporate preferences at the level of individual samples through a user-specified reward (Christiano et al., 2017; Rafailov et al., 2023; Uehara et al., 2024; Domingo-Enrich et al., 2025). None of these methods solves problem (1), which imposes a hard constraint at the distribution level.

Two prior works (Khalifa et al., 2021; Shen et al., 2024) propose fine-tuning procedures for distribution level constraints, but each applies to a single model class. Khalifa et al. (2021), the most similar to the present work, fine-tunes autoregressive language models to match distributional constraints with an algorithm similar to CGM-reward. Shen et al. (2024) propose a method for balancing class proportions in diffusion models that relies upon optimal transport. Compared to the present work, neither method reduces a majority of calibration error, and Khalifa et al. (2021) demonstrates their algorithm only for low-dimensional ($<10$) constraints.

Lastly, we clarify that our definition of generative model calibration differs from the definition of the same term adopted in the setting of supervised learning (e.g., Lichtenstein et al., 1977; Dawid, 1982; Naeini et al., 2015; Guo et al., 2017; Vaicenavicius et al., 2019). In this setting, calibration means that the conditional expectation of the response given the model prediction is equal to the prediction i.e., on average, the predictor is correct. Appendix A gives an extended discussion of related work.

## 2 CALIBRATING GENERATIVE MODELS WITH CGM-RELAX AND REWARD

The calibration problem is challenging for non-trivial generative models because both the objective and the calibration constraint in equation (1) are defined by intractable expectations. To address this problem, we propose two alternative objectives whose *unconstrained* optima approximate the solution to (1). These objectives still involve expectations under $p_\theta$, but we show how to compute unbiased estimates of their gradients, which permits their minimization by stochastic optimization.

We call our algorithms optimizing the two surrogate loss functions CGM-relax and CGM-reward (Algorithms 1 and 2, respectively). These algorithms require only that one can draw samples $\boldsymbol{x} \sim p_\theta$ and compute $p_\theta(\boldsymbol{x})$ and $\nabla_\theta \log p_\theta(\boldsymbol{x})$.

### 2.1 THE RELAX LOSS

The relax loss avoids the intractability of imposing the calibration constraint exactly by replacing it with a constraint violation penalty

$$\mathcal{L}^{\text{relax}}(\theta) := \underbrace{\| \mathbb{E}_{p_\theta}[\boldsymbol{h}(\boldsymbol{x})] - \boldsymbol{h}^* \|^2}_{\mathcal{L}^{\text{viol}}} + \lambda \underbrace{\mathrm{D}_{\mathrm{KL}}\left(p_\theta \parallel p_{\theta_{\text{base}}}\right)}_{\mathcal{L}^{\text{KL}}}, \tag{2}$$

where $\lambda > 0$ is a hyperparameter that trades off between satisfying the calibration constraint and minimizing the KL divergence. In our experiments we choose $\lambda$ by a grid-search. In the limit as $\lambda \to 0$, $\mathcal{L}^{\text{viol}}$ is the dominant term in the relax loss, and we expect the minimizer of (2) to approach the solution of the calibration problem (1).

Suppose we have $M$ independent samples $\{\boldsymbol{x}_m\}_{m=1}^M$ from $p_\theta$. To estimate $\mathcal{L}^{\text{relax}}$ we separately estimate the KL divergence ($\mathcal{L}^{\text{KL}}$) by

$$\widehat{\mathcal{L}}^{\text{KL}} := \frac{1}{M} \sum_{m=1}^M \log \frac{p_\theta(\boldsymbol{x}_m)}{p_{\theta_{\text{base}}}(\boldsymbol{x}_m)},$$

and the constraint penalty ($\mathcal{L}^{\text{viol}}$) by

$$\widehat{\mathcal{L}}^{\text{viol}} := \left\| \frac{1}{M} \sum_{m=1}^M \boldsymbol{h}(\boldsymbol{x}_m) - \boldsymbol{h}^* \right\|^2 - \frac{1}{M(M-1)} \sum_{m=1}^M \left\| \boldsymbol{h}(\boldsymbol{x}_m) - \frac{1}{M} \sum_{m'=1}^M \boldsymbol{h}(\boldsymbol{x}_{m'}) \right\|^2, \tag{3}$$

where the second term is a bias correction. Combining these estimators yields our overall estimator for the relax objective, $\widehat{\mathcal{L}}^{\text{relax}} = \widehat{\mathcal{L}}^{\text{viol}} + \lambda \widehat{\mathcal{L}}^{\text{KL}}$. Appendix B.1 shows $\widehat{\mathcal{L}}^{\text{relax}}$ is unbiased for $\mathcal{L}^{\text{relax}}$.

### 2.2 THE REWARD LOSS

The reward loss avoids the intractability of imposing the calibration constraint exactly by leveraging a connection between the calibration problem (1) and the *maximum entropy problem* (Jaynes, 1957;

Kullback, 1959; Csiszár, 1975). We first introduce the maximum entropy problem and then show how to approximate its solution with samples from $p_{\theta_{\text{base}}}$. Lastly, we propose the reward loss as a divergence to this approximate solution and describe connections to reward fine-tuning.

**Maximum entropy problem.** The maximum entropy problem solves

$$\underset{p \in \mathcal{P}(p_{\theta_{\text{base}}})}{\arg\min} \ \mathrm{D}_{\mathrm{KL}}\left(p \parallel p_{\theta_{\text{base}}}\right), \quad \text{such that } \mathbb{E}_p[\boldsymbol{h}(\boldsymbol{x})] = \boldsymbol{h}^* \tag{4}$$

where $\mathcal{P}(p)$ is the collection of probability distributions that have a density with respect to $p$. The calibration problem and the maximum entropy problem differ only in their domains: the domain of the calibration problem is generative models $p_\theta$ in the same parametric class as $p_{\theta_{\text{base}}}$, rather than the nonparametric set $\mathcal{P}(p_{\theta_{\text{base}}})$. Despite this difference, we obtain an alternative objective by considering the solution to (4). The following theorem characterizes this solution.

**Theorem 2.1.** *Suppose $\boldsymbol{h}^*$ lies in the relative interior of the set of all attainable moments of $\boldsymbol{h}$ by distributions $P \in \mathcal{P}(p_{\theta_{base}})$. Then there exists a vector $\boldsymbol{\alpha}^*$ for which*

$$p_{\boldsymbol{\alpha}^*}(\boldsymbol{x}) \propto p_{\theta_{base}}(\boldsymbol{x}) \exp\left\{r_{\boldsymbol{\alpha}^*}(\boldsymbol{x})\right\}, \quad r_{\boldsymbol{\alpha}}(\boldsymbol{x}) := \boldsymbol{\alpha}^\top \boldsymbol{h}(\boldsymbol{x}) \tag{5}$$

*satisfies $\mathbb{E}_{p_{\boldsymbol{\alpha}^*}}[\boldsymbol{h}(\boldsymbol{x})] = \boldsymbol{h}^*$ and $p_{\boldsymbol{\alpha}^*}$ is the solution to the maximum entropy problem.*

Appendix C provides further exposition to the maximum entropy problem as well as a proof.

The domain of the calibration problem may not contain $p_{\boldsymbol{\alpha}^*}$. However, if the class of generative models is sufficiently expressive, its optimum $p_{\theta^*}$ will be close to $p_{\boldsymbol{\alpha}^*}$. This observation suggests a second way to remove the constraint in equation (1): fine-tune $p_\theta$ to minimize a divergence to $p_{\boldsymbol{\alpha}^*}$.

In Appendix C.3 we demonstrate that a similar statement holds for the relax loss: when $p_\theta$ is sufficiently expressive, the optimum of the relax loss is close to $p_\lambda(\boldsymbol{x}) \propto p_{\theta_{\text{base}}}(\boldsymbol{x}) \exp\left\{r_{\boldsymbol{\alpha}_\lambda}(\boldsymbol{x})\right\}$, where $\boldsymbol{\alpha}_\lambda$ depends on the regularization strength $\lambda > 0$ and is not generally equal to $\boldsymbol{\alpha}^*$. However, as $\lambda \to 0$, $\|\boldsymbol{\alpha}_\lambda - \boldsymbol{\alpha}^*\|$ approaches zero at rate $\lambda$. This formalizes our intuition from Section 2.1 that as $\lambda \to 0$, the relax loss solves the calibration problem.

**Estimating $p_{\boldsymbol{\alpha}^*}$.** The idea of minimizing a divergence to $p_{\boldsymbol{\alpha}^*}$ introduces a challenge: even when the solution $p_{\boldsymbol{\alpha}^*}$ to the maximum entropy problem (4) exists, its parameters $\boldsymbol{\alpha}^*$ are not immediately computable. To address this challenge, we leverage Wainwright & Jordan (2008, Theorem 3.4), which states that when the assumptions of Theorem 2.1 hold and there are no redundancies among the constraints $\boldsymbol{h}$, solving problem (4) is equivalent to computing

$$\underset{\boldsymbol{\alpha}}{\arg\max} \ \boldsymbol{\alpha}^\top \boldsymbol{h}^* - \log\left(\int \exp\{r_{\boldsymbol{\alpha}}(\boldsymbol{x})\} p_{\theta_{\text{base}}}(\boldsymbol{x}) d\boldsymbol{x}\right). \tag{6}$$

In other words, by solving (6) one obtains the parameters $\boldsymbol{\alpha}^*$ of $r_{\boldsymbol{\alpha}}(\boldsymbol{x})$, which then determine the solution $p_{\boldsymbol{\alpha}^*}$ to the maximum entropy problem up to a normalizing constant.

However, a difficulty of solving (6) is that the integral in the second term will be intractable for most generative models. We propose drawing $N$ independent samples $\{\boldsymbol{x}_n\}_{n=1}^N$ from $p_{\theta_{\text{base}}}$ and replacing the integral with respect to $p_{\theta_{\text{base}}}$ by the integral with respect to the empirical distribution that places probability mass $N^{-1}$ on each of the samples $\boldsymbol{x}_n$,

$$\widehat{\boldsymbol{\alpha}}_N = \underset{\boldsymbol{\alpha}}{\arg\max} \ \boldsymbol{\alpha}^\top \boldsymbol{h}^* - \log\left(\frac{1}{N}\sum_{n=1}^N \exp\{r_{\boldsymbol{\alpha}}(\boldsymbol{x}_n)\}\right). \tag{7}$$

Problem (7) is concave, and when $\widehat{\boldsymbol{\alpha}}_N$ is well-defined (see Appendix C.2), it can be found by convex solvers. We demonstrate in Appendix C.4 that $\widehat{\boldsymbol{\alpha}}_N$ converges to $\boldsymbol{\alpha}^*$ in the limit of many samples $N$, and we derive an expression for the asymptotic variance of $\widehat{\boldsymbol{\alpha}}_N$.

**$\mathcal{L}^{\text{reward}}$ and its estimation.** With $\widehat{\boldsymbol{\alpha}}_N$ in hand, we formulate our second loss as a divergence to $p_{\widehat{\boldsymbol{\alpha}}_N}$. For simplicity and because it avoids the requirement to compute the normalizing constant of $p_{\widehat{\boldsymbol{\alpha}}_N}$, we again choose the KL divergence. In particular, we define the reward loss $\mathcal{L}^{\text{reward}}$ to be

$$\mathcal{L}^{\text{reward}}(\theta) = \mathrm{D}_{\mathrm{KL}}\left(p_\theta \parallel p_{\widehat{\boldsymbol{\alpha}}_N}\right) = \underbrace{\mathbb{E}_{p_\theta}[\log p_\theta(\boldsymbol{x})/p_{\theta_{\text{base}}}(\boldsymbol{x})]}_{\mathcal{L}^{\mathrm{KL}} = \mathrm{D}_{\mathrm{KL}}\left(p_\theta \parallel p_{\theta_{\text{base}}}\right)} + \underbrace{\mathbb{E}_{p_\theta}[-r_{\widehat{\boldsymbol{\alpha}}_N}(\boldsymbol{x})]}_{\mathcal{L}^{\mathrm{r}}} + C, \tag{8}$$

**Algorithm 1** CGM-relax fine-tuning

**Require:** $p_{\theta_{\text{base}}}, \boldsymbol{h}(\cdot), \boldsymbol{h}^*, M$, and $\lambda$

   ▷ Initialize and optimize
$p_\theta \leftarrow p_{\theta_{\text{base}}}$
**while** not converged **do**
   ▷ Sample and compute weights
   $\boldsymbol{x}_1, \ldots, \boldsymbol{x}_M \overset{i.i.d.}{\sim} p_{\texttt{stop-grad}(\theta)}$
   $w_m \leftarrow p_\theta(\boldsymbol{x}_m) / p_{\texttt{stop-grad}(\theta)}(\boldsymbol{x}_m)$

   ▷ KL loss with LOO baseline
   $l_m \leftarrow \log p_{\texttt{stop-grad}(\theta)}(\boldsymbol{x}_m)/p_{\theta_{\text{base}}}(\boldsymbol{x}_m)$
   $l_m^{\text{LOO}} \leftarrow l_m - \frac{1}{M-1}\sum_{m'\neq m} l_{m'}$
   $\widehat{\mathcal{L}}^{\text{KL}} \leftarrow \frac{1}{M}\sum w_m l_m^{\text{LOO}}$

   ▷ Constraint violation loss
   $\boldsymbol{h}_m \leftarrow w_m(\boldsymbol{h}(\boldsymbol{x}_m) - \boldsymbol{h}^*)$
   $\widehat{\mathcal{L}}^{\text{viol}} \leftarrow \|\frac{1}{M}\sum \boldsymbol{h}_m\|^2 - \frac{1}{M}\widehat{\text{Var}}[\boldsymbol{h}_{1:M}],$
   $\widehat{\text{Var}}[\boldsymbol{h}_{1:M}] = \frac{1}{M-1}\sum \|\boldsymbol{h}_m - \frac{1}{M}\sum \boldsymbol{h}_{m'}\|^2$

   ▷ Total loss and update
   $\widehat{\mathcal{L}}^{\text{relax}} = \lambda\widehat{\mathcal{L}}^{\text{KL}} + \widehat{\mathcal{L}}^{\text{viol}}$
   $\theta \leftarrow \text{gradient-step}(\theta, \nabla_\theta \widehat{\mathcal{L}}^{\text{relax}})$

---

**Algorithm 2** CGM-reward fine-tuning

**Require:** $p_{\theta_{\text{base}}}, \boldsymbol{h}(\cdot), \boldsymbol{h}^*, M, N$
   ▷ Estimate $\boldsymbol{\alpha}^*$ for reward
   $\boldsymbol{x}_1, \ldots, \boldsymbol{x}_N \overset{i.i.d.}{\sim} p_{\theta_{\text{base}}}$
   $\widehat{\boldsymbol{\alpha}}_N \leftarrow \arg\max \boldsymbol{\alpha}^\top \boldsymbol{h}^* - \log\sum\exp\{r_{\boldsymbol{\alpha}}(\boldsymbol{x}_n)\}$

   ▷ Initialize and optimize
$p_\theta \leftarrow p_{\theta_{\text{base}}}$
**while** not converged **do**
   ▷ Sample and compute weights
   $\boldsymbol{x}_1, \ldots, \boldsymbol{x}_M \overset{iid}{\sim} p_{\texttt{stop-grad}(\theta)}$
   $w_m \leftarrow p_\theta(\boldsymbol{x}_m) / p_{\texttt{stop-grad}(\theta)}(\boldsymbol{x}_m)$

   ▷ KL loss with LOO baseline
   $l_m \leftarrow \log p_{\texttt{stop-grad}(\theta)}(\boldsymbol{x}_m)/p_{\theta_{\text{base}}}(\boldsymbol{x}_m)$
   $l_m^{\text{LOO}} \leftarrow l_m - \frac{1}{M-1}\sum_{m'\neq m} l_{m'}$
   $\widehat{\mathcal{L}}^{\text{KL}} \leftarrow \frac{1}{M}\sum w_m l_m^{\text{LOO}}$

   ▷ Negative reward with LOO baseline
   $r_m^{\text{LOO}} \leftarrow r_{\widehat{\boldsymbol{\alpha}}}(\boldsymbol{x}_m) - \frac{1}{M-1}\sum_{m'\neq m} r_{\widehat{\boldsymbol{\alpha}}}(\boldsymbol{x}_{m'})$
   $\widehat{\mathcal{L}}^{\text{r}} \leftarrow -\frac{1}{M}\sum w_m r_m^{\text{LOO}}$

   ▷ Total loss and update
   $\widehat{\mathcal{L}}^{\text{reward}} = \widehat{\mathcal{L}}^{\text{KL}} + \widehat{\mathcal{L}}^{\text{r}}$
   $\theta \leftarrow \text{gradient-step}(\theta, \nabla_\theta \widehat{\mathcal{L}}^{\text{reward}})$

where $C = \mathbb{E}_{p_{\theta_{\text{base}}}}[\exp\{r_{\widehat{\boldsymbol{\alpha}}_N}(\boldsymbol{x})\}]$ is a normalizing constant that does not depend on $\theta$.

We call $r_{\boldsymbol{\alpha}}(\boldsymbol{x})$ the *reward* and $\mathcal{L}^{\text{reward}}$ the reward loss because $\mathcal{L}^{\text{reward}}$ coincides with the objective of reward fine-tuning algorithms. The goal of reward fine-tuning is to fine-tune the base generative model $p_{\theta_{\text{base}}}$ to a tilted version of itself, where the tilt is determined by a so-called reward $r(\boldsymbol{x})$.

Just as for $\mathcal{L}^{\text{KL}}$ in the relax loss (2), Monte Carlo sampling provides an unbiased estimate of $\mathcal{L}^{\text{r}}$. This, in turn, gives us an unbiased estimate of the reward loss $\mathcal{L}^{\text{reward}}$.

## 2.3 GRADIENT ESTIMATION

We next describe our approach to computing unbiased estimates for the gradients of $\mathcal{L}^{\text{relax}}(\theta)$ and $\mathcal{L}^{\text{reward}}(\theta)$. This enables optimization of the relax and reward losses via stochastic optimization. We leverage the score function gradient estimator (Williams, 1992; Ranganath et al., 2014) and a similar importance sampling-based gradient estimator for the relax loss.

**Score function gradient estimation.** The primary challenge to computing gradients is the inability to directly exchange the order of the gradients and expectations taken with respect to $\theta$. That is, because $\nabla_\theta \mathcal{L}(\theta) = \nabla_\theta \mathbb{E}_{p_\theta}[f(\boldsymbol{x}, \theta)] \neq \mathbb{E}_{p_\theta}[\nabla_\theta f(\boldsymbol{x}, \theta)]$, $\nabla_\theta \mathcal{L}(\theta)$ can not in general be usefully approximated by $M^{-1}\sum \nabla_\theta f(\boldsymbol{x}_m, \theta)$ from samples $\boldsymbol{x}_m$ of $p_\theta$. To address this challenge, we observe

$$\mathcal{L}(\theta) = \mathcal{L}(\theta, \theta') := \mathbb{E}_{p_{\theta'}}\left[\frac{p_\theta(\boldsymbol{x})}{p_{\theta'}(\boldsymbol{x})} f(\boldsymbol{x}, \theta)\right], \tag{9}$$

for any set of model parameters $\theta'$. Since the expectation in equation (9) does not depend on $\theta$, we *can* approximate its gradient with Monte Carlo samples from $p_{\theta'}$. The density ratio $p_\theta(\boldsymbol{x}_m)/p_{\theta'}(\boldsymbol{x}_m)$ in equation (9) can be understood as the weights of an importance sampling estimate against target $p_\theta$ with proposal $p_{\theta'}$.

To estimate the gradients of the relax and reward losses, we choose proposal equal to the current model, i.e., $\theta'=\theta$. In this case, the importance weight is equal to 1 while its gradient is the "score" function $(\nabla_\theta p_\theta(\boldsymbol{x}_m))/p_\theta(\boldsymbol{x}_m) = \nabla \log p_\theta(\boldsymbol{x})$, which is nonzero in general (Mohamed et al., 2020). Algorithms 1 and 2 each demonstrate an implementation that computes these weights with a copy of the parameters $\theta$ detached from the computational graph, which we denote by $\texttt{stop-grad}(\theta)$.

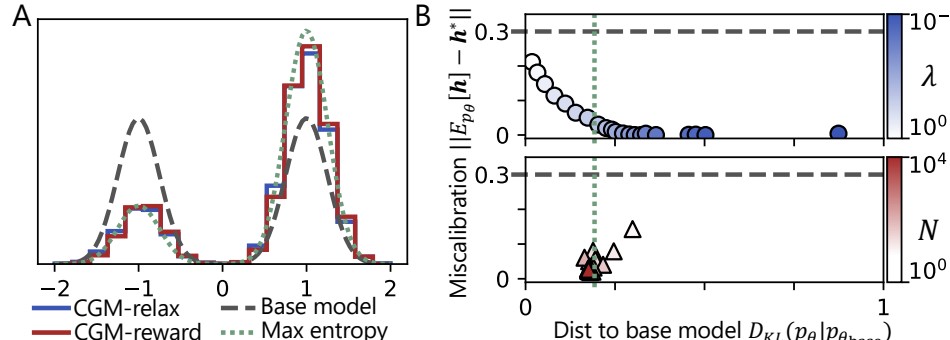

Figure 1: Calibrating mixture proportions in a diffusion model targeting a 1D GMM. **A**: The CGM-relax and CGM-reward solutions closely approximate the maximum entropy solution. **B**: (top) The CGM-relax regularization parameter $\lambda$ trades off between constraint satisfaction and closeness to the base model (bottom) CGM-reward is accurate when enough samples $N$ are used to estimate $\boldsymbol{\alpha}^*$.

Although the term $\mathcal{L}^{\mathrm{viol}}$ that appears in the relax loss is not of the form $\mathbb{E}_{p_\theta}[f(\boldsymbol{x}, \theta)]$, we can still construct an unbiased estimate to its gradient using importance sampling (see Appendix B.2).

Although score function gradient estimates are known to suffer from high variance (Mohamed et al., 2020), we show that, paired with variance reduction strategies (Appendix B.2), they perform well even in problem settings with high-dimensional latent variables, such as diffusion models and masked language models (Section 4.1).

## 3 SIMULATIONS: DETERMINING WHEN CGM THRIVES AND STRUGGLES

To understand the success and failure cases of CGM, we perform evaluations in a tractable "toy" setting. This setting allows us to understand the role of the CGM hyperparameters $\lambda$ and $N$, and to test CGM in challenging problem settings, including rare events and high-dimensional constraints.

We provide additional discussion of our simulation experiments in Appendix D, including an overview of diffusion models in Appendix D.1 and a comparison between CGM and the Augmented Lagrangian (AL) algorithm (Hestenes, 1969) in Appendix D.4.

**Simulation setup and evaluation.** We consider fine-tuning a diffusion model targeting a Gaussian mixture model (GMM) to reweight the mixture proportions of each mode. Here, $p_\theta(\boldsymbol{x})$ is a generative model of continuous paths $\boldsymbol{x} = (\boldsymbol{x}(t))_{t \in [0,1]}$, whose evolution is described by a stochastic differential equation (SDE). To sample from $p_\theta$, one first draws $\boldsymbol{x}(0)$ from the tractable initial distribution and then simulates the SDE starting from time $t{=}0$ up until time $t{=}1$.

Evaluating CGM on a diffusion model whose terminal distribution is a GMM has several advantages. First, we may choose the base diffusion model so that the final marginal $p_\theta(\boldsymbol{x}(1))$ exactly matches the target GMM (Anderson, 1982; Song et al., 2021); this enables us to focus solely on calibration rather than fitting the base model. And since the calibration constraint depends on the path only at time $t{=}1$, we can compute the KL divergence of the maximum entropy solution to the base model, and thereby measure the suboptimality of the solutions produced by CGM.

**Selecting hyperparameters for CGM-relax and CGM-reward.** We first initialize our base model $p_{\theta_{\mathrm{base}}}$ such that $p_{\theta_{\mathrm{base}}}(\boldsymbol{x}(1))$ is a one-dimensional Gaussian mixture with two well separated modes (Figure 1 A). We define the calibration problem with statistic $\boldsymbol{h}(\boldsymbol{x}) = \mathbb{1}\{\boldsymbol{x}(1) > 0\}$ to upweight the mass in right mode from $\mathbb{E}_{p_{\theta_{\mathrm{base}}}}[\boldsymbol{h}(\boldsymbol{x}(1))] = 0.5$ to $\boldsymbol{h}^* = 0.8$.

For CGM-relax we observe that the regularization parameter $\lambda$ trades off between constraint satisfaction and deviation from the base model (Figure 1B). With large $\lambda$ the model deviates little from $p_{\theta_{\mathrm{base}}}$ but does not satisfy the constraint, whereas for small $\lambda$ the model satisfies the constraint but has KL to $p_{\theta_{\mathrm{base}}}$ that exceeds that of the maximum entropy solution. For CGM-reward, we observe that increasing $N$ results in more accurate recovery of the variational parameters $\boldsymbol{\alpha}^*$ and thereby a better approximation to the maximum entropy solution. For appropriate hyperparameters, both solve the calibration problem to high accuracy. In the remaining experiments, we perform grid-search to select $\lambda$ in CGM-relax and use $N = 10^5$ samples to estimate $\boldsymbol{\alpha}^*$ in CGM-reward.

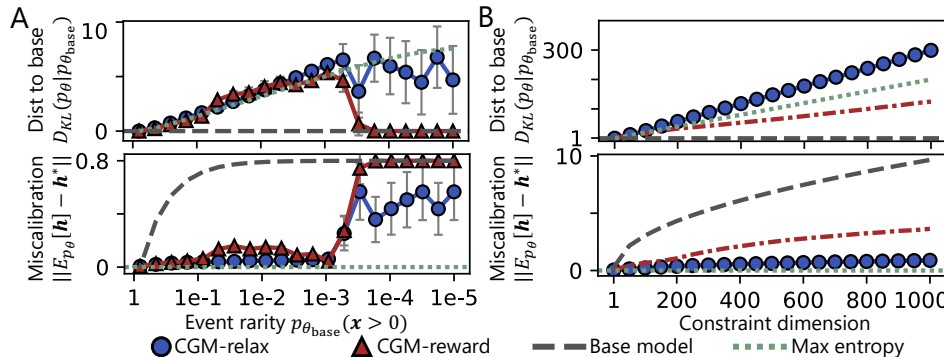

Figure 2: **A**: CGM effectively upweights the probability of a rare mode in a 1D GMM. **B**: CGM-relax calibrates the base model to up to $10^3$ constraints, whereas CGM-reward is not well-defined for $>30$ constraints. When $\widehat{\boldsymbol{\alpha}}_N$ is fixed to $\boldsymbol{\alpha}^*$ (red dashed line), CGM-relax outperforms CGM-reward.

**Upweighting rare events.** Increasing the proportion of generations belonging to rare classes is central to applications including protein ensemble modeling (Lewis et al., 2025) and reinforcement learning (O'Kelly et al., 2018). To assess the performance of CGM in this setting, we consider variations of the GMM reweighting problem in which we consider increasingly small mixture proportions $\pi = \mathbb{E}_{p_{\theta_{\text{base}}}}[\boldsymbol{h}(\boldsymbol{x}(1))]$ of the mode to upweight by calibration. We vary $\pi$ from $\boldsymbol{h}^* = 0.8$ (already calibrated) to $10^{-5}$ and use a constant batch size $M = 10^2$.

We find that both algorithms perform well with base model event rarity as small as $\pi = 10^{-3}$; the majority of miscalibration is reduced without divergence from the base model much larger than the maximum entropy solution (Figure 2A). This is surprising since for $\pi = 10^{-3}$, most batches sampled from $p_{\theta_{\text{base}}}$ contain no samples belonging to the second mode. Performance degrades below this threshold, but we suspect larger batch sizes would allow upweighting even rarer events.

**Scalability to high-dimensional models and constraints.** We next evaluate how performance depends on the dimensionality, $k$, of the GMM and the constraint. We take the base model to be a product of one-dimensional GMMs with marginals as in Figure 1A. For the calibration constraint, we choose the $\boldsymbol{h}(\boldsymbol{x}) = [\mathbb{1}\{\boldsymbol{x}(1)[1] > 0\}, \ldots, \mathbb{1}\{\boldsymbol{x}(1)[k] > 0\}]$, where $\boldsymbol{x}(1)[i]$ is the $i$th dimension of $\boldsymbol{x}(1)$ and $\boldsymbol{h}^* = [0.8, \ldots, 0.8]$. Since both the base model $p_{\theta_{\text{base}}}$ and maximum entropy solution $p_{\boldsymbol{\alpha}^*}$ are independent across dimension, the KL distance between these two distributions grows linearly in dimension. The multimodality of this model, with $2^k$ modes, mimics the multimodality of practical generative models. We perform CGM-relax and CGM-reward with batch size $M = 10^4$.

In this high-dimensional regime, significant discrepancies emerge between CGM-relax and CGM-reward (Figure 2B). CGM-relax consistently eliminates the majority of constraint violation up to $k=10^3$, albeit with a non-trivial excess KL divergence to $p_{\theta_{\text{base}}}$ compared to the maximum entropy solution $p_{\boldsymbol{\alpha}^*}$ that increases linearly with dimension. Although CGM-reward performs well for low-dimensional constraints ($<10$), we find that the empirical maximum entropy problem (7) is infeasible with high probability for $>30$ constraints. In fact, even when $\widehat{\boldsymbol{\alpha}}_N$ is fixed to its oracle value $\boldsymbol{\alpha}^*$ (Figure 2B), CGM-relax still outperforms CGM-reward.

## 4 CASE-STUDIES WITH DIVERSE MODELS, DATA, AND CONSTRAINTS

We evaluate the capacity of CGM-reward and CGM-relax to solve practical calibration problems through three applications involving diverse model, data, and constraint types. Section 4.1 calibrates a diffusion model (Lin et al., 2024a) and a masked language model (Hayes et al., 2025) of protein structure to more closely match statistics of natural proteins. Section 4.2 calibrates a normalizing flow model (Zhai et al., 2025) of images to reduce class imbalances on the basis of LLM image-to-text annotations. Lastly, Section 4.3 calibrates a small autoregressive LM to eliminate gender bias in generated children's stories (Eldan & Li, 2023).

Across all examples, CGM reduces the majority of calibration error without significantly degrading the quality of generations. Consistent with our results in Section 3 we find that optimally-tuned CGM-relax outperforms CGM-reward, which falls short of meeting the calibration constraints.

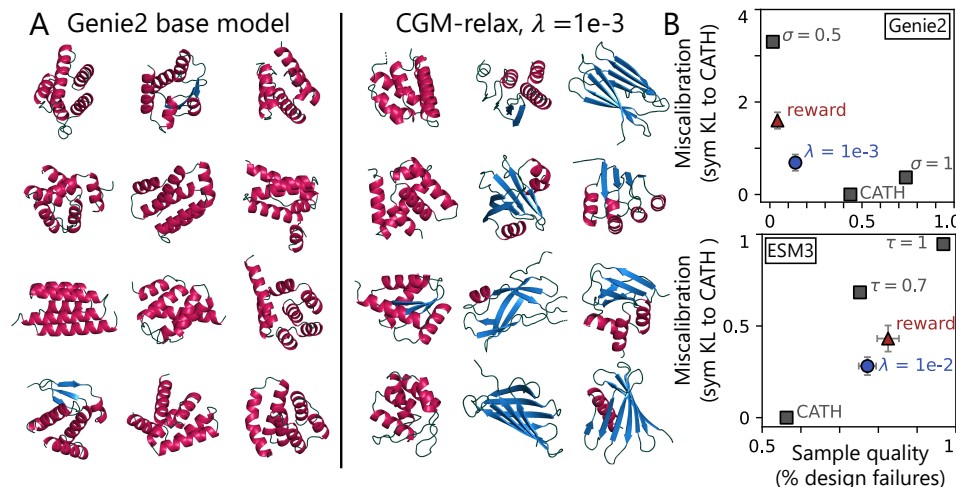

Figure 3: **A**: Samples from the Genie2 protein generative models before and after calibration with CGM-relax ($\lambda{=}10^{-3}$). **B**: CGM-relax reduces the distance of secondary structure content to natural proteins by $>4$ times for Genie2 and $>2$ times for ESM3 while maintaining biophysical plausibility.

**Baselines.** Only two prior works have proposed algorithms that intend to solve the calibration problem. Khalifa et al. (2021) propose a method for LLMs that we compare to in Section 4.3. Second, Shen et al. (2024) propose a method for class-balancing in diffusion models. However, it assumes an existing probabilistic classifier and so is not applicable in our setting.

**Compute cost.** Each experiment is run on a single H100 GPU. We provide additional details regarding our experimental setup in Appendix E.

### 4.1 CALIBRATING PROTEIN DESIGN MODELS TO MATCH STATISTICS OF NATURAL PROTEINS

Diffusion generative models have become a central tool in protein design (Trippe et al., 2023; Watson et al., 2023). However, heuristics such as reduced noise during sampling (see e.g., Yim et al., 2023) have been necessary to ensure a high proportion of the sampled structures are biophysically plausible. These heuristics substantially reduce the diversity of samples compared to proteins found in nature and thereby pose a trade-off between reliability and diversity. For two such protein design models, we investigate whether this trade-off can be mitigated by calibrating the models to match the secondary structure composition of natural proteins.

**Protein models Genie2 and ESM3 and their miscalibration.** The two protein design models we consider are (1) Genie2 (Lin et al., 2024a), a 15M parameter equivariant diffusion model, and (2) ESM3-open (Hayes et al., 2025), a 1.4B parameter masked language model on tokenized representations of protein backbones. For each model, we generate protein backbones consisting of 100 amino acids i.e., residues. Both Genie2 and ESM3-open suffer low diversity compared to natural protein domains in the CATH dataset (Sillitoe et al., 2021); specifically, they produce few generations with high beta-strand content (Figure 3A). Beta strands, along with alpha helices and loops, constitute what is known as a protein's secondary structure.

**Calibration constraints on secondary structure diversity.** To represent protein secondary structure as a calibration constraint, we use the empirical bivariate cumulative density function (CDF) of the fraction of residues in alpha-helical and beta-strand segments. We place up to $d = 99$ cutoff pairs $(\tau_{\alpha,i}, \tau_{\beta,i}) \in [0,1]^2$ and define a $d$-dimensional indicator vector $\boldsymbol{h}(\boldsymbol{x})$ with components $\boldsymbol{h}(\boldsymbol{x})[i] = \mathbb{1}\{ f_\alpha(\boldsymbol{x}) \leq \tau_{\alpha,i}, \ f_\beta(\boldsymbol{x}) \leq \tau_{\beta,i} \}, \ i = 1,\ldots,d$, where $f_\alpha(\boldsymbol{x})$ and $f_\beta(\boldsymbol{x})$ are the secondary-structure fractions of protein structure $\boldsymbol{x}$. We set the calibration target $\boldsymbol{h}^*$ to the corresponding values of the CATH empirical bivariate CDF at these cutoffs.

**Results.** Performing calibration with CGM-relax yields a nearly fivefold improvement in the diversity of sampled protein structures for Genie2 and a twofold improvement for ESM3-open, as quantified by the symmetrized KL distance between the secondary structure distributions of the generative models and CATH domains (Figure 3B). This improvement comes at the cost of an increased proportion of 'design failures', as defined in Appendix E.1. The ESM3-open base model

Figure 4: Generations from the conditional TarFlow model (Zhai et al., 2025) before and after calibration with CGM-relax ($\lambda = 10^{-4}$). CGM reweights the proportions of animals generated and produces realistic images. Some visual artifacts exist after calibration (see e.g., fox).

generates a high proportion of design failures compared to Genie2 (consistent with Xiong et al. (2025), for example) and this fraction increases slightly upon calibration with CGM.

CGM-reward achieves more modest improvements in secondary structure diversity, which may in part be due to difficulty in computing $\widehat{\boldsymbol{\alpha}}_N$. In order for equation (7) to be feasible with $N = 2.5 \times 10^3$ samples, we need to reduce the number of cutoff pairs from 99 to 15. CGM-reward fine-tuning reduces the symmetrized KL distance to CATH by two times for Genie2 and 1.6 times for ESM3-open. However, for Genie2, CGM-reward also produces fewer design failures than CGM-relax.

The gains in secondary structure diversity achieved by CGM cannot be obtained by simply increasing the sampling noise of Genie2 or the sampling temperature of ESM3. In Figure 3B, we show that increasing the sampling noise of Genie2 to $\sigma = 1$ improves structure diversity, but at the cost of 5.3 times more design failures (failure rate 74%) than CGM. The same is true for ESM3 with increased sampling temperature $\tau = 1$, which yields a 1.3 times higher failure rate of 97%.

## 4.2 Calibrating Class Proportions in a Conditional Flow Model

We next demonstrate that CGM is capable of effectively calibrating state-of-the-art normalizing flow models. Normalizing flows generate samples $\boldsymbol{x} \in \mathbb{R}^k$ according to $\boldsymbol{x} = f_\theta^{-1}(\boldsymbol{\epsilon})$, where $\boldsymbol{\epsilon} \sim p_\epsilon$ is a distribution from which sampling is tractable and $f_\theta(\boldsymbol{x})$ is a map that is invertible in $\boldsymbol{x}$ for each $\theta$ (Tabak & Vanden-Eijnden, 2010; Rezende & Mohamed, 2015). By the change-of-variable formula, the density of $\boldsymbol{x}$ is $p_\epsilon(f_\theta(\boldsymbol{x}))|\det(df_\theta(\boldsymbol{x})/d\boldsymbol{x})|$. This expression enables computation of exact likelihoods for maximum likelihood training and calibration.

For our calibration problem, we consider the 463M-parameter TarFlow model (Zhai et al., 2025), which parameterizes $f_\theta$ as an autoregressive vision transformer (Dosovitskiy et al., 2021) such that attention is performed over a sequence of image patches. We examine the model trained conditionally on the $256 \times 256$ AFHQ dataset (Choi et al., 2020), which consists of images of animals faces belonging to one of three classes: {cat, dog, wildlife}. The wildlife class is further comprised of {lion, tiger, fox, wolf, cheetah, leopard}. We observe that, conditional on the wildlife class, approximately 36% of generations from the TarFlow model are lions and very few (< 7% total) are foxes or wolves. We apply CGM to calibrate the conditional TarFlow model to generate samples containing animals from the wildlife class with equal proportions. For $h$, we query GPT o5-mini to classify each image as containing one of the six animals or None.

**Results.** We find CGM-relax reduces miscalibration to the base TarFlow model with little visible degradation of sample realism (Figure 4). CGM-relax ($\lambda = 10^{-4}$) reduces the total variation distance of animal proportions, as classified by an image-to-text model, to the uniform distribution from 0.306 to 0.108. However, the Fréchet inception distance (FID) to real images in the AFHQ wildlife class is larger for the calibrated model than for the base model (21.9 vs. 15.9). Since this metric is sensitive to class proportions, we evaluate the calibrated model on the training dataset after balancing classes. The discrepancy in FID can be explained by two types of visual artifacts introduced by calibration: some images depict animals outside the wildlife class ($\sim 8\%$) and some "blend" multiple animals. Appendix Figure 9 shows random samples from both models. The model fine-tuned with CGM-reward remains close to the base model but fails to reduce constraint violation.

## 4.3 Eliminating Profession-Specific Gender Imbalance in Children's Stories

As a third example, we calibrate a language model that generates short children's stories to remove gender bias. TinyStories-33M is an autoregressive transformer trained on children's stories

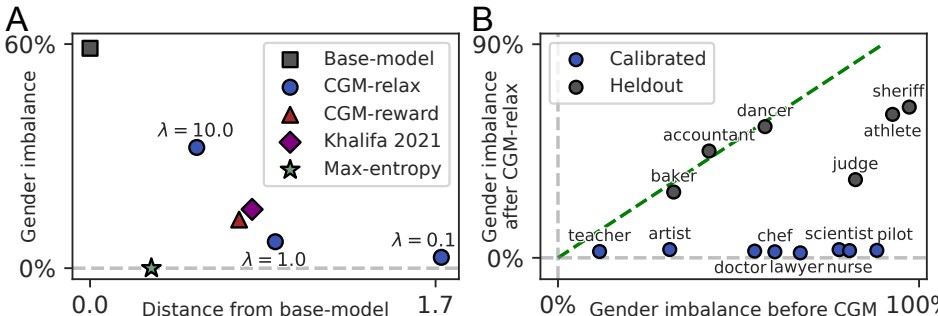

Figure 5: **A**: Gender imbalance and distance from base-model (symmetrized KL from pre-trained TinyStories-33M). **B**: Gender imbalance for professions included and heldout from calibration before and after CGM-relax ($\lambda = 0.1$). Points below the diagonal were improved by CGM.

generated by GPT-3.5 and GPT4 (Eldan & Li, 2023). We find significant imbalances in prompt-conditional generations that introduce a character's profession. For example, only 16% of stories beginning "Once upon a time there was a lawyer" feature a female lawyer, whereas 41% of U.S. attorneys were women in 2024 (American Bar Association, 2024).

**Gender parity as a calibration constraint and conditional calibration.** We evaluate whether CGM can eliminate profession-specific gender imbalance in stories completed from the prompt "Once upon a time there was a <profession>" across eight professions that exhibit gender bias under the base model: doctor, lawyer, teacher, pilot, chef, scientist, nurse, and artist. In contrast to earlier experiments, this requires *conditional* calibration: for each profession $i$ with prompt prompt$_i$, we aim to find $\theta$ such that $\mathbb{E}_{p_\theta}[\boldsymbol{h}(\boldsymbol{x}) \mid \text{prompt}_i] = 0$, where $\boldsymbol{x}$ represents a completed story, and $\boldsymbol{h}(\boldsymbol{x}) \in \{-1, 0, 1\}$ encodes the character's gender (male, ambiguous, or female, respectively). Rather than fine-tuning a separate model for each profession, we amortize training costs by fine-tuning a single model with the sum of CGM losses for each condition.

**Results on explicitly calibrated professions.** Both CGM-reward and CGM-relax reduce gender imbalance, as measured by the average absolute per-profession frequency difference (Figure 5A). As expected, decreasing the regularization strength $\lambda$ improves constraint satisfaction at the cost of greater distance to the base-model, as measured by symmetrized KL. Notably, even the least-regularized model attains a low symmetrized KL of 1.7, which corresponds to an average token log-probability difference of $< 0.01$ nats/token. Appendix E.4.4 provides example generations before and after fine-tuning showing no visible degradation in story quality.

Compared to Khalifa et al. (2021), CGM-reward yields a small but statistically significant improvement in both miscalibration and distance to the base model. CGM-relax reduces gender imbalance by over five times more than Khalifa et al. (2021) but deviates further from the base-model.

**Transference of calibration to heldout professions.** We evaluate how conditional calibration affects the calibration of "held-out" professions not considered during fine-tuning. Such generalization could be particularly valuable in applications where it is impractical to foresee and explicitly calibrate for every possible prompt. To evaluate this, we consider six held-out professions: sheriff, judge, accountant, dancer, athlete, and baker. While CGM does not result in gender parity for the held-out professions, the imbalance is significantly reduced for some (Figure 5B).

## 5 CONCLUSION

CGM-relax and CGM-reward provide practical approaches for calibrating generative models to satisfy distribution-level constraints. In applications to protein design, conditional image generation, and language modeling, CGM consistently reduces calibration error under hundreds of simultaneous constraints and in models with up to one billion parameters while preserving generation quality.

Still, our results highlight that the calibration problem is not yet solved. Current objectives leave residual error, especially in the rare-event setting that is especially relevant to protein structure modeling, for example. More broadly, the CGM framework is tied to models with tractable likelihoods, raising the challenge of extending calibration to VAEs, GANs, and other implicit models. These open questions point to calibration as a practical tool as well as a fruitful research direction.

## 6 ETHICS STATEMENT

This work develops algorithms for calibrating generative models by aligning distribution-level statistics to desired targets. Our motivation is to improve the fidelity of generative models across diverse domains, including, but not limited to, protein design, image generation, and language modeling. Potential ethical benefits include reducing harmful biases (e.g., gender imbalance in text outputs) and improving scientific utility (e.g., protein structure design). However, as is the case for all works that fine-tune generative models, our methods could also be misused to enforce constraints that amplify harmful or discriminatory content. We emphasize that the choice of constraints should be made responsibly, with careful attention to societal and scientific impacts.

All datasets used in this work are publicly available, and no sensitive personal data were employed.

## 7 REPRODUCIBILITY STATEMENT

We have taken multiple steps to ensure reproducibility of our theoretical and empirical results. **Theory.** All mathematical claims are supported by detailed theorem statements and proofs in Appendices B and C, and assumptions for all claims are clearly stated. **Algorithms.** We provide complete pseudocode for the algorithms we propose (Algorithms 1 and 2), including clear descriptions of loss estimation and gradient computation. Our implementations are entirely reproducible from these algorithms. **Experiments.** For each of our experiments in Sections 3 and 4, we specify the datasets, models, and calibration constraints in detail. Hyperparameter choices (e.g., model architecture, optimizer, learning rate, number of epochs, batch size $M$, regularization strength $\lambda$, sample size $N$) are reported in Appendices D and E. We include additional samples from the pre-trained (i.e., base) and fine-tuned (i.e., calibrated) generative models in these appendices. **Code.** Upon publication, we will release a public codebase implementing CGM-relax and CGM-reward for arbitrary generative models. We will include scripts to reproduce experimental results.

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

# APPENDIX CONTENTS

# A    EXTENDED DISCUSSION OF RELATED WORK

**The calibration problem.** Several previous works have proposed algorithms whose goal it is to impose distributional constraints on generative models. However, each of these methods applies only to specific model classes and either suffers from poor empirical performance or imposes constraint satisfaction during training time (rather than fine-tuning).

Most closely related to the present work, Khalifa et al. (2021) fine-tune autoregressive language models to match distributional constraints. Like CGM-reward, their approach also targets the maximum entropy solution (5), but through a different divergence; they choose the KL divergence in the "forward" direction, $D_{KL}(p_{\alpha^*} \| p_\theta)$, rather than in the "reverse" direction, $D_{KL}(p_\theta \| p_{\alpha^*})$, as in CGM-reward.

Empirically, the approximate solutions to the calibration problem (1) found by Khalifa et al. (2021) fall shorter of constraint satisfaction compared to CGM, particularly CGM-relax. Khalifa et al. (2021) achieves comparable, albeit slightly worse, performance to CGM-reward in the TinyStories gender rebalancing experiment (Section 4.3), reducing miscalibration by roughly 85%. CGM-relax, on the other hand, reduces constraint violation up to 98%.

In follow-up work, Go et al. (2023) propose an algorithm for aligning language models to a specified target distribution by minimizing an arbitrary $f$-divergence (including the forward and reverse KL divergence). One example they consider is when the target distribution is the maximum entropy distribution corresponding to some constraint functions; the choice of forward KL then reduces to Khalifa et al. (2021). However, they obtain $< 50\%$ reduction in constraint violation.

Shen et al. (2024) proposes a method for balancing class proportions in text-to-image diffusion models. They rely on an optimal transport objective that applies narrowly to diffusion models and find empirically their approach falls short of meeting desired class proportions.

In concurrent work, Cardei et al. (2025) impose constraints on discrete diffusion models at sampling time using an augmented Lagrangian method. Their algorithm involves simultaneously optimizing the model output and a set of Lagrange multipliers. Also concurrent to our work, Gutjahr et al. (2025) fine-tunes a diffusion generative model subject to inequality constraints on the expected value of a statistic to maximize an expected reward with a KL penalty to the base model. Their approach applies only to diffusion models and continuous normalizing flows.

**Incorporating distributional constraints during training.** Several other works have sought to impose distributional constraints during training time but differ from CGM in that they are not fine-tuning procedures and apply only to a specific model classes. Wu et al. (2020) propose a method for training generative adversarial networks (GANs) that includes a penalty term similar to $\mathcal{L}^{\mathrm{viol}}$ that encourages agreement with statistics of the training data. Zhu et al. (2024) solve for the maximum entropy model of short (length 7) protein sequences with expected "fitness" surpassing a fixed threshold. Khalafi et al. (2024) propose a primal-dual algorithm to enforce distributional constraints on diffusion models; their constraints, however, are specified at the level of entire distributions, rather than their moments. Friedrich et al. (2023) develop a training procedure for diffusion models that balances the conditional distributions of samples, given some attribute e.g., gender.

**Reward fine-tuning and conditional generation.** As we point out in Section 2.2, the idea of minimizing the KL divergence of the generative model to an exponential tilt of the base model (5) connects CGM to the rich research topic of reward fine-tuning. Reward fine-tuning algorithms, used in the contexts of reinforcement learning (Rafailov et al., 2023; Fan et al., 2023; Black et al., 2024; Wallace et al., 2024) and preference optimization (Tang, 2024; Uehara et al., 2024; Domingo-Enrich et al., 2025), minimize the same loss (8) as CGM-reward, but with $r_\alpha(\boldsymbol{x})$ replaced by a user-specified "reward". Unlike reward fine-tuning algorithms, though, CGM does not require a reward; rather, the constraints themselves act as the reward.

Conditional generation (Dhariwal & Nichol, 2021; Ho & Salimans, 2021; Denker et al., 2024) can also be viewed through the lens of model calibration, where the calibration constraint is the indicator function of the set $C$ from which one would like to sample $\boldsymbol{h}(\boldsymbol{x}) = \mathbb{1}\{\boldsymbol{x} \in C\}$ and $\boldsymbol{h}^*$, the target proportion of samples that belong to $C$, approaches 1. In this case the optimal variational parameter $\boldsymbol{\alpha}^*$ approach infinity, and the maximum entropy solution approaches $p_{\theta_{\mathrm{base}}}(\boldsymbol{x})\mathbb{1}\{\boldsymbol{x} \in C\}$.

**Calibration of molecular ensembles.** Computational methods for producing Boltzmann ensembles frequently fail to exactly align with experimental observables that measure ensemble averages; this misalignment can arise from inaccuracies in the energy functions used or insufficient sampling. Several works have sought to calibrate these ensembles to agree with ensemble observables. In the context of molecular dynamics simulations, (Różycki et al., 2011; Köfinger et al., 2019; Bottaro et al., 2020) leverage Theorem 2.1 to reweight Monte Carlo samples of molecular configurations to match experimental observations of ensemble averages. Lewis et al. (2025) consider a diffusion generative model approximation of protein structure ensembles and introduce an auxiliary training loss that resembles $\mathcal{L}^{\text{viol}}$, but they do not demonstrate whether this approach leads to a significant reduction in calibration error.

**Calibration in prediction problems.** Beyond generative modeling, calibration is a major topic in supervised machine-learning. In the context of classification the goal is to have, among a collection of predictions with a given class probability, the fraction of labels of that class in agreement with that prediction probability (Dawid, 1982). This can be obtained with post-hoc calibration procedures such as Platt scaling (Platt, 1999) or conformal methods (Shafer & Vovk, 2008) for more general prediction sets.

## B  CGM-RELAX AND CGM-REWARD ALGORITHMS

In this section, we provide further detail on the CGM-relax and CGM-reward algorithms. First, we show in Appendix B.1 that our estimates for the relax and reward losses are unbiased. In Appendix B.2 we then discuss how to compute our gradient estimates for the relax and reward losses, and we show they are unbiased.

Throughout this section we will make the following regularity assumptions on the generative model $p_\theta$ and the constraint functions $\boldsymbol{h}$.

**Assumption B.1** (Regularity of $p_\theta$). The functions $p_{\tilde{\theta}}(\boldsymbol{x})/p_\theta(\boldsymbol{x})$, $\nabla_{\tilde{\theta}} p_{\tilde{\theta}}(\boldsymbol{x})/p_\theta(\boldsymbol{x})$, $\log p_{\tilde{\theta}}(\boldsymbol{x})$, $\nabla_{\tilde{\theta}} \log p_{\tilde{\theta}}(\boldsymbol{x})$ are uniformly dominated by a function that is square integrable with respect to $p_\theta(\boldsymbol{x})$, for all $\tilde{\theta}$ belonging to some neighborhood of $\theta$. Also, $\boldsymbol{h}(\boldsymbol{x})$, $\log p_{\theta_{\text{base}}}(\boldsymbol{x})$ have finite second moment under $p_\theta(\boldsymbol{x})$.

These assumptions are sufficient to exchange integration and differentiation in Appendix B.2 with dominated convergence.

### B.1  LOSS ESTIMATES

We begin by proving that our estimates $\widehat{\mathcal{L}}^{\text{relax}}$ and $\widehat{\mathcal{L}}^{\text{reward}}$ for $\mathcal{L}^{\text{relax}}$ and $\mathcal{L}^{\text{reward}}$, respectively, are, on average, correct.

**Proposition B.2.** $\widehat{\mathcal{L}}^{relax}$ is unbiased for the relax loss $\mathcal{L}^{relax}$.

*Proof.* We prove unbiasedness of $\widehat{\mathcal{L}}^{\text{relax}}$ by showing that $\widehat{\mathcal{L}}^{\text{KL}}$ is unbiased for $\mathcal{L}^{\text{KL}} = \mathrm{D}_{\text{KL}}\left(p_\theta \parallel p_{\theta_{\text{base}}}\right)$ and that $\widehat{\mathcal{L}}^{\text{viol}}$ is unbiased for $\mathcal{L}^{\text{viol}} = \|\mathbb{E}_{p_\theta}[\boldsymbol{h}(\boldsymbol{x})] - \boldsymbol{h}^*\|^2$.

As for $\widehat{\mathcal{L}}^{\text{KL}}$, its expectation is

$$\mathbb{E}_{p_\theta}\left[\widehat{\mathcal{L}}^{\text{KL}}\right] = \frac{1}{M}\sum_{m=1}^{M} \mathbb{E}_{p_\theta}\left[\log \frac{p_\theta(\boldsymbol{x}_m)}{p_{\theta_{\text{base}}}(\boldsymbol{x}_m)}\right] = \frac{1}{M}\sum_{m=1}^{M} \mathrm{D}_{\text{KL}}\left(p_\theta \parallel p_{\theta_{\text{base}}}\right) = \mathrm{D}_{\text{KL}}\left(p_\theta \parallel p_{\theta_{\text{base}}}\right).$$

In the first equality we invoke the linearity of expectation and in the second we use our assumption that $\{\boldsymbol{x}_m\}_{m=1}^{M}$ are sampled from $p_\theta$.

And for $\widehat{\mathcal{L}}^{\text{viol}}$, we recall that for a real-valued random variable $Z$, $\mathbb{E}[Z^2] = \mathbb{E}[Z]^2 + \text{Var}(Z)$. Applying this to each dimension of $M^{-1}\sum_{m=1}^{M} \tilde{\boldsymbol{h}}_m = M^{-1}\sum_{m=1}^{M}(\boldsymbol{h}(\boldsymbol{x}_m) - \boldsymbol{h}^*)$, we obtain

$$\mathbb{E}_{p_\theta}\left\|\frac{1}{M}\sum_{m=1}^{M}\tilde{\boldsymbol{h}}_m\right\|^2 = \|\mathbb{E}_{p_\theta}[\tilde{\boldsymbol{h}}(\boldsymbol{x})]\|^2 + \frac{1}{M}\mathbb{E}_{p_\theta}\|\tilde{\boldsymbol{h}}(\boldsymbol{x}) - \mathbb{E}_{p_\theta}[\tilde{\boldsymbol{h}}(\boldsymbol{x})]\|^2, \tag{10}$$

where $\tilde{\boldsymbol{h}}(\boldsymbol{x}) = \boldsymbol{h}(\boldsymbol{x}) - \boldsymbol{h}^*$. Next, we replace the final term in (10) with $\mathbb{E}_{p_\theta}[M^{-1}(M - 1)^{-1} \sum_m \|\tilde{\boldsymbol{h}}_m - M^{-1} \sum_{m'} \tilde{\boldsymbol{h}}_{m'}\|^2]$. The quantity $M^{-1}(M-1)^{-1} \sum_m \|\tilde{\boldsymbol{h}}_m - M^{-1} \sum_{m'} \tilde{\boldsymbol{h}}_{m'}\|^2$ is simply the trace of the sample covariance matrix of $\{\tilde{\boldsymbol{h}}_m\}_{m=1}^M$, scaled by $M^{-1}$. The sample covariance of $\{\tilde{\boldsymbol{h}}_m\}_{m=1}^M$ is unbiased for $\mathrm{Cov}[\tilde{\boldsymbol{h}}]$. Rearranging the above expression yields

$$\|\mathbb{E}_{p_\theta}[\tilde{\boldsymbol{h}}(\boldsymbol{x})]\|^2 = \mathbb{E}_{p_\theta} \left\| \frac{1}{M} \sum_{m=1}^M \tilde{\boldsymbol{h}}_m \right\|^2 - \frac{1}{M} \mathbb{E}_{p_\theta} \left[ \frac{1}{(M-1)} \sum_{m=1}^M \left\| \tilde{\boldsymbol{h}}_m - \frac{1}{M} \sum_{m'=1}^M \tilde{\boldsymbol{h}}_{m'} \right\|^2 \right]$$

$$= \mathbb{E}_{p_\theta}[\widehat{\mathcal{L}}^{\mathrm{viol}}]$$

This proves $\widehat{\mathcal{L}}^{\mathrm{viol}}$ is unbiased for $\|\mathbb{E}_{p_\theta}[\boldsymbol{h}(\boldsymbol{x})] - \boldsymbol{h}^*\|^2$. $\qquad \square$

Likewise, we demonstrate that our estimate for the reward loss is unbiased.

**Proposition B.3.** $\widehat{\mathcal{L}}^{reward}$ *is unbiased for the reward loss* $\mathcal{L}^{reward}$.

*Proof.* In the proof of Proposition B.2 we already demonstrated $M^{-1} \sum_{m=1}^M \log \frac{p_\theta(\boldsymbol{x}_m)}{p_{\theta_{\mathrm{base}}}(\boldsymbol{x}_m)}$ is unbiased for $\mathcal{L}^{\mathrm{KL}}$. By an identical argument, $-M^{-1} \sum_{m=1}^M r_{\widehat{\boldsymbol{\alpha}}_N}(\boldsymbol{x}_m)$ is unbiased for $\mathcal{L}^{\mathrm{r}} = \mathbb{E}_{p_\theta}[-r_{\widehat{\boldsymbol{\alpha}}_N}(\boldsymbol{x})]$ (again, it is a Monte Carlo estimate). $\qquad \square$

## B.2 UNBIASED GRADIENT ESTIMATES

As we detailed in Section 2.3, the naïve idea of taking the unbiased loss estimators $\widehat{\mathcal{L}}^{\mathrm{relax}}$, $\widehat{\mathcal{L}}^{\mathrm{reward}}$ and differentiating them with respect to $\theta$ will not yield unbiased estimates for the gradients of $\mathcal{L}^{\mathrm{relax}}$ and $\mathcal{L}^{\mathrm{reward}}$. This is because the probability distribution with respect to which the expectation is taken also depends on $\theta$, which needs to be taken into account in the gradient estimate.

For CGM-reward, we propose the gradient estimate

$$\widehat{G}^{\mathrm{reward}} = \frac{1}{M} \sum_{m=1}^M (\nabla_\theta w_m(\theta, \theta')) \left( l_m^{\mathrm{LOO}} - r_m^{\mathrm{LOO}} \right), \quad w_m(\theta, \theta') = \frac{p_\theta(\boldsymbol{x}_m)}{p_{\theta'}(\boldsymbol{x}_m)}$$

$$l_m^{\mathrm{LOO}} = l_m - \frac{1}{M-1} \sum_{m' \neq m} l_{m'}, \quad l_m = \log \frac{p_\theta(\boldsymbol{x}_m)}{p_{\theta_{\mathrm{base}}}(\boldsymbol{x}_m)} \tag{11}$$

$$r_m^{\mathrm{LOO}} = r_m - \frac{1}{M-1} \sum_{m' \neq m} r_{m'}, \quad r_m = r_{\widehat{\boldsymbol{\alpha}}_N}(\boldsymbol{x}_m)$$

As we explained in Section 2.3, $w_m(\theta, \theta')$ can be viewed as the weights of an *importance sampling* scheme, where $p_{\theta'}$ is the proposal distribution and $p_\theta$ is the target distribution. We choose $\theta' = \theta$ so that the proposal distribution is equal to the target distribution. For this choice of proposal, the weights $w_m$ are all equal to 1. However, their gradient with respect to $\theta$ is equal to the score of the calibrated model at $\boldsymbol{x}_m$, $\nabla_\theta \log p_\theta(\boldsymbol{x}_m)$. The expression (11), excluding the terms $(M - 1)^{-1} \sum_{m' \neq m} l_{m'}$ and $(M-1)^{-1} \sum_{m' \neq m} r_{m'}$ is known as the score function gradient estimate or, in the terminology of reinforcement learning, the REINFORCE gradient estimate (Williams, 1992).

The terms $(M-1)^{-1} \sum_{m' \neq m} l_{m'}$ and $(M-1)^{-1} \sum_{m' \neq m} r_{m'}$ in (11) are known as leave-one-out baselines (Kool et al., 2019) corresponding to sample $\boldsymbol{x}_m$. Including these terms adds to the score function gradient estimate a *control variate*, which is a term that has expectation zero under $p_\theta$ but is correlated with each individual term in the estimate (Lavenberg & Welch, 1981; Ranganath et al., 2014; Mohamed et al., 2020). Indeed, we observe that by independence of the samples $\{\boldsymbol{x}_m\}_{m=1}^M$, it holds that for each $m \neq m'$,

$$\mathbb{E}_{p_\theta} [(\nabla_\theta \log p_\theta(\boldsymbol{x}_m))(l_{m'} - r_{m'})] = \mathbb{E}_{p_\theta} [\nabla_\theta \log p_\theta(\boldsymbol{x}_m)] \mathbb{E}_{p_\theta} [l_{m'} - r_{m'}] = 0.$$

Consequently, while the inclusion of the leave-one-out averages does not affect the unbiasedness of our gradient estimate, they can reduce its variance.

**Proposition B.4.** $\widehat{G}^{reward}$ *is unbiased for the gradient of the reward loss,* $\nabla_\theta \mathcal{L}^{reward}$.

*Proof.* We start by writing out the gradient of $\mathcal{L}^{\text{reward}}$ directly:

$$\nabla_\theta \mathcal{L}^{\text{reward}}(\theta) = \nabla_\theta \mathbb{E}_{p_\theta} \left[ \log \frac{p_\theta(\boldsymbol{x})}{p_{\theta_{\text{base}}}(\boldsymbol{x})} - r_{\widehat{\boldsymbol{\alpha}}_N}(\boldsymbol{x}) \right]$$

$$= \nabla_\theta \int \left\{ \log \frac{p_\theta(\boldsymbol{x})}{p_{\theta_{\text{base}}}(\boldsymbol{x})} - r_{\widehat{\boldsymbol{\alpha}}_N}(\boldsymbol{x}) \right\} p_\theta(d\boldsymbol{x})$$

$$= \nabla_\theta \int \frac{p_\theta(\boldsymbol{x})}{p_{\theta'}(\boldsymbol{x})} \left\{ \log \frac{p_\theta(\boldsymbol{x})}{p_{\theta_{\text{base}}}(\boldsymbol{x})} - r_{\widehat{\boldsymbol{\alpha}}_N}(\boldsymbol{x}) \right\} p_{\theta'}(d\boldsymbol{x})$$

$$\overset{(\star)}{=} \mathbb{E}_{p_\theta} \left[ \left( \nabla_\theta \frac{p_\theta(\boldsymbol{x})}{p_{\theta'}(\boldsymbol{x})} \right) \left\{ \log \frac{p_\theta(\boldsymbol{x})}{p_{\theta_{\text{base}}}(\boldsymbol{x})} - r_{\widehat{\boldsymbol{\alpha}}_N}(\boldsymbol{x}) \right\} \right] + \mathbb{E}_{p_\theta} \left[ \nabla_\theta \frac{p_\theta(\boldsymbol{x})}{p_{\theta'}(\boldsymbol{x})} \right] \quad (12)$$

where $\theta' = \texttt{stop-grad}(\theta)$. In equality $(\star)$, exchange of the gradient and expectation is permissible as a consequence of dominated convergence and Assumption B.1. The second term is the expected score, which is zero. And so the gradient of the reward loss is

$$\nabla_\theta \mathcal{L}^{\text{reward}}(\theta) = \mathbb{E}_{p_\theta} \left[ (\nabla_\theta \log p_\theta(\boldsymbol{x})) \left\{ \log \frac{p_\theta(\boldsymbol{x})}{p_{\theta_{\text{base}}}(\boldsymbol{x})} - r_{\widehat{\boldsymbol{\alpha}}_N}(\boldsymbol{x}) \right\} \right]. \quad (13)$$

Looking at our gradient estimator $\widehat{G}^{\text{reward}}$ in (11) and ignoring the leave-one-out averages, we see that it is exactly the Monte Carlo estimate of the gradient of $\mathcal{L}^{\text{reward}}$ (13). $\qquad\square$

Dropping the potentially noisy expected score term in (12), as is done by Ranganath et al. (2014), also reduces variance of our gradient estimate.

Deriving an unbiased gradient estimate for the relax loss is more challenging, since the loss cannot be written as the expectation of some objective under $p_\theta$. Just as we did for the reward loss, we can compute an unbiased estimate for the gradient of $\mathcal{L}^{\text{KL}}$ in the relax loss by drawing independent samples $\boldsymbol{x}_m \sim p_{\theta'}$ and then differentiating the importance sampling weights $w_m(\theta, \theta')$

$$\widehat{G}_{\text{KL}} = \frac{1}{M} \sum_{m=1}^M (\nabla_\theta w_m(\theta, \theta')) l_m^{\text{LOO}}.$$

And so it remains to compute an unbiased gradient estimate for $\mathcal{L}^{\text{viol}}$. To do so, we first recall the unbiased estimate $\widehat{\mathcal{L}}^{\text{viol}}$ for $\mathcal{L}^{\text{viol}}$ that we introduced in Section 2.1

$$\widehat{\mathcal{L}}^{\text{viol}}(\{\tilde{\boldsymbol{h}}_m\}_{m=1}^M) := \left\| \frac{1}{M} \sum_{m=1}^M \tilde{\boldsymbol{h}}_m \right\|^2 - \frac{1}{M(M-1)} \sum_{m=1}^M \left\| \tilde{\boldsymbol{h}}_m - \frac{1}{M} \sum_{m'=1}^M \tilde{\boldsymbol{h}}_{m'} \right\|^2,$$

where $\boldsymbol{x}_m$ are independent samples from $p_\theta$ and $\tilde{\boldsymbol{h}}_m = \boldsymbol{h}(\boldsymbol{x}_m) - \boldsymbol{h}^*$. We propose a modification to this estimate wherein we draw independent samples $\boldsymbol{x}_m \sim p_{\theta'}$ and replace $\{\tilde{\boldsymbol{h}}_m\}_{m=1}^M$ by $\{w_m(\theta, \theta') \tilde{\boldsymbol{h}}_m\}_{m=1}^M$. To estimate the gradient of $\|\mathbb{E}_{p_\theta}[\boldsymbol{h}] - \boldsymbol{h}^*\|^2 = \|\mathbb{E}_{p_\theta}[\tilde{\boldsymbol{h}}]\|^2$, we compute the gradient of $\widehat{\mathcal{L}}^{\text{viol}}(\{w_m(\theta, \theta') \tilde{\boldsymbol{h}}_m\}_{m=1}^M)$ with respect to $\theta$ and then evaluate the result at $\theta' = \theta$. In Algorithms 1 and 2, we implement our gradient estimate for $\mathcal{L}^{\text{viol}}$ by sampling $\boldsymbol{x}_m$ independently from $p_{\texttt{stop-grad}(\theta)}$ and differentiating $\widehat{\mathcal{L}}^{\text{viol}}(\{w_m(\theta, \theta') \tilde{\boldsymbol{h}}_m\}_{m=1}^M)$ with $\theta' = \texttt{stop-grad}(\theta)$.

This yields the overall gradient estimator for the relax loss

$$\widehat{G}^{\text{relax}} = \nabla_\theta \widehat{\mathcal{L}}^{\text{viol}} \left( \left\{ w(\theta, \theta') \tilde{\boldsymbol{h}}_m \right\}_{m=1}^M \right) + \lambda \widehat{G}_{\text{KL}}, \quad \boldsymbol{x}_m \overset{i.i.d.}{\sim} p_{\theta'}, \quad \theta' = \texttt{stop-grad}(\theta).$$

In order to prove that $\widehat{G}^{\text{relax}}$ is unbiased for $\nabla_\theta \mathcal{L}^{\text{relax}}$, we need to show $\widehat{\mathcal{L}}^{\text{viol}}(\{w_m(\theta, \theta') \tilde{\boldsymbol{h}}_m\}_{m=1}^M)$ remains unbiased for $\mathcal{L}^{\text{viol}}$ when $\boldsymbol{x}_m$ are sampled independently from $p_{\theta'}$. Then, since the distribution from which $\boldsymbol{x}_m$ are sampled does not depend on $\theta$, it is allowable to exchange the gradient with the expectation.

**Proposition B.5.** *$\widehat{G}^{\text{relax}}$ is unbiased for the gradient of the relax loss, $\nabla_\theta \mathcal{L}^{\text{relax}}$.*

*Proof.* From Proposition B.2, we know that $\widehat{G}_{\mathrm{KL}}$ is unbiased for $\nabla_\theta \mathcal{L}^{\mathrm{KL}}$, and so it only remains to verify that the second term is unbiased for $\nabla_\theta \mathcal{L}^{\mathrm{viol}} = \nabla_\theta \|\mathbb{E}_{p_\theta}[\boldsymbol{h}] - \boldsymbol{h}^*\|^2$. To this end, by repeating the proof of Proposition B.2 (i.e., using the definition of the variance), it is straightforward to show

$$\mathbb{E}_{p_{\theta'}}\left[\widehat{\mathcal{L}}^{\mathrm{viol}}\left(\left\{\frac{p_\theta(\boldsymbol{x}_m)}{p_{\theta'}(\boldsymbol{x}_m)}\tilde{\boldsymbol{h}}_m\right\}_{m=1}^M\right)\right] = \left\|\mathbb{E}_{p_{\theta'}}\left[\frac{p_\theta(\boldsymbol{x}_m)}{p_{\theta'}(\boldsymbol{x}_m)}\tilde{\boldsymbol{h}}_m\right]\right\|^2 = \|\mathbb{E}_{p_\theta}[\tilde{\boldsymbol{h}}]\|^2.$$

In other words, $\widehat{\mathcal{L}}^{\mathrm{viol}}(\{w_m(\theta,\theta')\tilde{\boldsymbol{h}}_m\}_{m=1}^M)$ is unbiased for $\mathcal{L}^{\mathrm{viol}}$. However, since the samples $\{\boldsymbol{x}_m\}_{m=1}^M$ are drawn from $p_{\theta'}$, a probability distribution that does not depend on $\theta$, then we can exchange the gradient and expectation by appealing to dominated convergence and Assumption B.1. In particular, we have

$$\mathbb{E}_{p_{\theta'}}\left[\nabla_\theta \widehat{\mathcal{L}}^{\mathrm{viol}}\left(\left\{\frac{p_\theta(\boldsymbol{x}_m)}{p_{\theta'}(\boldsymbol{x}_m)}\tilde{\boldsymbol{h}}_m\right\}_{m=1}^M\right)\right] = \nabla_\theta \mathbb{E}_{p_{\theta'}}\left[\widehat{\mathcal{L}}^{\mathrm{viol}}\left(\left\{\frac{p_\theta(\boldsymbol{x}_m)}{p_{\theta'}(\boldsymbol{x}_m)}\tilde{\boldsymbol{h}}_m\right\}_{m=1}^M\right)\right]$$
$$= \nabla_\theta \mathcal{L}^{\mathrm{viol}},$$

where the final line follows from the unbiasedness of $\widehat{\mathcal{L}}^{\mathrm{viol}}(\{w_m\tilde{\boldsymbol{h}}_m\}_{m=1}^M)$ for $\mathcal{L}^{\mathrm{viol}}$. $\qquad\square$

As we discussed, the key insight from the proof of Proposition B.5 is that, by introducing importance weights, we can compute an unbiased estimate to $\|\mathbb{E}_{p_\theta}[\boldsymbol{h}] - \boldsymbol{h}^*\|^2 = \|\mathbb{E}_{p_\theta}[\tilde{\boldsymbol{h}}]\|$ without sampling directly from $p_\theta$.

## C   MAXIMUM ENTROPY PRINCIPLE

In this section, we provide an overview of the maximum entropy principle, which we use in Section 2.2 to define the reward loss $\mathcal{L}^{\mathrm{reward}}$. First, in Appendix C.1 we formally state and prove the maximum entropy principle. In Appendix C.2, we provide greater detail on our estimate $\widehat{\boldsymbol{\alpha}}_N$ for the parameters $\boldsymbol{\alpha}^*$ of the maximum entropy solution. In Appendix C.3, we characterize the relationship between the relax and reward losses by considering a problem whose solution is close to the optimum of the relax loss, and which resembles the maximum entropy problem. Lastly, in Appendix C.4, we study the behavior of the estimate $\widehat{\boldsymbol{\alpha}}_N$ in the limit as the number of samples $N$ becomes large.

Prior to jumping into the details of the maximum entropy principle, we work through an illustrative example that we discuss throughout this section.

**Example.** Suppose $\boldsymbol{x} \in \mathbb{R}$, $\boldsymbol{h}(\boldsymbol{x}) = \mathbb{1}\{\boldsymbol{x} > 0\}$, and $\boldsymbol{h}^* \in \mathbb{R}$. Also define $h_b = \mathbb{P}_{p_{\theta_{\mathrm{base}}}}(\boldsymbol{x} > 0)$, and assume $0 < h_b < 1$. In this example, the calibration problem amounts to either upweighting or downweighting the amount of probability mass $h_b$ that lies above 0 under the base model $p_{\theta_{\mathrm{base}}}$. By Theorem 2.1, the maximum entropy solution has the form $p_{\boldsymbol{\alpha}^*} \propto p_{\theta_{\mathrm{base}}}(\boldsymbol{x})\exp\{\boldsymbol{\alpha}^*\boldsymbol{h}(\boldsymbol{x})\}$ for some $\boldsymbol{\alpha}^* \in \mathbb{R}$ that we need to determine. From this expression for $p_{\boldsymbol{\alpha}^*}$, we obtain

$$1 - \boldsymbol{h}^* = \mathbb{E}_{p_{\boldsymbol{\alpha}^*}}[1 - \boldsymbol{h}(\boldsymbol{x})] = \frac{1}{h_b\exp(\boldsymbol{\alpha}^*) + (1 - h_b)}(1 - h_b),$$

$$\boldsymbol{h}^* = \mathbb{E}_{p_{\boldsymbol{\alpha}^*}}[\boldsymbol{h}(\boldsymbol{x})] = \frac{1}{h_b\exp(\boldsymbol{\alpha}^*) + (1 - h_b)}h_b\exp(\boldsymbol{\alpha}^*).$$

Dividing the first equation by the second and rearranging yields $\boldsymbol{\alpha}^* = \log(\frac{\boldsymbol{h}^*(1-h_b)}{(1-\boldsymbol{h}^*)h_b})$. Following the same argument for the empirical distribution of $\{\boldsymbol{x}_n\}_{n=1}^N$, our estimator for $\boldsymbol{\alpha}^*$ is $\widehat{\boldsymbol{\alpha}}_N = \log(\frac{\boldsymbol{h}^*(1-\bar{\boldsymbol{y}}_N)}{(1-\boldsymbol{h}^*)\bar{\boldsymbol{y}}_N})$, where $\bar{\boldsymbol{y}}_N = \frac{1}{N}\sum_{n=1}^N \boldsymbol{y}_n$, $\boldsymbol{y}_n = \mathbb{1}\{\boldsymbol{x}_n > 0\} \stackrel{d}{=} \mathrm{Bernoulli}(h_b)$ for $\boldsymbol{x}_n \stackrel{i.i.d.}{\sim} p_{\theta_{\mathrm{base}}}$.

We point out that $\boldsymbol{\alpha}^*$ and $\widehat{\boldsymbol{\alpha}}_N$ can equivalently be derived by differentiating the objectives (6) and (7), respectively, and setting them equal to 0.

### C.1 PRECISE STATEMENT

Since the maximum entropy problem is not specific to generative model calibration, we present it in a more general setting. Our presentation builds on standard results from exponential families and convex analysis. We recommend Wainwright & Jordan (2008) for relevant background.

In particular, we consider $X := (X, \mathcal{X})$ a measurable space, $P$ a probability measure defined on $X$, $\boldsymbol{h} : X \to \mathbb{R}^d$ an $X$-measurable constraint function, and $\boldsymbol{h}^*$ a target value for the moment of $\boldsymbol{h}$. The maximum entropy problem corresponding to probability measure $P$, constraint $\boldsymbol{h}$, and target moment $\boldsymbol{h}^*$ is

$$\inf_{Q \in \mathcal{P}(P)} \mathrm{D_{KL}}\left(Q \parallel P\right), \quad \text{such that } \mathbb{E}_Q[\boldsymbol{h}(\boldsymbol{x})] = \boldsymbol{h}^*. \tag{14}$$

$\mathcal{P}(P)$ is the collection of all probability measures having a density with respect to $P$, which, by the Radon-Nikodym Theorem, is equal to the collection of all absolutely continuous probability measures with respect to $P$. Choosing $P = p_{\theta_{\mathrm{base}}}$ yields the maximum entropy problem corresponding to the calibration problem.

As we mentioned in Section 2.2, we impose a condition on the target moment $\boldsymbol{h}^*$ to ensure (i) there exists a solution to the maximum entropy problem (ii) and this solution is an exponential tilt of $P$.

**Assumption C.1** (Interior moment condition). Define the subset $\mathcal{M}$ of $\mathbb{R}^d$ comprised of all possible moments of $\boldsymbol{h}$ attainable by probability distributions $Q$ having a density with respect to $P$

$$\mathcal{M} = \left\{ \int \boldsymbol{h}(\boldsymbol{x}) Q(d\boldsymbol{x}) \;\middle|\; Q \in \mathcal{P}(P), \int \|\boldsymbol{h}(\boldsymbol{x})\| Q(d\boldsymbol{x}) < \infty \right\}.$$

$\boldsymbol{h}^*$ lies in the *relative interior* of $\mathcal{M}$, written relint($\mathcal{M}$).

Since $\mathcal{M}$ is a convex set, the condition $\boldsymbol{h}^* \in \mathrm{relint}(\mathcal{M})$ can equivalently be stated as for every $\boldsymbol{y} \neq \boldsymbol{h}^*$ in $\mathcal{M}$, there exists some $\boldsymbol{z}$ in $\mathcal{M}$ and $\kappa \in (0, 1)$ for which $\boldsymbol{h}^* = \kappa \boldsymbol{z} + (1 - \kappa)\boldsymbol{y}$.

To see why Assumption C.1 is necessary for the solution to be an exponential tilt of $p_{\theta_{\mathrm{base}}}$, recall the example discussed at the beginning of Appendix C. In this case, relint($\mathcal{M}$) $= (0, 1)$. If $\boldsymbol{h}^* \notin [0, 1]$, then there does not exist any probability distribution $p$ having density with respect to $p_{\theta_{\mathrm{base}}}$ for which $\mathbb{E}_p[\boldsymbol{h}(\boldsymbol{x})] = \boldsymbol{h}^*$. And if $\boldsymbol{h}^*$ is either 0 or 1, then the solution to the maximum entropy problem is proportional to $p_{\theta_{\mathrm{base}}}(\boldsymbol{x})\mathbb{1}\{\boldsymbol{x} \leq 0\}$ or $p_{\theta_{\mathrm{base}}}(\boldsymbol{x})\mathbb{1}\{\boldsymbol{x} > 0\}$, respectively. Neither of these solutions is an exponential tilt of $p_{\theta_{\mathrm{base}}}$, equation (5).

Our proof of the maximum entropy principle leverages classical convex duality (Rockafellar, 1970) by showing that (14) is a convex problem, defined on the infinite-dimensional space of all probability densities for which $\boldsymbol{h}$ has a finite moment. The corresponding *dual problem* is

$$\sup_{\boldsymbol{\alpha} \in \mathbb{R}^d} \boldsymbol{\alpha}^\top \boldsymbol{h}^* - A_P(\boldsymbol{\alpha}), \quad A_P(\boldsymbol{\alpha}) := \log\left( \int \exp\{r_{\boldsymbol{\alpha}}(\boldsymbol{x})\} P(d\boldsymbol{x}) \right), \quad r_{\boldsymbol{\alpha}}(\boldsymbol{x}) = \boldsymbol{\alpha}^\top \boldsymbol{h}(\boldsymbol{x}), \tag{15}$$

which is concave. $A_P : \mathbb{R}^d \to \mathbb{R} \cup \{+\infty\}$ is known as the *log-normalizer* or *cumulant generating function* corresponding to the exponential family

$$\exp\{r_{\boldsymbol{\alpha}}(\boldsymbol{x}) - A_P(\boldsymbol{x})\} P(d\boldsymbol{x}). \tag{16}$$

We will make the standard assumption that the domain of $A_P$ is open

**Assumption C.2** (Domain of log-normalizer). The subset $\Xi = \{\boldsymbol{\alpha} \in \mathbb{R}^d \mid A_P(\boldsymbol{\alpha}) < \infty\}$ is open.

Whenever $A_P$ is finite, (16) is a well-defined probability measure on $X$. $\Xi$ is known as the *natural parameter space* of the exponential family (16). When Assumption C.2 holds, the exponential family is said to be *regular*.

The log-normalizer $A_P$ possesses many nice properties: for instance, it is convex and infinitely differentiable on $\Xi$. Convexity can be seen by computing the Hessian of $A_P(\boldsymbol{\alpha})$

$$\nabla^2_{\boldsymbol{\alpha}} A_P(\boldsymbol{\alpha}) = \frac{\int (\boldsymbol{h}(\boldsymbol{x}) - \nabla_{\boldsymbol{\alpha}} A_P(\boldsymbol{\alpha}))(\boldsymbol{h}(\boldsymbol{x}) - \nabla_{\boldsymbol{\alpha}} A_P(\boldsymbol{\alpha}))^\top \exp\{r_{\boldsymbol{\alpha}}(\boldsymbol{x})\} P(d\boldsymbol{x})}{\int \exp\{r_{\boldsymbol{\alpha}}(\boldsymbol{x})\} P(d\boldsymbol{x})} \tag{17}$$

and recognizing that it is positive semi-definite. Differentiability is addressed in the remark following Lemma C.11.

Now that we have introduced the dual of the maximum entropy problem, we are prepared to give a precise statement and proof of the maximum entropy principle

**Theorem C.3** (Kullback (1959)). *Suppose Assumptions C.1 and C.2 hold. Then there exists a probability measure $Q^* \in \mathcal{P}(P)$ with density $dQ^*/dP \propto \exp\left(r_{\boldsymbol{\alpha}^*}(\boldsymbol{x})\right)$. Moreover, $Q^*$ is the solution to the maximum entropy problem (14) and is unique up to $P$-null sets.*

Unlike the primal problem (14), the dual problem (15) is defined on finite-dimensional Euclidean space, which makes it simpler to analyze. We first argue by weak duality that the value of (14) is at least as large as (15). We then identify a vector $\boldsymbol{\alpha}^*$ and a distribution $Q^*$ for which the primal and dual objectives are equal. By weak duality, this implies that $Q^*$ is optimal for the primal problem.

*Proof of Theorem C.3.* We first rewrite the primal problem (14) in the form

$$\inf_q \psi(q) + g(\mathcal{A}q)$$

$$\psi(q) = \begin{cases} \int q(\boldsymbol{x}) \log(q(\boldsymbol{x})) P(d\boldsymbol{x}) & \text{if } q \geq 0 \\ +\infty & \text{else} \end{cases}, \quad g(\boldsymbol{y}_0, \boldsymbol{y}_1) = \begin{cases} 0 & \text{if } \boldsymbol{y}_0 = 1 \text{ and } \boldsymbol{y}_1 = \boldsymbol{h}^* \\ +\infty & \text{else} \end{cases},$$

$$\mathcal{A}(q) = \left( \int q(\boldsymbol{x}) P(d\boldsymbol{x}), \int \boldsymbol{h}(\boldsymbol{x}) q(\boldsymbol{x}) P(d\boldsymbol{x}) \right)$$

defined on the space of $X$-measurable functions $q$ for which $\int |q(\boldsymbol{x})| P(d\boldsymbol{x}) < \infty$ and $\int \|\boldsymbol{h}(\boldsymbol{x})\| q(\boldsymbol{x}) P(d\boldsymbol{x}) < \infty$. Here, $q$ represents the density of measure $Q$ with respect to $P$. $g(\mathcal{A}q)$ imposes the constraint that $Q$ is a probability measure and that the expectation of $\boldsymbol{h}$ under $Q$ is $\boldsymbol{h}^*$. And $\psi(q)$ is equal to the KL divergence between $Q$ and $P$.

Observe $\mathcal{A}$ is a bounded, linear map defined on this space. And $\psi$ and $g$ are convex. By Fenchel-Rockafellar duality (Borwein & Zhu, 2005, Theorem 4.4.2), weak duality holds for the maximum entropy problem and its dual (15).

Wainwright & Jordan (2008, Theorem 3.3) states that $\nabla_{\boldsymbol{\alpha}} A_P$ is a surjective mapping from $\Xi$ onto $\text{relint}(\mathcal{M})$. Hence, there exists $\boldsymbol{\alpha}^* \in \Xi$ for which $\nabla_{\boldsymbol{\alpha}} A_P(\boldsymbol{\alpha}^*) = \boldsymbol{h}^*$. The value of the dual at $\boldsymbol{\alpha}^*$ is

$$(\boldsymbol{\alpha}^*)^\top \boldsymbol{h}^* - A_P(\boldsymbol{\alpha}^*).$$

By differentiating the dual objective at $\boldsymbol{\alpha}^*$, we obtain,

$$0 = \nabla_{\boldsymbol{\alpha}} (\boldsymbol{\alpha}^\top \boldsymbol{h}^* - A_P(\boldsymbol{\alpha})) \implies \boldsymbol{h}^* = \frac{\int \boldsymbol{h}(\boldsymbol{x}) \exp\{r_{\boldsymbol{\alpha}^*}(\boldsymbol{x})\} P(d\boldsymbol{x})}{\int \exp\{r_{\boldsymbol{\alpha}^*}(\boldsymbol{x})\} P(d\boldsymbol{x})}.$$

In other words, the distribution $Q^* \in \mathcal{P}(P)$ defined such that $dQ^*/dP \propto \exp\{r_{\boldsymbol{\alpha}^*}(\boldsymbol{x})\}$ satisfies the moment constraint $\mathbb{E}_{Q^*}[\boldsymbol{h}(\boldsymbol{x})] = \boldsymbol{h}^*$. Moreover, the value of the primal objective at $Q^*$ is

$$D_{\text{KL}}(Q^* \| P) = (\boldsymbol{\alpha}^*)^\top \boldsymbol{h}^* - A_P(\boldsymbol{\alpha}^*),$$

which is equal to the value of the dual objective at $\boldsymbol{\alpha}^*$. By weak duality, we conclude $Q^*$ is the solution to the maximum entropy problem.

Uniqueness follows from the fact that the KL divergence $\psi$ is strictly convex. $\qquad \square$

## C.2 ESTIMATING THE MAXIMUM ENTROPY SOLUTION

Next, we discuss our estimator $\widehat{\boldsymbol{\alpha}}_N$ for the parameters $\boldsymbol{\alpha}^*$ of the maximum entropy solution. In particular, we provide verifiable conditions under which $\widehat{\boldsymbol{\alpha}}_N$ is well-defined, and we show that this estimator can be interpreted as the solution to a finite-sample version of the maximum entropy problem (4).

So far, the only assumptions we have made on the maximum entropy problem (14) are the relative interior condition on $\boldsymbol{h}^*$ (Assumption C.1) and the openness condition for the domain of $A_P$ (Assumption C.2). As we demonstrated in Appendix C.1, these conditions ensure that the solution to

the maximum entropy problem exists and is unique. However, the solution to the dual problem need not be unique. Suppose, for example, that $\boldsymbol{h}$ is $d$-dimensional but has two identical components $\boldsymbol{h}(\boldsymbol{x})[i] = \boldsymbol{h}(\boldsymbol{x})[j]$. Then if $\boldsymbol{\alpha}^*$ is optimal for the dual problem, so is $\boldsymbol{\alpha}^* - t\boldsymbol{e}[i] + t\boldsymbol{e}[j]$ for all $t \in \mathbb{R}$, where $\boldsymbol{e}[i]$ and $\boldsymbol{e}[j]$ denote the $i$ and $j$th standard basis vectors, respectively. Specifically, the set of optima for the dual problem is a hyperplane in $\mathbb{R}^d$. In order to estimate $\boldsymbol{\alpha}^*$, we want to ensure that the dual problem (15) also has a unique maximum.

As suggested by our example, in order to ensure that the dual optimum is unique, it suffices to eliminate linear redundancies among the statistics $\boldsymbol{h}(\boldsymbol{x})$.

**Assumption C.4** (Uniqueness of dual optimum). No linear combination of the components of $\boldsymbol{h}(\boldsymbol{x})$ is equal to a constant with $P$ probability one.

If Assumption C.4 holds, then the exponential family (16) is said to be *minimal*. An exponential family for which Assumption C.2 holds is minimal if and only if the log-normalizer $A_P(\boldsymbol{\alpha})$ is strictly convex on $\Xi$ (Wainwright & Jordan, 2008, Proposition 3.1).

For non-trivial generative models, solving the dual problem (15) for $P = p_{\theta_{\text{base}}}$ is intractable since $A_{p_{\theta_{\text{base}}}}(\boldsymbol{\alpha})$ cannot be computed in closed-form. The estimator $\widehat{\boldsymbol{\alpha}}_N$ that we propose in (7) involves first drawing $N$ independent samples $\{\boldsymbol{x}_n\}_{n=1}^N$ from the base model $p_{\theta_{\text{base}}}$ and then solving the dual problem with the integral replaced by the empirical average from our samples. This is equivalent to solving the dual problem for $P$ equal to the empirical distribution of our samples $\frac{1}{N}\sum_{n=1}^N \delta_{\boldsymbol{x}_n}$, where $\delta_{\boldsymbol{x}}$ is the delta function at $\boldsymbol{x}$.

However, in order for $\widehat{\boldsymbol{\alpha}}_N$ to be well-defined, the interior point condition and uniqueness of the dual optimum must hold for the maximum entropy problem with $P = \frac{1}{N}\sum_{n=1}^N \delta_{\boldsymbol{x}_n}$. For this problem, these two conditions are straightforward to verify: (i) $\boldsymbol{h}^*$ lies in the the relative interior of the convex hull of $\{\boldsymbol{h}(\boldsymbol{x}_n)\}_{n=1}^N$ and (ii) the empirical covariance matrix of $\{\boldsymbol{h}(\boldsymbol{x}_n)\}_{n=1}^N$ has full rank. For the example we provided at the beginning of the section, conditions (i) and (ii) are satisfied if and only if $\{\boldsymbol{h}(\boldsymbol{x}_n)\} = \{0, 1\}$ and $\boldsymbol{h}^* \in (0, 1)$.

It is possible for Assumptions C.1 and C.4 to hold for $p_{\theta_{\text{base}}}$ but not for $\frac{1}{N}\sum_{n=1}^N \delta_{\boldsymbol{x}_n}$. For our example, if $\{\boldsymbol{h}(\boldsymbol{x}_n)\} = \{0\}$ and $\boldsymbol{h}^* = 0$ (or $\{\boldsymbol{h}(\boldsymbol{x}_n)\} = \{1\}$ and $\boldsymbol{h}^* = 1$), then the maximum entropy solution exists and is equal to $Q^* = \frac{1}{N}\sum_{n=1}^N \delta_{\boldsymbol{x}_n}$, but every vector $\boldsymbol{\alpha} \in \mathbb{R}$ is optimal for the dual problem (15). We demonstrate in Appendix C.4 that the probability of this event approaches zero as the number of samples $N$ approaches infinity. However, we observe (e.g., Figure 2B) that when the base model $p_{\theta_{\text{base}}}$ lies far from the maximum entropy solution $p_{\boldsymbol{\alpha}^*}$, estimating $\boldsymbol{\alpha}^*$ with small variance requires many samples, and may even be computationally intractable.

## C.3 Connection Between the Relax and Reward Losses

In this section, we elucidate the connection between the relax and reward losses. We first introduce a problem corresponding to the relax loss that, similar to the maximum entropy problem (4), is defined on the space $\mathcal{P}(p_{\theta_{\text{base}}})$ of probability distributions that have a density with respect to $p_{\theta_{\text{base}}}$. When the generative model class $p_\theta$ is sufficiently expressive, the solution to this problem well approximates the minimizer of the relax loss. We then show that, under conditions, the solution to this related problem approaches the solution to the maximum entropy problem as $\lambda \to 0$. This confirms our intuition that when $\lambda \to 0$, minimizing the relax loss is equivalent to solving the calibration problem.

As in Appendix C.1, we let $X := (X, \mathcal{X})$ be a measurable space, $P$ be a probability measure defined on $X$, and $\boldsymbol{h} : X \to \mathbb{R}^d$ be a $X$-measurable function, and $\boldsymbol{h}^*$ be a target moment. We consider the problem

$$\inf_{Q \in \mathcal{P}(P)} \|\mathbb{E}_Q[\boldsymbol{h}] - \boldsymbol{h}^*\|^2 + \lambda \mathrm{D}_{\text{KL}}\left(Q \parallel P\right), \quad \text{s.t. } \mathbb{E}_Q[\|\boldsymbol{h}\|] < \infty \tag{18}$$

In convex analysis (e.g., Boyd & Vandenberghe, 2004; Ben-Tal & Nemirovski, 2023), (18) is known as a penalty problem.

When $P = p_{\theta_{\text{base}}}$, then (18) agrees with the problem of minimizing the relax loss (2), except the domain of the problem is $\mathcal{P}(p_{\theta_{\text{base}}})$ rather than the class of generative models $p_\theta$. Suppose momentarily that the infimum of (18), denoted by $Q_\lambda$, is attained. The minimizer of the relax loss (2) will not in general be equal to $Q_\lambda$ since $Q_\lambda$ does not lie in the class of generative models. However, as we

argued when we proposed the reward loss, we would expect $Q_\lambda$ and the minimizer of the relax loss to be close in KL distance when the class of generative models $p_\theta$ is sufficiently expressive.

Introducing the problem (18) is helpful insofar as, similar to the maximum entropy problem, we can obtain a closed-form expression for the solution $Q_\lambda$.

**Proposition C.5.** *Suppose Assumption C.2 holds. Then there exists a unique solution $\boldsymbol{\alpha}_\lambda$ to the fixed point equation*

$$\boldsymbol{\alpha} = -\frac{2}{\lambda}(\nabla_{\boldsymbol{\alpha}} A_P(\boldsymbol{\alpha}) - \boldsymbol{h}^*), \quad \boldsymbol{\alpha} \in \Xi.$$

*Moreover, $Q_\lambda$ defined by $dQ_\lambda/dP \propto \exp\{\boldsymbol{\alpha}_\lambda^\top \boldsymbol{h}(\boldsymbol{x})\}$ is the unique solution to (18).*

Our proof mirrors that for the maximum entropy principle (Theorem C.3). Namely, we invoke Fenchel-Rockafellar duality (Rockafellar, 1970) to relate the convex problem (18), defined on the space of probability densities with respect to $P$ with finite $\boldsymbol{h}$ moment, to its concave dual problem

$$\sup_{\boldsymbol{\alpha} \in \mathbb{R}^p} F_\lambda(\boldsymbol{\alpha}), \quad F_\lambda(\boldsymbol{\alpha}) = \lambda\left(-\frac{\lambda}{4}\|\boldsymbol{\alpha}\|^2 - A_P(\boldsymbol{\alpha}) + \boldsymbol{\alpha}^\top \boldsymbol{h}^*\right) \tag{19}$$

defined on Euclidean space. We then show that $\boldsymbol{\alpha}_\lambda$ is the unique solution to the dual problem, and we use this solution to construct a solution to the primal problem. Interestingly, $\boldsymbol{\alpha}_\lambda$ is the unique solution to the dual problem even when there is redundancy among the constraints $\boldsymbol{h}$ (i.e., Assumption C.4 does not hold).

*Proof of Proposition C.5.* We rewrite the primal problem (18) in the form

$$\inf_q \psi(q) + g(\mathcal{A}q),$$

$$\psi(q) = \begin{cases} \lambda \int q(\boldsymbol{x}) \log(q(\boldsymbol{x})) P(d\boldsymbol{x}) & \text{if } q \geq 0 \\ +\infty & \text{else} \end{cases}, \quad g(\boldsymbol{y}_0, \boldsymbol{y}_1) = \begin{cases} \|\boldsymbol{y}_1 - \boldsymbol{h}^*\|^2 & \text{if } \boldsymbol{y}_0 = 1 \\ +\infty & \text{else} \end{cases},$$

$$\mathcal{A}(q) = \left(\int q(\boldsymbol{x}) P(d\boldsymbol{x}), \int \boldsymbol{h}(\boldsymbol{x}) q(\boldsymbol{x}) P(d\boldsymbol{x})\right)$$

defined on the space of $X$-measurable functions $q$ for which $\int |q(\boldsymbol{x})| P(d\boldsymbol{x}) < \infty$ and $\int \|\boldsymbol{h}(\boldsymbol{x})\| |q(\boldsymbol{x})| P(d\boldsymbol{x}) < \infty$. Here, $q$ represents the density of measure $Q$ with respect to $P$. $g(\mathcal{A}q)$ is equal to $\|\mathbb{E}_Q[\boldsymbol{h}(\boldsymbol{x})] - \boldsymbol{h}^*\|^2$ if $Q$ is a probability measure and is infinite otherwise. $\psi(q)$ is equal to the KL divergence between $Q$ and $P$, scaled by $\lambda$.

As in the proof of Theorem C.3, $\mathcal{A}$ is a bounded, linear map defined on this space, and $\psi$ and $g$ are convex. By Fenchel-Rockafellar duality (Borwein & Zhu, 2005, Theorem 4.4.2), weak duality holds for the problem (18) and its dual (19).

By the remark following Lemma C.11, $F_\lambda$ is infinitely differentiable on $\Xi$, and taking two derivatives of $F_\lambda(\boldsymbol{\alpha})$ yields

$$\nabla_{\boldsymbol{\alpha}} F_\lambda(\boldsymbol{\alpha}) = \lambda\left(-\frac{\lambda}{2}\boldsymbol{\alpha} - \nabla_{\boldsymbol{\alpha}} A_P(\boldsymbol{\alpha}) + \boldsymbol{h}^*\right), \quad \nabla_{\boldsymbol{\alpha}}^2 F_\lambda(\boldsymbol{\alpha}) = \lambda\left(-\frac{\lambda}{2}\mathbb{I} - \nabla_{\boldsymbol{\alpha}}^2 A_P(\boldsymbol{\alpha})\right).$$

Since $\nabla_{\boldsymbol{\alpha}}^2 A_P(\boldsymbol{\alpha})$ is positive semi-definite, then the problem (19) is strongly concave. And by our assumption that $\Xi$ is open, $F_\lambda(\boldsymbol{\alpha})$ is equal to $-\infty$ for $\boldsymbol{\alpha}$ belonging to the boundary of $\Xi$. Together with strong concavity, this implies a unique maximizer $\boldsymbol{\alpha}_\lambda$ of $F_\lambda$ exists.

In particular, $\boldsymbol{\alpha}_\lambda$ is the unique $\boldsymbol{\alpha} \in \mathbb{R}^d$ that satisfies the fixed-point equation

$$\nabla_{\boldsymbol{\alpha}} F_\lambda(\boldsymbol{\alpha}) = \lambda\left(-\frac{\lambda}{2}\boldsymbol{\alpha} - \nabla_{\boldsymbol{\alpha}} A(\boldsymbol{\alpha}) + \boldsymbol{h}^*\right) = \boldsymbol{0} \implies \boldsymbol{\alpha} = -\frac{2}{\lambda}(\nabla_{\boldsymbol{\alpha}} A_P(\boldsymbol{\alpha}) - \boldsymbol{h}^*).$$

And the probability measure $Q_{\boldsymbol{\alpha}_\lambda} \propto \exp\{r_{\boldsymbol{\alpha}_\lambda}(\boldsymbol{x})\} P(d\boldsymbol{x})$ satisfies

$$\lambda D_{KL}(Q_{\boldsymbol{\alpha}_\lambda} \| P) + \|\mathbb{E}_{Q_{\boldsymbol{\alpha}_\lambda}}[\boldsymbol{h}(\boldsymbol{x})] - \boldsymbol{h}^*\|$$

$$= \lambda(\boldsymbol{\alpha}_\lambda^\top \nabla_{\boldsymbol{\alpha}} A_P(\boldsymbol{\alpha}_\lambda) - A_P(\boldsymbol{\alpha}_\lambda)) + \frac{\lambda^2}{4}\|\boldsymbol{\alpha}_\lambda\|^2$$

$$= \lambda\left(\boldsymbol{\alpha}_\lambda^\top\left(\boldsymbol{h}^* - \frac{\lambda}{2}\boldsymbol{\alpha}_\lambda\right) - A_P(\boldsymbol{\alpha}_\lambda)\right) + \frac{\lambda^2}{4}\|\boldsymbol{\alpha}_\lambda\|^2$$

$$= F_\lambda(\boldsymbol{\alpha}_\lambda).$$

By weak duality, this implies $Q_\lambda := Q_{\boldsymbol{\alpha}_\lambda}$ is optimal for the primal problem. Moreover, strict convexity of $\psi$ implies that the optimum of the primal problem is unique. $\square$

Next, we show that as the regularization parameter $\lambda \to 0$, then $Q_\lambda$ achieves the minimum possible Euclidean norm constraint violation i.e., Euclidean norm difference between $\mathbb{E}_{Q_\lambda}[\boldsymbol{h}]$ and $\boldsymbol{h}^*$. We also give a finite $\lambda$ bound on the constraint violation.

**Proposition C.6.** *The distribution $Q_\lambda$ satisfies*

$$\lim_{\lambda \to 0} \|\mathbb{E}_{Q_\lambda}[\boldsymbol{h}(\boldsymbol{x})] - \boldsymbol{h}^*\| = \inf_{\substack{Q \in \mathcal{P}(P) \\ D_{KL}(Q \| P) < \infty}} \|\mathbb{E}_Q[\boldsymbol{h}(\boldsymbol{x})] - \boldsymbol{h}^*\|.$$

*Moreover, we have the finite-sample bound on the Euclidean norm constraint violation of $Q_\lambda$*

$$\|\mathbb{E}_{Q_\lambda}[\boldsymbol{h}(\boldsymbol{x})] - \boldsymbol{h}^*\| \leq \inf_{Q \in \mathcal{P}(P)} \left\{ \sqrt{\lambda D_{KL}(Q \| P)} + \|\mathbb{E}_Q[\boldsymbol{h}(\boldsymbol{x})] - \boldsymbol{h}^*\| \right\}.$$

*Proof.* Fix $\varepsilon > 0$ and let $Q_\varepsilon$ be such that $\|\mathbb{E}_{Q_\varepsilon}[\boldsymbol{h}(\boldsymbol{x})] - \boldsymbol{h}^*\| \leq \inf_{Q \in \mathcal{P}(P)} \|\mathbb{E}_Q[\boldsymbol{h}(\boldsymbol{x})] - \boldsymbol{h}^*\| + \varepsilon$. Then by the optimality of $Q_\lambda$ for the objective (18),

$$\|\mathbb{E}_{Q_\lambda}[\boldsymbol{h}(\boldsymbol{x})] - \boldsymbol{h}^*\|^2 \leq \lambda D_{\mathrm{KL}}(Q_\lambda \| P) + \|\mathbb{E}_{Q_\lambda}[\boldsymbol{h}(\boldsymbol{x})] - \boldsymbol{h}^*\|^2$$
$$\leq \lambda D_{\mathrm{KL}}(Q_\varepsilon \| P) + \|\mathbb{E}_{Q_\varepsilon}[\boldsymbol{h}(\boldsymbol{x})] - \boldsymbol{h}^*\|^2. \tag{20}$$

Our choice of $Q_\varepsilon$ yields

$$\|\mathbb{E}_{Q_\lambda}[\boldsymbol{h}(\boldsymbol{x})] - \boldsymbol{h}^*\|^2 \leq \lambda D_{\mathrm{KL}}(Q_\varepsilon \| P) + \inf_{Q \in \mathcal{P}(P)} \|\mathbb{E}_Q[\boldsymbol{h}(\boldsymbol{x})] - \boldsymbol{h}^*\| + \varepsilon.$$

Taking $\lambda \to 0$ and then $\varepsilon \to 0$ yields the first result. Replacing $Q_\varepsilon$ with $Q \in \mathcal{P}(P)$ in (20) and taking the infimum over $Q$ yields the second result. $\square$

In the setting of Proposition C.9 where a solution to the maximum entropy problem exists, then a bound on the Euclidean norm constraint violation of $Q_\lambda$ is simply $\sqrt{\lambda D_{\mathrm{KL}}(Q^* \| P)}$. This implies that $\|\mathbb{E}_{Q_\lambda}[\boldsymbol{h}(\boldsymbol{x})] - \boldsymbol{h}^*\| = \mathcal{O}(\sqrt{\lambda})$.

From our proof of Proposition C.6, it is clear that we did not take advantage of the structure to the solution $Q_\lambda$. When Assumptions C.1 and C.4 hold, we can obtain a faster rate of convergence of $\mathbb{E}_{Q_\lambda}[\boldsymbol{h}(\boldsymbol{x})]$ to $\boldsymbol{h}^*$, and we can show that $\boldsymbol{\alpha}_\lambda$ converges to the parameters $\boldsymbol{\alpha}^*$ of the maximum entropy distribution.

**Proposition C.7.** *Suppose Assumptions C.1, C.2, and C.4 hold, which imply that the maximum entropy solution $dQ^*/dP \propto \exp\{r_{\boldsymbol{\alpha}^*}(\boldsymbol{x})\}$ exists. Then $\boldsymbol{\alpha}_\lambda \to \boldsymbol{\alpha}^*$ as $\lambda \to 0$. In particular,*

*(i)* $\|\boldsymbol{\alpha}_\lambda - \boldsymbol{\alpha}^*\| = \mathcal{O}(\lambda)$

*(ii)* $\|\mathbb{E}_{Q_\lambda}[\boldsymbol{h}(\boldsymbol{x})] - \boldsymbol{h}^*\| = \mathcal{O}(\lambda)$

*(iii)* $|D_{KL}(Q_\lambda \| P) - D_{KL}(Q^* \| P)| = \mathcal{O}(\lambda)$.

*Proof.* Prior to proving (i)-(iii), we first establish $\|\boldsymbol{\alpha}_\lambda - \boldsymbol{\alpha}^*\| = o(1)$. From the proof of Proposition C.5, we know that $\boldsymbol{\alpha}_\lambda$ maximizes $\lambda^{-1} F_\lambda(\boldsymbol{\alpha}) = -\frac{\lambda}{4}\|\boldsymbol{\alpha}\|^2 - A_P(\boldsymbol{\alpha}) + \boldsymbol{\alpha}^\top \boldsymbol{h}^*$ for each $\lambda > 0$. And from (15), we know that $\boldsymbol{\alpha}^*$ maximizes $F_0(\boldsymbol{\alpha}) = -A_P(\boldsymbol{\alpha}) + \boldsymbol{\alpha}^\top \boldsymbol{h}^*$. Clearly, $F_\lambda(\boldsymbol{\alpha}) \to F_0(\boldsymbol{\alpha})$ pointwise as $\lambda \to 0$. Since each of $F_\lambda$ and $F_0$ is concave on $\Xi$, a classical result in convex analysis Rockafellar (1970, Theorem, 10.8) implies that the convergence $F_\lambda(\boldsymbol{\alpha}) \to F_0(\boldsymbol{\alpha})$ is uniform on closed, bounded subsets of $\Xi$ containing $\boldsymbol{\alpha}^*$.

Fix $\epsilon > 0$ such that the Euclidean ball of radius $\epsilon$ centered at $\boldsymbol{\alpha}^*$ is contained in $\Xi$. By Assumption C.4, $F_0$ is strictly concave, since then $\nabla_{\boldsymbol{\alpha}}^2 A_P(\boldsymbol{\alpha})$ positive definite for every $\boldsymbol{\alpha} \in \Xi$. Hence, there exists a $\kappa$ such that for all $\|\boldsymbol{\alpha} - \boldsymbol{\alpha}^*\| = \epsilon$,

$$F_0(\boldsymbol{\alpha}) > \kappa > F_0(\boldsymbol{\alpha}^*).$$

This is because the left-hand side of the above inequality attains its minimum on the compact set $\|\boldsymbol{\alpha} - \boldsymbol{\alpha}^*\| = \epsilon$ and (ii) by strict concavity this minimum must be strictly greater than the right-hand

side. Moreover, by uniform convergence of $F_\lambda$ to $F_0$, there exists $\lambda_\epsilon > 0$ such that for all $\lambda < \lambda_\epsilon$ and all $\|\boldsymbol{\alpha} - \boldsymbol{\alpha}^*\| = \epsilon$

$$F_\lambda(\boldsymbol{\alpha}) > \kappa > F_\lambda(\boldsymbol{\alpha}^*). \tag{21}$$

Since $F_\lambda$ is also concave, (21) implies that the maximizer of $F_\lambda$, $\boldsymbol{\alpha}_\lambda$, must lie in the Euclidean ball of radius $\epsilon$ centered at $\boldsymbol{\alpha}^*$. This establishes $\|\boldsymbol{\alpha}_\lambda - \boldsymbol{\alpha}^*\| = o(1)$.

We are now prepared to prove (i). By Taylor expanding $\nabla_{\boldsymbol{\alpha}} A_P(\boldsymbol{\alpha})$ at $\boldsymbol{\alpha}_\lambda$ about $\boldsymbol{\alpha}^*$, we obtain

$$\nabla_{\boldsymbol{\alpha}} A_P(\boldsymbol{\alpha}_\lambda) = \boldsymbol{h}^* + \nabla_{\boldsymbol{\alpha}}^2 A_P(\boldsymbol{\alpha}^*)(\boldsymbol{\alpha}_\lambda - \boldsymbol{\alpha}^*) + \boldsymbol{r}_\lambda, \quad \|\boldsymbol{r}_\lambda\| = o(\|\boldsymbol{\alpha}_\lambda - \boldsymbol{\alpha}^*\|). \tag{22}$$

By Proposition C.5, $\boldsymbol{\alpha}_\lambda$ satisfies $\boldsymbol{\alpha}_\lambda = -\frac{2}{\lambda}(\nabla_{\boldsymbol{\alpha}} A(\boldsymbol{\alpha}_\lambda) - \boldsymbol{h}^*)$. Multiplying (22) by $-2/\lambda$ and substituting in this expression for $\boldsymbol{\alpha}_\lambda$ yields

$$\boldsymbol{\alpha}_\lambda = -\frac{2}{\lambda}\nabla_{\boldsymbol{\alpha}}^2 A_P(\boldsymbol{\alpha}^*)(\boldsymbol{\alpha}_\lambda - \boldsymbol{\alpha}^*) + \frac{1}{\lambda}\boldsymbol{r}_\lambda.$$

Solving for $\boldsymbol{\alpha}_\lambda - \boldsymbol{\alpha}^*$ yields

$$\boldsymbol{\alpha}_\lambda - \boldsymbol{\alpha}^* = -\left(\mathbb{I} + \frac{2}{\lambda}\nabla_{\boldsymbol{\alpha}}^2 A_P(\boldsymbol{\alpha}^*)\right)^{-1}\left(\boldsymbol{\alpha}^* + \frac{1}{\lambda}\boldsymbol{r}_\lambda\right)$$

$$= -\lambda\left(\lambda\mathbb{I} + 2\nabla_{\boldsymbol{\alpha}}^2 A_P(\boldsymbol{\alpha}^*)\right)^{-1}\boldsymbol{\alpha}^* + \tilde{\boldsymbol{r}}_\lambda \tag{23}$$

for $\tilde{\boldsymbol{r}}_\lambda = o(\|\boldsymbol{\alpha}_\lambda - \boldsymbol{\alpha}^*\|)$. And because $\|\boldsymbol{\alpha}_\lambda - \boldsymbol{\alpha}^*\| = o(1)$, then for all $\lambda$ sufficiently small, $\|\tilde{\boldsymbol{r}}_\lambda\| \le \frac{1}{2}\|\boldsymbol{\alpha}_\lambda - \boldsymbol{\alpha}^*\|$. Taking the norm of both sides of (23) and rearranging yields

$$\|\boldsymbol{\alpha}_\lambda - \boldsymbol{\alpha}^*\| \le 2\lambda\|\left(\lambda\mathbb{I} + 2\nabla_{\boldsymbol{\alpha}}^2 A_P(\boldsymbol{\alpha}^*)\right)^{-1}\boldsymbol{\alpha}^*\|$$

for all $\lambda$ sufficiently small. This proves (i).

For (ii), the relationship $\boldsymbol{\alpha}_\lambda = -\frac{2}{\lambda}(\nabla_{\boldsymbol{\alpha}} A(\boldsymbol{\alpha}_\lambda) - \boldsymbol{h}^*)$ yields

$$\|\mathbb{E}_{Q_\lambda}[\boldsymbol{h}(\boldsymbol{x})] - \boldsymbol{h}^*\| = \|\nabla_{\boldsymbol{\alpha}} A(\boldsymbol{\alpha}_\lambda) - \boldsymbol{h}^*\| \le \frac{\lambda}{2}\|\boldsymbol{\alpha}_\lambda - \boldsymbol{\alpha}^*\| + \frac{\lambda}{2}\|\boldsymbol{\alpha}^*\| = \mathcal{O}(\lambda).$$

Lastly for (iii),

$$\begin{aligned}
D_{\mathrm{KL}}(Q_\lambda \parallel P) &= \boldsymbol{\alpha}_\lambda^\top \mathbb{E}_{Q_\lambda}[\boldsymbol{h}(\boldsymbol{x})] - A_P(\boldsymbol{\alpha}_\lambda) \\
&= (\boldsymbol{\alpha}_\lambda^\top \boldsymbol{h}^* + \mathcal{O}(\lambda)) - \{A_P(\boldsymbol{\alpha}^*) + \nabla_{\boldsymbol{\alpha}} A_P(\boldsymbol{\alpha}^*)^\top(\boldsymbol{\alpha}_\lambda - \boldsymbol{\alpha}^*) + o(\|\boldsymbol{\alpha}_\lambda - \boldsymbol{\alpha}^*\|)\} \\
&= \boldsymbol{\alpha}_\lambda^\top \boldsymbol{h}^* - A_P(\boldsymbol{\alpha}^*) + \mathcal{O}(\lambda) \\
&= D_{\mathrm{KL}}(Q^* \parallel P) + \mathcal{O}(\lambda).
\end{aligned}$$

$\square$

In Section 2.2 we derived the reward loss as the KL divergence of the model $p_\theta$ to the maximum entropy solution $p_{\boldsymbol{\alpha}^*}$. The relax loss can also be viewed as a divergence to a tilt of the base model $p_{\theta_{\mathrm{base}}}$, except that the tilt depends on the current model $p_\theta$. In particular, the stationary points of the relaxed loss are exactly the stationary points of the objective

$$D_{\mathrm{KL}}(p_\theta \parallel p_{\theta_{\mathrm{base}}}) + \frac{2}{\lambda}(\mathbb{E}_{p_{\mathrm{sg}(\theta)}}[\boldsymbol{h}(\boldsymbol{x})] - \boldsymbol{h}^*)^\top \mathbb{E}_{p_\theta}[\boldsymbol{h}(\boldsymbol{x})]. \tag{24}$$

This can be seen by taking the gradient of (24). By identifying $\boldsymbol{\alpha} = -\frac{2}{\lambda}(\mathbb{E}_{p_{\mathrm{sg}(\theta)}}[\boldsymbol{h}(\boldsymbol{x})] - \boldsymbol{h}^*)$ and $q_{\boldsymbol{\alpha}} \propto p_{\theta_{\mathrm{base}}}(\boldsymbol{x})\exp\{r_{\boldsymbol{\alpha}}(\boldsymbol{x})\}$, we observe that (24) is exactly equal to $D_{\mathrm{KL}}(p_\theta \parallel q_{\boldsymbol{\alpha}})$. $q_{\boldsymbol{\alpha}}$ *can be understood as our current best approximation to the solution of* (18). Unlike the solution of (18), though, $\mathbb{E}_{q_{\boldsymbol{\alpha}}}[\boldsymbol{h}(\boldsymbol{x})]$ is not equal to $\mathbb{E}_{p_{\mathrm{sg}(\theta)}}[\boldsymbol{h}(\boldsymbol{x})]$. For sufficiently expressive class of generative models $p_\theta$, we would expect $\mathbb{E}_{q_{\boldsymbol{\alpha}}}[\boldsymbol{h}(\boldsymbol{x})]$ and $\mathbb{E}_{p_{\mathrm{sg}(\theta)}}[\boldsymbol{h}(\boldsymbol{x})]$ to be approximately equal at the optimum.

## C.4 CONSISTENCY AND ASYMPTOTIC NORMALITY

In this section, we discuss the large sample behavior of the estimator $\widehat{\boldsymbol{\alpha}}_N$ for the parameters $\boldsymbol{\alpha}^*$ of the reward loss. Under Assumptions C.1, C.2, and C.4, we show that as $N \to \infty$ and $d$ remains fixed, then $\widehat{\boldsymbol{\alpha}}_N$ is close to $\boldsymbol{\alpha}^*$ with high probability. And under stronger conditions, we demonstrate that $\widehat{\boldsymbol{\alpha}}_N$ has a limiting normal distribution. The asymptotics of $\widehat{\boldsymbol{\alpha}}_N$ have previously been studied in the subject of empirical likelihood (Qin & Lawless, 1994; Kitamura & Stutzer, 1997; Owen, 2001).

We first aim to establish that $\widehat{\boldsymbol{\alpha}}_N$ is close to $\boldsymbol{\alpha}^*$ with high probability as $N \to \infty$ i.e., $\widehat{\boldsymbol{\alpha}}_N$ is *consistent* for $\boldsymbol{\alpha}^*$. Define the functions

$$A(\boldsymbol{\alpha}) := A_{p_{\theta_{\text{base}}}}(\boldsymbol{\alpha}), \quad A_N(\boldsymbol{\alpha}) := \log \left( \frac{1}{N} \sum_{n=1}^{N} \exp\{r_{\boldsymbol{\alpha}}(\boldsymbol{x}_n)\} \right),$$

where $A_P$ is defined in Appendix C.1. Observe that $A_N$ is random and depends on the independent samples $\{\boldsymbol{x}_n\}_{n=1}^N$ drawn from $p_{\theta_{\text{base}}}$. The dual problem corresponding to $p_{\theta_{\text{base}}}$ maximizes $\boldsymbol{\alpha}^\top \boldsymbol{h}(\boldsymbol{x}) - A(\boldsymbol{\alpha})$, whereas the dual problem corresponding to the distribution of samples $\{\boldsymbol{x}_n\}_{n=1}^N$ maximizes $\boldsymbol{\alpha}^\top \boldsymbol{h}(\boldsymbol{x}) - A_N(\boldsymbol{\alpha})$. By the Strong Law of Large Numbers (SLLN), for any $\boldsymbol{\alpha} \in \Xi$, $A_N(\boldsymbol{\alpha}) \to A(\boldsymbol{\alpha})$ with $p_{\theta_{\text{base}}}$ probability one. In order for our estimator $\widehat{\boldsymbol{\alpha}}_N$ to approach $\boldsymbol{\alpha}^*$, though, we need to argue that the dual objective corresponding to $\{\boldsymbol{x}_n\}_{n=1}^N$ *uniformly* approaches the dual objective corresponding to $p_{\theta_{\text{base}}}$ on some neighborhood containing $\boldsymbol{\alpha}^*$.

**Lemma C.8.** *For any closed, bounded subset $K$ of $\Xi$,*

$$\sup_{\boldsymbol{\alpha} \in K} |A_N(\boldsymbol{\alpha}) - A(\boldsymbol{\alpha})| \to 0$$

*with $p_{\theta_{base}}$ probability one.*

*Proof.* By the SLLN, we can construct a Borel set $\widetilde{N}$ of probability zero under $p_{\theta_{\text{base}}}$ such that on its complement $A_N(\boldsymbol{\alpha}) \to A(\boldsymbol{\alpha})$ holds for each $\boldsymbol{\alpha} \in \Xi \cap \mathbb{Q}^d$ (apply the SLLN for an individual $\boldsymbol{\alpha} \in \Xi \cap \mathbb{Q}^d$, then take a union over probability zero sets).

Rockafellar (1970, Theorem 10.8) states that if a sequence of finite convex functions defined on an open, convex set $C$ converges pointwise on a dense subset of $C$ to a limiting function, then the limiting function is convex on $C$, and the convergence is uniform on closed and bounded subsets of $C$. Applying this result to our setting, on the complement of $\widetilde{N}$

$$\sup_{\boldsymbol{\alpha} \in K} |A_N(\boldsymbol{\alpha}) - A(\boldsymbol{\alpha})| \to 0$$

for $K$ a closed and bounded subset of $\Xi$. $\qquad\qquad\square$

Once we have proven uniform convergence, our proof of consistency for $\widehat{\boldsymbol{\alpha}}_N$ is nearly identical to our proof that $\|\boldsymbol{\alpha}_\lambda - \boldsymbol{\alpha}^*\| = o(1)$ in Proposition C.7.

**Proposition C.9** (Consistency of $\widehat{\boldsymbol{\alpha}}_N$). *Suppose Assumptions C.1, C.2, and C.4 hold. For any $\epsilon > 0$,*

$$\mathbb{P}_{p_{\theta_{base}}}(\|\widehat{\boldsymbol{\alpha}}_N - \boldsymbol{\alpha}^*\| > \epsilon) \to 0 \quad \text{as } N \to \infty.$$

*Proof.* From Appendix C.1, we know that both $A$ and $A_N$ are convex functions. Moreover by Assumption C.4, $A$ is strictly convex.

From Lemma C.8, there exists a closed, bounded subset $K$ of containing $\boldsymbol{\alpha}^*$ on which $\sup_{\boldsymbol{\alpha} \in K} |A_N(\boldsymbol{\alpha}) - A(\boldsymbol{\alpha})| \to 0$ with $p_{\theta_{\text{base}}}$ probability one. Since $\Xi$ is open (Assumption C.2), $K$ can be chosen to have positive diameter. Fix $\epsilon > 0$ sufficiently small such that the Euclidean ball centered at $\boldsymbol{\alpha}^*$ of radius $\epsilon$ is contained in $K$. Just as in the proof of Proposition C.7, there exists some $\kappa \in \mathbb{R}$ such that for all $\|\boldsymbol{\alpha} - \boldsymbol{\alpha}^*\| = \epsilon$,

$$\boldsymbol{\alpha}^\top \boldsymbol{h}^* - A(\boldsymbol{\alpha}) < \kappa < (\boldsymbol{\alpha}^*)^\top \boldsymbol{h}^* - A(\boldsymbol{\alpha}^*).$$

Fix $\delta > 0$. By uniform convergence, there exists $N_{\epsilon,\delta} \in \mathbb{N}$ such that $\forall N \geq N_{\epsilon,\delta}$ and for all $\|\boldsymbol{\alpha} - \boldsymbol{\alpha}^*\| = \epsilon$,

$$\boldsymbol{\alpha}^\top \boldsymbol{h}^* - A_N(\boldsymbol{\alpha}) < \kappa < (\boldsymbol{\alpha}^*)^\top \boldsymbol{h}^* - A_N(\boldsymbol{\alpha}^*).$$

with probability at least $1 - \delta$ under $p_{\theta_{\text{base}}}$. And since the dual objective corresponding to $\{\boldsymbol{x}_n\}_{n=1}^N$ is concave, this implies that, on this event, its maximum occurs in the Euclidean ball of radius $\epsilon$.

In other words, we have proven that for every $\epsilon > 0, \delta > 0$, there exists $N_{\epsilon,\delta}$ such that for every $N \geq N_{\epsilon,\delta}$,

$$\mathbb{P}_{p_{\theta_{\text{base}}}} \left( \|\widehat{\boldsymbol{\alpha}}_N - \boldsymbol{\alpha}^*\| > \epsilon \right) \leq \delta.$$

$\square$

Next, we show that under stronger conditions on the problem, $\widehat{\boldsymbol{\alpha}}_N$ has a normal limiting distribution, and we derive its variance.

**Proposition C.10** (Asymptotic normality of $\widehat{\boldsymbol{\alpha}}_N$). *Suppose Assumptions C.1, C.2, and C.4 hold. Moreover, assume $2\boldsymbol{\alpha}^* \in \Xi$, for $\Xi$ defined in Appendix C.1. Then the estimator $\widehat{\boldsymbol{\alpha}}_N$ is asymptotically normal:*

$$\sqrt{N}(\widehat{\boldsymbol{\alpha}}_N - \boldsymbol{\alpha}^*) \xrightarrow{d} \mathcal{N}(\boldsymbol{0}, (Var_{p_{\boldsymbol{\alpha}^*}}[\boldsymbol{h}(\boldsymbol{x})])^{-1} \boldsymbol{\Sigma} (Var_{p_{\boldsymbol{\alpha}^*}}[\boldsymbol{h}(\boldsymbol{x})])^{-1}),$$

$$\boldsymbol{\Sigma} = \frac{\mathbb{E}_{p_{\theta_{\text{base}}}}[(\boldsymbol{h}(\boldsymbol{x}) - \boldsymbol{h}^*)(\boldsymbol{h}(\boldsymbol{x}) - \boldsymbol{h}^*)^\top \exp\{r_{2\boldsymbol{\alpha}^*}(\boldsymbol{x})\}]}{(\mathbb{E}_{p_{\theta_{\text{base}}}}[\exp\{r_{\boldsymbol{\alpha}^*}(\boldsymbol{x})\}])^2}.$$

Prior to stating the proof of Proposition C.10, we build some intuition by working out the asymptotic variance for the example we presented at the beginning of the section. Recall that the constraint function is $\boldsymbol{h}(\boldsymbol{x}) = \mathbb{1}\{\boldsymbol{x} > 0\}$, $\boldsymbol{h}^*$ is its target value, and $h_b = \mathbb{P}_{\theta_{\text{base}}}(\boldsymbol{x} > 0)$ is the expected value of $\boldsymbol{h}$ under $p_{\theta_{\text{base}}}$. By directly solving for $\widehat{\boldsymbol{\alpha}}_N$ in the expression (5) for the maximum entropy solution, we showed $\widehat{\boldsymbol{\alpha}}_N = \log(\frac{\boldsymbol{h}^*(1-\bar{\boldsymbol{y}}_N)}{(1-\boldsymbol{h}^*)\bar{\boldsymbol{y}}_N})$, where $\bar{\boldsymbol{y}}_N = \frac{1}{N}\sum_{n=1}^N \boldsymbol{y}_n$, $\boldsymbol{y}_n = \boldsymbol{h}(\boldsymbol{x}_n) \stackrel{d}{=} \text{Bernoulli}(h_b)$ for $\boldsymbol{x}_n \stackrel{i.i.d.}{\sim} p_{\theta_{\text{base}}}$. Next, we compute

$$Var_{p_{\boldsymbol{\alpha}^*}}[\boldsymbol{h}(\boldsymbol{x})] = \boldsymbol{h}^*(1 - \boldsymbol{h}^*)$$

$$\boldsymbol{\Sigma} = \frac{(\boldsymbol{h}^*)^2(1 - h_b) + (1 - \boldsymbol{h}^*)^2 \exp(2\boldsymbol{\alpha}^*)h_b}{(h_b \exp(\boldsymbol{\alpha}^*) + (1 - h_b))^2} = \frac{(\boldsymbol{h}^*)^2(1 - h_b) + \frac{(1-h_b)^2(\boldsymbol{h}^*)^2}{h_b}}{\left(\frac{1-h_b}{1-\boldsymbol{h}^*}\right)^2} = \frac{(\boldsymbol{h}^*)^2(1 - \boldsymbol{h}^*)^2}{h_b(1 - h_b)}.$$

Combining these two yields the asymptotic variance

$$Var_{p_{\boldsymbol{\alpha}^*}}[\boldsymbol{h}(\boldsymbol{x})]^{-2} \boldsymbol{\Sigma} = \frac{1}{h_b(1 - h_b)},$$

according to Proposition C.10. In other words, the estimator $\widehat{\boldsymbol{\alpha}}_N$ has greatest asymptotic variance when $h_b$ is close to either $0$ or $1$. Notice that we can compute the asymptotic variance of $\widehat{\boldsymbol{\alpha}}_N$ directly (i.e., without using Proposition C.10) by applying the delta method to $\bar{\boldsymbol{y}}_N$ and the function $z \mapsto \log(\frac{\boldsymbol{h}^*(1-z)}{(1-\boldsymbol{h}^*)z})$, in which case we obtain the same value.

The proof of Proposition C.10 relies on the technical result Lemma C.11, the statement and proof of which we defer to the end of the section.

*Proof of Proposition C.10.* Let $D_N$ be the set on which the strong duality holds for $P = \frac{1}{N}\sum_{n=1}^N \delta_{\boldsymbol{x}_n}$ and the dual optimum is uniquely achieved. From the proof of Proposition C.9, we can see $\mathbb{P}_{p_{\theta_{\text{base}}}}(D_N) \to 1$ as $N \to \infty$. Moreover, on the set $D_N$, $\widehat{\boldsymbol{\alpha}}_N$ is the unique root of

$$\frac{1}{N}\sum_{n=1}^N \psi(\boldsymbol{x}_n, \boldsymbol{\alpha}) = 0, \quad \psi(\boldsymbol{x}, \boldsymbol{\alpha}) := (\boldsymbol{h}(\boldsymbol{x}) - \boldsymbol{h}^*)\exp\{r_{\boldsymbol{\alpha}}(\boldsymbol{x})\}.$$

Also, from the proof of Proposition C.9, we know that Assumption C.4 implies $Var_{p_{\boldsymbol{\alpha}^*}}[\boldsymbol{h}(\boldsymbol{x})]$ is positive definite.

By Van der Vaart (2000, Theorem 5.21), if we can show $\psi(\boldsymbol{x}, \boldsymbol{\alpha})$ satisfies the Lipschitz condition

$$\|\psi(\boldsymbol{x}, \boldsymbol{\alpha}) - \psi(\boldsymbol{x}, \boldsymbol{\alpha}')\| \leq M(\boldsymbol{x})\|\boldsymbol{\alpha} - \boldsymbol{\alpha}'\| \tag{25}$$

for all $\boldsymbol{\alpha}, \boldsymbol{\alpha}'$ belonging to some neighborhood of $\boldsymbol{\alpha}^*$ and $\mathbb{E}_{p_{\theta_{\text{base}}}}[M(\boldsymbol{x})^2] < \infty$, then the previous facts imply that $\widehat{\boldsymbol{\alpha}}_N$ is asymptotically normal with variance

$$\underbrace{\mathbb{E}_{p_{\theta_{\text{base}}}}[(\boldsymbol{h}(\boldsymbol{x}) - \boldsymbol{h}^*)\boldsymbol{h}(\boldsymbol{x})^\top \exp\{r_{\boldsymbol{\alpha}^*}(\boldsymbol{x})\}]^{-1}(\mathbb{E}_{p_{\theta_{\text{base}}}}[\exp\{r_{\boldsymbol{\alpha}^*}(\boldsymbol{x})\}])}_{=\text{Var}_{p_{\boldsymbol{\alpha}^*}}[\boldsymbol{h}(\boldsymbol{x})]^{-1}} \boldsymbol{\Sigma}$$

$$\underbrace{(\mathbb{E}_{p_{\theta_{\text{base}}}}[\exp\{r_{\boldsymbol{\alpha}^*}(\boldsymbol{x})\}])(\mathbb{E}_{p_{\theta_{\text{base}}}}[(\boldsymbol{h}(\boldsymbol{x}) - \boldsymbol{h}^*)\boldsymbol{h}(\boldsymbol{x})^\top \exp\{r_{\boldsymbol{\alpha}^*}(\boldsymbol{x})\}]^{-1})^\top}_{\text{Var}_{p_{\boldsymbol{\alpha}^*}}[\boldsymbol{h}(\boldsymbol{x})]^{-1}}.$$

And so it remains only to establish the Lipschitz condition (25). First, we compute the derivative of $\psi$ with respect to $\boldsymbol{\alpha}$

$$\nabla_{\boldsymbol{\alpha}}\psi(\boldsymbol{x}, \boldsymbol{\alpha}) = (\boldsymbol{h}(\boldsymbol{x}) - \boldsymbol{h}^*)\boldsymbol{h}(\boldsymbol{x})^\top \exp\{r_{\boldsymbol{\alpha}}(\boldsymbol{x})\} = \nabla_{\boldsymbol{\alpha}}^2 \exp\{r_{\boldsymbol{\alpha}}(\boldsymbol{x})\} - \boldsymbol{h}^*(\nabla_{\boldsymbol{\alpha}}\exp\{r_{\boldsymbol{\alpha}}(\boldsymbol{x})\})^\top$$

Next, we appeal to Lemma C.11, which tells us that for all $\boldsymbol{\alpha}$ belonging to an open neighborhood of $\boldsymbol{\alpha}^*$, the derivatives of $\exp\{r_{\boldsymbol{\alpha}}(\boldsymbol{x})\}$ have norm dominated by a function $M(\boldsymbol{x})$ that is $p_{\theta_{\text{base}}}$-square integrable. Also, by the Mean Value Theorem, for all $\boldsymbol{\alpha}, \boldsymbol{\alpha}'$ belonging to this neighborhood,

$$\psi(\boldsymbol{x}, \boldsymbol{\alpha}) - \psi(\boldsymbol{x}, \boldsymbol{\alpha}') = \nabla\psi(\boldsymbol{x}, \tilde{\boldsymbol{\alpha}})(\boldsymbol{\alpha} - \boldsymbol{\alpha}')$$

for some $\tilde{\boldsymbol{\alpha}}$ on the line segment connecting $\boldsymbol{\alpha}$ to $\boldsymbol{\alpha}'$. By taking the norm on both sides and using $\|\nabla\psi(\boldsymbol{x}, \tilde{\boldsymbol{\alpha}})\| \leq M(\boldsymbol{x})$, we obtain the Lipschitz condition (25). $\square$

**Lemma C.11.** *Under the assumptions of Proposition C.10, there exists an open neighborhood of $\boldsymbol{\alpha}^*$ on which all derivatives of $\exp\{r_{\boldsymbol{\alpha}}(\boldsymbol{x})\}$ with respect to $\boldsymbol{\alpha}$ are dominated by a $p_{\theta_{base}}$-square integrable function.*

*Proof.* In Proposition C.10, we assume $\mathbb{E}_{p_{\theta_{\text{base}}}}[\exp\{r_{2\boldsymbol{\alpha}^*}(\boldsymbol{x})\}] < \infty$; in other words, $2\boldsymbol{\alpha}^*$ is contained in the natural parameter space $\Xi$. Let $\varepsilon$ be defined such that the Euclidean ball of radius $\varepsilon$ centered at $2\boldsymbol{\alpha}^*$ is contained in $\Xi$. Fix any $\tilde{\boldsymbol{\alpha}}$ such that $\|\tilde{\boldsymbol{\alpha}} - \boldsymbol{\alpha}^*\| < \varepsilon/(2d)$, where $d$ is the dimension of the constraint $\boldsymbol{h}(\boldsymbol{x})$. Then by Cauchy-Schwarz

$$\exp\{\tilde{\boldsymbol{\alpha}}^\top \boldsymbol{h}(\boldsymbol{x})\} \leq \exp\{(\boldsymbol{\alpha}^*)^\top \boldsymbol{h}(\boldsymbol{x}) + \varepsilon/(2d)\|\boldsymbol{h}(\boldsymbol{x})\|\}. \tag{26}$$

Define the $2d$ vectors $(\boldsymbol{\beta}^{(\pm,l)})_{l=1}^d$ by $\boldsymbol{\beta}^{(+,l)} = \boldsymbol{e}[l]$, $\boldsymbol{\beta}^{(-,l)} = -\boldsymbol{e}[l]$, where $\boldsymbol{e}[l]$ denotes the $l$th standard basis vector. Then we can upper bound the second term using

$$\exp\{\|\boldsymbol{h}(\boldsymbol{x})\|\} \leq \prod_{l=1}^d \exp\{|\boldsymbol{h}_l(\boldsymbol{x})|\} \leq \prod_{l=1}^d (\exp\{\boldsymbol{h}_l(\boldsymbol{x})\} + \exp\{-\boldsymbol{h}_l(\boldsymbol{x})\}) \leq \sum_{l=1}^{2d} 2^d \exp\left\{d(\boldsymbol{\beta}^{(l)})^\top \boldsymbol{h}(\boldsymbol{x})\right\}.$$

Plugging this bound into (26) yields

$$\exp\{\tilde{\boldsymbol{\alpha}}^\top \boldsymbol{h}(\boldsymbol{x})\} \leq \sum_{l=1}^{2d} 2^d \exp\left\{(\boldsymbol{\alpha}^* + (\varepsilon/2)\boldsymbol{\beta}^{(l)})^\top \boldsymbol{h}(\boldsymbol{x})\right\}. \tag{27}$$

Squaring both sides of (27) yields

$$(\exp\{\tilde{\boldsymbol{\alpha}}^\top \boldsymbol{h}(\boldsymbol{x})\})^2 \leq 2^{2d} \sum_{l=1}^{2d} \sum_{k=1}^{2d} \exp\left\{(2\boldsymbol{\alpha}^* + (\varepsilon/2)(\boldsymbol{\beta}^{(l)} + \boldsymbol{\beta}^{(k)}))^\top \boldsymbol{h}(\boldsymbol{x})\right\}. \tag{28}$$

However, we notice $\|2\boldsymbol{\alpha}^* + (\varepsilon/2)(\boldsymbol{\beta}^{(l)} + \boldsymbol{\beta}^{(k)}) - 2\boldsymbol{\alpha}^*\| \leq \varepsilon$, so each term on the right-hand side of (28) has finite expectation under $p_{\theta_{\text{base}}}$. This implies $\exp\{r_{\boldsymbol{\alpha}}(\boldsymbol{x})\}$ is dominated by the right-hand side of (27), which is square integrable under $p_{\theta_{\text{base}}}$, for all $\|\boldsymbol{\alpha} - \boldsymbol{\alpha}^*\| < \varepsilon/(2d)$.

As for the derivatives of $\exp\{r_{\boldsymbol{\alpha}}(\boldsymbol{x})\}$, notice that the $k$th derivative with respect to $\boldsymbol{\alpha}$, $\nabla_{\boldsymbol{\alpha}}^{(k)} \exp\{r_{\boldsymbol{\alpha}}(\boldsymbol{x})\}$, is given by $\boldsymbol{h}(\boldsymbol{x})^{\otimes k} \exp\{r_{\boldsymbol{\alpha}}(\boldsymbol{x})\}$, where $\otimes$ denotes the tensor product. Moreover, by equivalence of norms, for any $\tau > 0$ there exists constants $c_k, c_{\tau,k} \geq 0$ such that $\|\boldsymbol{h}(\boldsymbol{x})^{\otimes k}\| \leq c_k \|\boldsymbol{h}(\boldsymbol{x})\|^k \leq c_{\tau,k} \exp\{\tau\|\boldsymbol{h}(\boldsymbol{x})\|\}$. So by choosing $\tau$ such that the Euclidean ball of radius $\varepsilon + 2d\tau$ centered at $2\boldsymbol{\alpha}^*$ is contained in $N(2\boldsymbol{\alpha}^*)$, our same argument yields a dominating function of the form (27) for $\|\boldsymbol{\alpha} - \boldsymbol{\alpha}^*\| < \varepsilon/(2d)$, with exponent $(\boldsymbol{\alpha}^* + (\varepsilon/2 + d\tau)\boldsymbol{\beta}^{(\ell)})^\top \boldsymbol{h}(\boldsymbol{x})$. $\square$

**Remark.** Under weaker assumptions (Assumptions C.1 and C.2), the proof of Lemma C.11 implies that the log-normalizer $A_P(\boldsymbol{\alpha})$ has derivatives of all orders on $\Xi$. Indeed, this is a consequence of equation (27), which implies that for every $\boldsymbol{\alpha}$, there exists a neighborhood of $\boldsymbol{\alpha}$ contained in $\Xi$ on which the $k$th derivative of $\exp\{r_{\boldsymbol{\alpha}}(\boldsymbol{x})\}$ is uniformly $p_{\theta_{\text{base}}}$-dominated. This allows one to exchange differentiation and integration in the definition of $A_P(\boldsymbol{\alpha})$.

# D SIMULATION EXPERIMENTS ADDITIONAL DETAILS

In this section, we provide details for our experiments calibrating mixture proportions in a product of GMMs (Section 3). First, in Appendix D.1 we give background on continuous-time diffusion models, including how we sample from $p_\theta$ and compute densities $p_\theta/p$ with respect to a dominating measure $p$. This enables us to employ CGM-relax and CGM-reward for calibrating a pre-trained diffusion model. In Appendix D.2, we describe how the base diffusion model can be initialized to generate exact samples from a GMM or product of GMMs. In Appendix D.3 we provide details regarding our implementation of the CGM calibration algorithm, including optimizer, neural network architecture, and hyperparameters. Finally, in Appendix D.4 we discuss how CGM-relax compares to an augmented Lagrangian method (Hestenes, 1969) on the same set of experiments.

## D.1 CONTINUOUS-TIME DIFFUSION MODELS

A continuous-time diffusion model is the solution to the $k$-dimensional stochastic differential equation (SDE)

$$d\boldsymbol{x}(t) = \boldsymbol{b}_\theta(\boldsymbol{x}(t), t)dt + \sigma(t)d\boldsymbol{w}(t), \ \boldsymbol{x}(0) \sim p_{\text{init}}, \tag{29}$$

where $(\boldsymbol{w}(t))_{0 \leq t \leq 1}$ is a standard $k$-dimensional Brownian motion, $\boldsymbol{b}_\theta$ is a neural network drift function, $\sigma$ is a diffusion coefficient, and $p_{\text{init}}$ is a known distribution from which sampling is tractable. Oksendal (2013, Theorem 5.2.1) provides conditions on $\boldsymbol{b}_\theta$ and $\sigma$ that ensure there exists a unique solution to the SDE (29). We denote the solution, which is a probability distribution on continuous paths, by $p_\theta$, and we write $p_\theta(\boldsymbol{x}(t))$ for the distribution of the state at time $t$.

**Sampling from diffusion models.** To sample from $p_\theta$, we use the Euler-Maruyama method. Specifically, we discretize $[0, 1]$ into $T$ time bins $[0, 1/T], \ldots, [(T-1)/T, 1]$ and sample a path $(\widehat{\boldsymbol{x}}(t))_{0 \leq t \leq 1}$ according to $\widehat{\boldsymbol{x}}(0) \sim p_{\text{init}}$

$$\widehat{\boldsymbol{x}}(t + \Delta t) = \widehat{\boldsymbol{x}}(t) + \Delta t \boldsymbol{b}_\theta(\widehat{\boldsymbol{x}}(t), t) + \sigma(t)\sqrt{\Delta t}\boldsymbol{z}(t), \ 0 < \Delta t \leq 1/T \tag{30}$$

for each $t = 0, 1/T, \ldots, (T-1)/T$, where $\boldsymbol{z}(0), \ldots, \boldsymbol{z}((T-1)/T)$ are independent standard multivariate normal random variables. The Euler-Maruyama method with additive noise $\sigma(t)$ has strong order of convergence 1, meaning its error in approximating the solution to the SDE (29) is

$$\mathbb{E}_{p_\theta}[\|\widehat{\boldsymbol{x}}(t) - \boldsymbol{x}(t)\|] \leq C(T^{-1}), \quad 0 \leq t \leq 1$$

for $C$ a constant independent of $T$. In other words, as we increase the number of time bins $T$, we can expect our sample paths drawn according to the Euler-Maruyama scheme to more faithfully approximate samples from the distribution $p_\theta$.

**Computing densities.** In order to employ CGM-relax and CGM-reward, $p_\theta$ and $p_{\theta_{\text{base}}}$ must have densities with respect to one another, and it must be possible to compute these densities. Girsanov's Theorem (Cameron & Martin, 1944; Girsanov, 1960) provides conditions that guarantee these densities to exist and an expression for computing them.

**Theorem D.1** (Girsanov's Theorem). *Suppose the SDEs*

$$\nu_1(\boldsymbol{x}) : d\boldsymbol{x}(t) = \boldsymbol{b}_1(\boldsymbol{x}(t), t)dt + \sigma(t)d\boldsymbol{w}(t), \quad 0 \leq t \leq 1$$
$$\nu_2(\boldsymbol{x}) : d\boldsymbol{x}(t) = (\boldsymbol{b}_1(\boldsymbol{x}(t), t) + \sigma(t)\boldsymbol{b}_2(\boldsymbol{x}(t), t))dt + \sigma(t)d\boldsymbol{w}(t), \quad 0 \leq t \leq 1$$

*satisfy $\sigma(t) > 0$, $0 < t < 1$, have the same initial law $\nu_1(\boldsymbol{x}_0) = \nu_2(\boldsymbol{x}_0)$, and admit unique, strong solutions, $\nu_1$ and $\nu_2$. Suppose also*

$$\left[\frac{\nu_2(\boldsymbol{x})}{\nu_1(\boldsymbol{x})}\right]_t := \exp\left\{\sum_{i=1}^{k}\int_0^t \boldsymbol{b}_2(\boldsymbol{x}(t), t)[i]d\boldsymbol{w}^{\nu_1}(t)[i] - \frac{1}{2}\int_0^t \|\boldsymbol{b}_2(\boldsymbol{x}(t), t)\|^2 dt\right\} \tag{31}$$

*is a $\nu_1$-martingale, where $(\boldsymbol{w}^{\nu_1}(t))_{0 \le t \le 1}$ is a $k$-dimensional $\nu_1$-Brownian motion and $d\boldsymbol{w}^{\nu_1}(t)[i]$, $i = 1, \ldots, k$ denotes the Itô stochastic integral. Then the probability measure $\nu_2$ has a density with respect to $\nu_1$. In particular, for any bounded functional $\Phi$ defined on $C[0,1]^k$,*

$$\mathbb{E}_{\nu_2}[\Phi(\boldsymbol{x})] = \mathbb{E}_{\nu_1}\left[\Phi(\boldsymbol{x})\left[\frac{\nu_2(\boldsymbol{x})}{\nu_1(\boldsymbol{x})}\right]_1\right].$$

If $\|\sigma(t)^{-1}(\boldsymbol{b}_\theta(\boldsymbol{x}(t), t) - \boldsymbol{b}_{\theta_{\text{base}}}(\boldsymbol{x}(t), t))\|$ is bounded, $([p_\theta(\boldsymbol{x})/p_{\theta_{\text{base}}}(\boldsymbol{x})]_t)_{0 \le t \le 1}$ is a martingale with respect to $p_{\theta_{\text{base}}}$. Consequently, Girsanov's Theorem tells us that the probability density of $p_\theta$ with respect to $p_{\theta_{\text{base}}}$ is given by

$$\frac{p_\theta(\boldsymbol{x})}{p_{\theta_{\text{base}}}(\boldsymbol{x})} := \exp\left\{\sum_{i=1}^{k} \int_0^1 u_\theta(\boldsymbol{x}(t), t)[i]d\boldsymbol{w}^{p_{\theta_{\text{base}}}}(t)[i] - \frac{1}{2}\int_0^1 \|u_\theta(\boldsymbol{x}(t), t)\|^2 dt\right\}, \tag{32}$$

$$u_\theta(\boldsymbol{x}(t), t) := \sigma(t)^{-1}(\boldsymbol{b}_\theta(\boldsymbol{x}(t), t) - \boldsymbol{b}_{\theta_{\text{base}}}(\boldsymbol{x}(t), t))$$

This expression for the density of $p_\theta$ with respect to $p_{\theta_{\text{base}}}$ allows us to compute the KL divergence between the probability measures $p_\theta$ and $p_{\theta_{\text{base}}}$ according to

$$\mathrm{D}_{\mathrm{KL}}\left(p_\theta \parallel p_{\theta_{\text{base}}}\right) = \frac{1}{2}\int_0^1 \mathbb{E}_{p_\theta}\|u_\theta(\boldsymbol{x}(t), t)\|^2 dt.$$

The stochastic integral term vanishes since it has expectation zero.

When $(\widehat{\boldsymbol{x}}(t))_{0 \le t \le 1}$ is sampled from the Euler-Maruyama approximation to $p_{\theta_{\text{base}}}$, we approximate (32) by replacing the integrals with

$$\int_0^1 u_\theta(\widehat{\boldsymbol{x}}(t), t)[i]d\boldsymbol{w}^{p_{\theta_{\text{base}}}}(t)[i] \approx T^{-1/2}\sum_{t=0}^{T-1} u_\theta(\widehat{\boldsymbol{x}}(t/T), t/T)[i](\boldsymbol{z}((t+1)/T) - \boldsymbol{z}(t/T))$$

$$\int_0^1 u_\theta(\widehat{\boldsymbol{x}}(t), t)^2[i]dt \approx T^{-1}\sum_{t=0}^{T-1} u_\theta(\widehat{\boldsymbol{x}}(t/T), t/T)^2[i]$$

where $\boldsymbol{z}(0), \ldots, \boldsymbol{z}((T-1)/T)$ are the same random variables from (30). This same approximation to the density ratio (32) can be derived by writing out the density ratio of $\widehat{p}_\theta(\widehat{\boldsymbol{x}}(0), \widehat{\boldsymbol{x}}(1/T), \ldots, \widehat{\boldsymbol{x}}(1))$ and $\widehat{p}_{\theta_{\text{base}}}(\widehat{\boldsymbol{x}}(0), \widehat{\boldsymbol{x}}(1/T), \ldots, \widehat{\boldsymbol{x}}(1))$, where $\widehat{p}_\theta$ is the probability distribution defined by the Euler-Maruyama discretization of $\widehat{p}_\theta$.

**Efficient gradient computation.** CGM-relax and CGM-reward require computing gradients of the density ratio $\frac{p_\theta(\boldsymbol{x})}{p_{\text{stop-grad}(\theta)}(\boldsymbol{x})}$. By applying Girsanov's Theorem to compute the density ratio, differentiating the result, and substituting in our approximations to the integrals, we obtain

$$\nabla_\theta \frac{p_\theta(\boldsymbol{x})}{p_{\text{stop-grad}(\theta)}(\boldsymbol{x})} = \sum_{i=1}^{k}\int_0^1 \nabla_\theta \sigma(t)^{-1}\boldsymbol{b}_\theta(\widehat{\boldsymbol{x}}(t), t)[i]d\boldsymbol{w}^{p_{\theta_{\text{base}}}}(t)[i]$$

$$\approx T^{-1/2}\sum_{i=1}^{k}\sum_{t=0}^{T-1}\nabla_\theta \sigma(t/T)^{-1}\boldsymbol{b}_\theta(\widehat{\boldsymbol{x}}(t/T), t/Y)[i](\boldsymbol{z}((t+1)/T)[i] - \boldsymbol{z}(t/T)[i])$$

$$= T^{-1/2}\sum_{t=0}^{T-1}\sigma(t/T)^{-1}\sum_{i=1}^{k}\nabla_\theta \boldsymbol{b}_\theta(\widehat{\boldsymbol{x}}(t/T), t/Y)[i](\boldsymbol{z}((t+1)/T)[i] - \boldsymbol{z}(t/T)[i]). \tag{33}$$

For high-dimensional diffusion models (e.g. Genie2 in our Section 4.1 experiments) memory constraints preclude the naive approach to computing equation (33) by instantiating each term in memory and simultaneously back-propagating gradients through all terms at once. However, because the gradient is a sum across time, it can computed in chunks. In practice, we divide $\{0, \ldots, T\}$ into $\lceil T/\texttt{chunk\_size}\rceil$ blocks of approximately equal size, where $\texttt{chunk\_size}$ is the largest chunk size that can fit into memory.

**Solution to the maximum entropy problem.** When the base model $p_{\theta_{\text{base}}}(\boldsymbol{x})$ constitutes a continuous-time diffusion model (29) satisfying certain regularity properties, and the constraint

function $\boldsymbol{h}$ depends only on the path at time $t = 1$, there exists a closed-form solution to the maximum entropy problem (4).

Let $p$ be the law of an SDE having diffusion coefficient $\sigma$ and initial distribution $p'_{\text{init}}(\boldsymbol{x}(0))$; this is necessary for $p \ll p_{\theta_{\text{base}}}$ by Girsanov's Theorem (Theorem D.1). By the chain rule for the KL divergence, the objective for the maximum entropy problem defined on the full path measures is

$$\mathrm{D_{KL}}\left(p(\boldsymbol{x}) \,\|\, p_{\theta_{\text{base}}}(\boldsymbol{x})\right)$$
$$=\mathrm{D_{KL}}\left(p(\boldsymbol{x}(1)) \,\|\, p_{\theta_{\text{base}}}(\boldsymbol{x}(1))\right) + \mathbb{E}_{p_\theta(\boldsymbol{x}(0))}[\mathrm{D_{KL}}\left(p(\cdot|\boldsymbol{x}(0)) \,\|\, p_{\theta_{\text{base}}}(\cdot|\boldsymbol{x}(0))\right)].$$

The KL divergence is computed according to Girsanov's Theorem.

From here, by the maximum entropy principle applied to the marginal at time $t = 1$, the first term in the objective is lower bounded by

$$\mathrm{D_{KL}}\left(p(\boldsymbol{x}(1)) \,\|\, p_{\theta_{\text{base}}}(\boldsymbol{x}(1))\right) \geq \mathrm{D_{KL}}\left(p_{\boldsymbol{\alpha}_0^*}(\boldsymbol{x}(1)) \,\|\, p_{\theta_{\text{base}}}(\boldsymbol{x}(1))\right)$$

where $p_{\boldsymbol{\alpha}_0^*}(\boldsymbol{x}(1))$ is the solution to the maximum entropy problem in $k$-dimensional Euclidean space. Consequently, if we can show that there exists an SDE $p(\boldsymbol{x})$ satifying $p(\boldsymbol{x}(1)) = p_{\boldsymbol{\alpha}_0^*}(\boldsymbol{x}(1))$ and $p(\cdot|\boldsymbol{x}(1)) = p_{\theta_{\text{base}}}(\cdot|\boldsymbol{x}(1))$, then $p$ is the solution to the maximum entropy problem. This is the subject of the following result:

**Proposition D.2** (Maximum entropy solution for a diffusion model). *Suppose that the constraint function $\boldsymbol{h}$ depends only on the value of the path at time $t = 1$ and is bounded and continuous. Moreover, assume that $\boldsymbol{x}(0) \perp \boldsymbol{x}(1)$ under the base model $p_{\theta_{base}}(\boldsymbol{x})$.*

*Then the solution to the maximum entropy problem is a diffusion process*

$$p^* : d\boldsymbol{x}(t) = \{\boldsymbol{b}_{\theta_{base}}(\boldsymbol{x}(t), t) + \sigma(t)\boldsymbol{u}^*(\boldsymbol{x}(t), t)\}dt + \sigma(t)d\boldsymbol{w}(t)$$

*satisfying $p^*(\boldsymbol{x}(0)) = p_{init}$. The drift $\boldsymbol{u}^*(\boldsymbol{x}(t), t)$ admits the Feynman-Kac characterization*

$$u^*(\boldsymbol{x}, t) = \sigma(t)\nabla_{\boldsymbol{x}} \log \mathbb{E}_{p_{\theta_{base}}}[\exp\{r_{\boldsymbol{\alpha}_0^*}(\boldsymbol{x}(1))\}|\boldsymbol{x}(t) = \boldsymbol{x}],$$

*where $\boldsymbol{\alpha}_0^*$ are the parameters corresponding to the maximum entropy solution in $k$-dimensional Euclidean space (4) with base distribution $p_{\theta_{base}}(\boldsymbol{x}(1))$ and constraint function $\boldsymbol{h}(\boldsymbol{x})$.*

*Finally, $p^*(\boldsymbol{x})$ satisfies $p^*(\cdot|\boldsymbol{x}(1)) = p_{\theta_{base}}(\cdot|\boldsymbol{x}(1))$ and $p^*(\boldsymbol{x}(1)) = p_{\boldsymbol{\alpha}_0^*}(\boldsymbol{x}(1))$.*

We refer the reader to Domingo-Enrich et al. (2025, Theorem 1) for a proof. The result is a consequence of standard results in the theory of diffusion processes, specifically the Doob $h$-transform (Oksendal, 2013, Chapter 7).

The assumption that, under $p_{\theta_{\text{base}}}(\boldsymbol{x})$, the path at time $t = 0$ is independent of the path at time $t = 1$ is necessary to ensure that $p^*(\boldsymbol{x}(0))$ can be chosen to be equal to $p_{\text{init}}(\boldsymbol{x}(0))$. This is desirable because, by design, $p_{\text{init}}$ is a distribution from which sampling is tractable. However, when the independence assumption does not hold, $p^*(\boldsymbol{x}(0))$ cannot be chosen to be equal to $p_{\theta_{\text{base}}}(\boldsymbol{x}(0))$ (Denker et al., 2024, Appendix G.2).

Although, at first glance, this independence assumption may appear strong, Domingo-Enrich et al. (2025, Theorem 1) proves that diffusion models whose initial distribution is Gaussian noise and whose terminal distribution is the data distribution satisfies this property (i.e., variance-preserving SDEs). One example of a commonly used noise schedule satisfying this property is $\sigma(t) = t^{-1}$, which is singular at time $t = 0$.

Proposition D.2 tells us that when we fine-tune a diffusion model with CGM to satisfy a constraint on its terminal distribution $\mathbb{E}_{p_{\theta_{\text{base}}}(\boldsymbol{x}(1))}[\boldsymbol{h}(\boldsymbol{x})] = \boldsymbol{h}^*$, we expect the terminal distribution of the base model to change to satisfy the constraint, while the conditional path distribution given the endpoint should be preserved. In other words, seeking the distribution over paths that is closest in KL distance to the base model amounts to shifting the terminal distribution while leaving the conditional distributions unchanged.

## D.2 INITIALIZING THE BASE DIFFUSION MODEL TO SAMPLE A GAUSSIAN MIXTURE

In each of our synthetic data experiments, we initialize our base diffusion model $p_{\theta_{\text{base}}}$ such that $p_{\theta_{\text{base}}}(\boldsymbol{x}(1))$ is equal to a GMM. We achieve this by representing $p_{\theta_{\text{base}}}$ as the reversal of a forward

diffusion process. A forward diffusion process draws samples from the target GMM density $\boldsymbol{x}(1) \sim p_{\text{target}}$ and then noises them according to the linear SDE

$$\overrightarrow{p} : d\boldsymbol{x}(t) = \frac{1}{2}\kappa(t)\boldsymbol{x}(t)dt + \sigma(t)d\boldsymbol{w}(t), \ 0 \leq t \leq 1. \tag{34}$$

When the diffusion coefficient is chosen such that $\sigma(t) = \sqrt{\kappa(t)}$ and the linear coefficient $(\kappa(t))_{0 \leq t \leq 1}$ satisfies $\kappa(t) \geq 0$, $\int_0^1 \kappa(t)dt = +\infty$, then $\overrightarrow{p}(\boldsymbol{x}(0)) \stackrel{d}{=} \mathcal{N}(\boldsymbol{0}, \mathbb{I})$. We choose $\kappa(t) = t^{-1}$. Simply, (34) turns samples from $p_{\text{target}}$ into Gaussian noise. In practice, since the drift and diffusion coefficients defined by $\kappa(t)$ are unbounded (which violates the assumptions for existence and uniqueness of the solution to the SDE from Appendix D.1), we cap $\kappa(t)$ at some large $M$.

A foundational result in diffusion processes (Anderson, 1982) states that the reversal of (34) is another diffusion process that is given by

$$\overleftarrow{p} : d\boldsymbol{x}(t) = \left\{\sigma(t)^2 \nabla_{\boldsymbol{x}} \log \overrightarrow{p}(\boldsymbol{x}(t)) + \frac{1}{2}\kappa(t)\boldsymbol{x}(t)\right\} dt + \sigma(t)d\boldsymbol{w}(t), \ 0 \leq t \leq 1 \tag{35}$$

with $\overleftarrow{p}(\boldsymbol{x}(0)) \stackrel{d}{=} \overrightarrow{p}(\boldsymbol{x}(0))$. The probability distributions defined by (34) and (35) are equal in law. $\nabla_{\boldsymbol{x}} \log \overrightarrow{p}(\boldsymbol{x}(t))$ is called the *score* of the forward process (34).

Equation (35) is useful since it tells how to generate samples from $p_{\text{target}}$: first draw samples from $\overrightarrow{p}(\boldsymbol{x}(0)) \approx \mathcal{N}(\boldsymbol{x}(0) \mid \boldsymbol{0}, \mathbb{I})$, then solve the SDE (35) numerically using Euler-Maruyama, for example. However, for general target distributions $p_{\text{target}}$, the score of the forward process is intractable, which yields the backward diffusion process (35) also intractable.

In the case of a GMM, though, the score of the forward process is tractable. Indeed, for $p_{\text{target}}(\boldsymbol{x}(1)) = \sum \pi_i \mathcal{N}(\boldsymbol{x}(1) \mid \boldsymbol{\mu}_i, \boldsymbol{\Sigma}_i)$, we compute

$$\overrightarrow{p}(\boldsymbol{x}(t)) = \int \overrightarrow{p}(\boldsymbol{x}(t)|\boldsymbol{x}(1))p_{\text{target}}(\boldsymbol{x}(1))d\boldsymbol{x}(1)$$

$$= \sum \pi_i \int \mathcal{N}(\boldsymbol{x}(t)|m(t)\boldsymbol{x}(1), s(t)^2\mathbb{I})\mathcal{N}(\boldsymbol{x}(1) \mid \boldsymbol{\mu}_i, \boldsymbol{\Sigma}_i)d\boldsymbol{x}(1)$$

$$= \sum \pi_i \mathcal{N}(\boldsymbol{x}(t)|m(t)\boldsymbol{\mu}_i, s(t)^2\mathbb{I} + m(t)^2\boldsymbol{\Sigma}_i),$$

where $m(t)$ and $s(t)$ are defined by the forward diffusion process $(\kappa(t))_{0 \leq t \leq 1}$. For $\kappa(t) = t^{-1}$, we have $m(t) = t^{1/2}$ and $s(t) = (1-t)^{1/2}$. Using this expression for $\overrightarrow{p}(\boldsymbol{x}(t))$, we initialize $p_{\theta_{\text{base}}}(\boldsymbol{x})$ to the exact reversal of the forward process (34) according to (35).

### D.3 EXPERIMENTAL DETAILS

We perform all synthetic data experiments using Adam (Kingma & Ba, 2015) with default momentum hyperparameters $\beta = (0.9, 0.999)$ and a cosine decay learning rate schedule (Loshchilov & Hutter, 2016). We perform $2 \times 10^3$ CGM iterations for every experiment. We train on a single H100 GPU.

In the diffusion generative model, we parameterize the drift function $\boldsymbol{b}_\theta$ as

$$\boldsymbol{b}_\theta(\boldsymbol{x}(t), t) = \sigma(t)^2\{\nabla_{\boldsymbol{x}} \log \overrightarrow{p}(\boldsymbol{x}(t)) - u_\theta(\boldsymbol{x}, t)\} + \frac{1}{2}\kappa(t)\boldsymbol{x}(t).$$

$u_\theta$ is a neural network with two hidden layers of dimension 256 and SiLU activations, and $\log \overrightarrow{p}(\boldsymbol{x}(t))$ is the analytical score of the forward process that we described in Appendix D.2. In addition to $\boldsymbol{x}(t)$, we feed as input to $u_\theta$ a sinusoidal time embedding of dimension 32. By initializing the weights of the output layer of $u_\theta$ to zero, we ensure that $p_\theta$ is initialized at $p_{\theta_{\text{base}}}$, the reversal of the forward diffusion process $\overrightarrow{p}$.

All synthetic data experiments are performed with batch size $M = 10^4$. For CGM-relax, we select $\lambda$ by first performing calibration for each $\lambda$ on a log linear grid from $10^0$ to $10^{-3}$ with 10 grid points. We choose the value of $\lambda$ for which $(\mathcal{L}^{\text{viol}})^{1/2}$ is reduced by a factor of 10 and $\mathcal{L}^{\text{KL}}$ is the smallest. If no such value exists, we choose $\lambda$ for which $\mathcal{L}^{\text{KL}}$ is smallest. For CGM-reward, we compute $\widehat{\boldsymbol{\alpha}}_N$ using $N = 10^5$ samples from $p_{\theta_{\text{base}}}$.

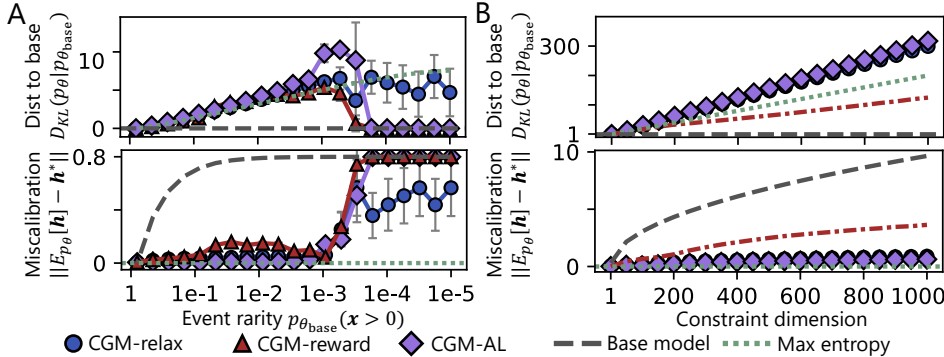

Figure 6: Comparison of CGM-relax and CGM-reward to the augmented Lagrangian algorithm ('CGM-AL') for calibrating a diffusion model that generates samples from a GMM. For CGM-relax and CGM-reward, we report the same values as in Figure 2. For CGM-AL, we report results with $\ell = 20$ in panel A and with $\ell = 100$ in panel B (best values). We find that CGM-AL performs comparably to CGM-relax, except in the rare event setting where it performs worse (panel A). **A**: Calibrating the mixture proportion of a rare mode in a 1D GMM. **B**: Calibrating a $k$-dimensional GMM to a $k$-dimensional constraint.

### D.4 COMPARISON TO AUGMENTED LAGRANGIAN METHOD

As an alternative to CGM-relax, we consider a variant of the augmented Lagrangian (AL) algorithm (Hestenes, 1969), which solves a general constrained optimization problem.

In particular, suppose one would like to solve

$$\min_{\boldsymbol{z} \in \mathbb{R}^p} f(\boldsymbol{z}), \quad \text{subject to } \boldsymbol{c}(\boldsymbol{z}) = \boldsymbol{c}^*,$$

where $f : \mathbb{R}^p \to \mathbb{R}$ is the objective function and $\boldsymbol{c} : \mathbb{R}^p \to \mathbb{R}^d$ is the constraint function. The AL method forms the "augmented Lagrangian" $\mathcal{L}_\lambda : \mathbb{R}^p \times \mathbb{R}^d \to \mathbb{R}$ with penalty parameter $\lambda$, defined

$$\mathcal{L}_\lambda(\boldsymbol{z}, \boldsymbol{u}) = f(\boldsymbol{z}) + \boldsymbol{u}^\top (\boldsymbol{c}(\boldsymbol{z}) - \boldsymbol{c}^*) + \frac{\lambda}{2} \|\boldsymbol{c}(\boldsymbol{z}) - \boldsymbol{c}^*\|^2. \tag{36}$$

$\mathcal{L}_\lambda$ differs from the ordinary Lagrangian due the presence of the penalty term $\frac{\lambda}{2}\|\boldsymbol{c}(\boldsymbol{z}) - \boldsymbol{c}^*\|^2$. $\boldsymbol{u} \in \mathbb{R}^d$ are the dual variables. The AL algorithm alternates between minimizing $\mathcal{L}_\lambda$ with respect to $\boldsymbol{z}$ and updating the dual variables:

$$\begin{aligned}
\boldsymbol{z}^{(k)} &\leftarrow \arg\min_{\boldsymbol{z} \in \mathbb{R}^p} \mathcal{L}_\lambda(\boldsymbol{z}, \boldsymbol{u}^{(k)}) \\
\boldsymbol{u}^{(k+1)} &\leftarrow \boldsymbol{u}^{(k)} + \lambda(\boldsymbol{c}(\boldsymbol{z}^{(k)}) - \boldsymbol{c}^*).
\end{aligned} \tag{37}$$

Some variants of the AL method also update the penalty $\lambda$ according to some schedule. The augmented Lagrangian algorithm can be viewed as a proximal-point algorithm applied to the dual function (Rockafellar, 1976)

$$\boldsymbol{u}^{(k+1)} \leftarrow \arg\max_{\boldsymbol{u} \in \mathbb{R}^d} \left\{ d(\boldsymbol{u}) - \frac{1}{2\lambda} \|\boldsymbol{u} - \boldsymbol{u}^{(k)}\| \right\}, \; d(\boldsymbol{u}) = \min_{\boldsymbol{z} \in \mathbb{R}^p} f(\boldsymbol{z}) + \boldsymbol{u}^\top (\boldsymbol{c}(\boldsymbol{z}) - \boldsymbol{c}^*).$$

In the setting of the calibration problem (1), we identify $\boldsymbol{z} = \theta$, $f(\theta) = \mathrm{D}_{\mathrm{KL}}(p_\theta \| p_{\theta_{\mathrm{base}}})$, $c(\theta) = \mathbb{E}_{p_\theta}[\boldsymbol{h}(\boldsymbol{x})]$, $\boldsymbol{c}^* = \boldsymbol{h}^*$. However, the augmented Lagrangian does not admit a closed-form minimizer with respect to the model parameters $\theta$. Consequently, the AL algorithm as stated in equation (37) cannot be applied. Instead, we propose alternating between performing a stochastic gradient update to the model parameters $\theta$ and updating the Lagrange multipliers $\boldsymbol{u}$. This introduces an additional algorithmic hyperparameter $\ell$ representing how frequently (i.e., after how many iterations) the dual variables are updated. We provide pseudocode for our implementation in Algorithm 3, which we refer to as 'CGM-AL'.

We compare CGM-AL against CGM-relax and CGM-reward on the two synthetic experiments described in Section 3 and Appendix D.3. For CGM-AL, we select $\lambda$ by performing a 10-point grid

---

**Algorithm 3** CGM-AL fine-tuning (augmented Lagrangian)

---

**Require:** $p_{\theta_{\text{base}}}$, $\boldsymbol{h}(\cdot)$, $\boldsymbol{h}^*$, $M$, $\lambda$, $\ell$

$\quad \triangleright$ Initialize model and dual variables
$\quad p_\theta \leftarrow p_{\theta_{\text{base}}}$, $\boldsymbol{u} \leftarrow \boldsymbol{0}$, $t \leftarrow 0$
$\quad$ **while** not converged **do**

$\quad\quad \triangleright$ Sample and compute importance weights
$\quad\quad \boldsymbol{x}_{1:M} \overset{i.i.d.}{\sim} p_{\text{stop-grad}(\theta)}$
$\quad\quad w_m \leftarrow p_\theta(\boldsymbol{x}_m)/p_{\text{stop-grad}(\theta)}(\boldsymbol{x}_m)$

$\quad\quad \triangleright$ KL loss with LOO baseline
$\quad\quad l_m \leftarrow \log\big(p_{\text{stop-grad}(\theta)}(\boldsymbol{x}_m)/p_{\theta_{\text{base}}}(\boldsymbol{x}_m)\big)$
$\quad\quad l_m^{\text{LOO}} \leftarrow l_m - \frac{1}{M-1}\sum_{m'\neq m} l_{m'}$
$\quad\quad \widehat{\mathcal{L}}^{\text{KL}} \leftarrow \frac{1}{M}\sum_{m=1}^{M} w_m l_m^{\text{LOO}}$

$\quad\quad \triangleright$ Constraint violation loss
$\quad\quad \boldsymbol{h}_m \leftarrow w_m(\boldsymbol{h}(\boldsymbol{x}_m) - \boldsymbol{h}^*)$
$\quad\quad \widehat{\Delta\boldsymbol{h}} \leftarrow \frac{1}{M}\sum_{m=1}^{M} \boldsymbol{h}_m$
$\quad\quad \widehat{\mathcal{L}}^{\text{viol}} \leftarrow \|\widehat{\Delta\boldsymbol{h}}\|^2 - \frac{1}{M}\widehat{\text{Var}}[\boldsymbol{h}_{1:M}]$,
$\quad\quad \widehat{\text{Var}}[\boldsymbol{h}_{1:M}] = \frac{1}{M-1}\sum \|\boldsymbol{h}_m - \widehat{\Delta\boldsymbol{h}}\|^2$

$\quad\quad \triangleright$ Augmented Lagrangian loss
$\quad\quad \widehat{\mathcal{L}}^{\text{AL}} \leftarrow \widehat{\mathcal{L}}^{\text{KL}} + \boldsymbol{u}^\top \widehat{\Delta\boldsymbol{h}} + \frac{\lambda}{2}\widehat{\mathcal{L}}^{\text{viol}}$

$\quad\quad \triangleright$ Primal update
$\quad\quad \theta \leftarrow \text{gradient-step}\big(\theta, \nabla_\theta \widehat{\mathcal{L}}^{\text{AL}}\big)$

$\quad\quad \triangleright$ Dual update every $k$ iterations
$\quad\quad$ **if** $t \bmod \ell = 0$ **then**
$\quad\quad\quad \boldsymbol{u} \leftarrow \boldsymbol{u} + \lambda\,\text{stop-grad}(\widehat{\Delta\boldsymbol{h}})$
$\quad\quad t \leftarrow t+1$

---

search on $[10^0, 10^2]$, and we consider two potential values for the Lagrange multiplier update frequency, $\ell \in \{20, 100\}$. We again perform stochastic gradient updates to $\theta$ using Adam and a cosine decay learning rate schedule with initial learning rate $10^{-4}$ and final learning rate $10^{-7}$.

From Figure 6, we observe that CGM-relax performs comparably to CGM-AL and, in the rare event mode reweighting example (Figure 6A), CGM-relax outperforms CGM-AL. While one might think that poor conditioning of the relax loss landscape for small $\lambda$ would result in inferior performance of CGM-relax to CGM-AL, we observe this is not the case. We attribute this to our choice of $\lambda$ via grid search: as described in Appendix D.3, we select $\lambda$ to balance between constraint violation and KL distance to the base model $p_{\theta_{\text{base}}}$. Since the AL method introduces nontrivial computational overhead in the choice of the dual parameter update frequency $\ell$, we prefer CGM-relax.

# E   CASE STUDY ADDITIONAL DETAILS

In this section, we describe the experimental setup for our case studies with CGM-relax and CGM-reward from Section 4. We provide explanations regarding the generative model classes $p_\theta$, CGM constraint functions $\boldsymbol{h}$ and targets $\boldsymbol{h}^*$, choice of CGM hyperparameters $\lambda$ and $N$, model architectures, and training procedures. We also include additional samples from our models before and after calibration.

Just as in our synthetic data experiments, we perform all experiments using Adam with default momentum hyperparameters $\beta = (0.9, 0.999)$ and a cosine decay learning rate schedule. Additional common training details are shown in Table 1. We train all models on a single H100 GPU.

Table 1: Training configurations for experiments. Batch (sub-batch) indicates the number of samples per batch and the sub-batch size used to fit gradient computations into memory. $\mathcal{S}_V^L$ denotes the set of all sequences with vocabulary size $V$ and length $L$.

| Hyperparameter | Genie2 | ESM3-open | TarFlow | TinyStories-33M |
|---|---|---|---|---|
| Initial learning rate | $10^{-5}$ | $10^{-4}$ | $10^{-6}$ | $2 \times 10^{-6}$ |
| Batch (sub-batch) | 64 (16) | 256 (64) | 256 (16) | 512 (64) |
| Training steps | 100 | 100 | 50 | 200 |
| $x$ Space | $(\mathbb{R}^{100 \times 3})^{100}$ | $(\mathcal{S}_{4096}^{100})^{50}$ | $\mathbb{R}^{256 \times 256 \times 3}$ | $\mathcal{S}_{10,000}^{200}$ |
| Constraint dims ($k$) | 99 | 99 | 5 | 8 |
| Model parameters | 15M | 1.4B | 463M | 33M |
| Training time (hrs) | 48 | 2.3 | 3 | 0.1 |

## E.1 Calibrating Genie2

For our experiments with Genie2, we represent $p_\theta$ as a continuous-time diffusion model defined over three-dimensional protein backbone coordinates with drift function defined by the SE(3)-equivariant encoder-decoder architecture from Lin et al. (2024a).

Since Genie2 is trained as the reversal of a discrete-time noising process (a DDPM, see Ho et al., 2020), we first convert the discrete-time denoising diffusion model to a (continuous-time) diffusion model. We achieve this by redefining the final timestep $T$ of the original denoising process to be time 1 of the continuous-time process. To define the drift function, we take the DDPM transition mean defined at each time $t$ in the discrete-time process, divide it by $1/T = T$, and define the drift function to be equal to the resulting value in between times $t/T$ and $(t+1)/T$. The diffusion coefficient is similarly defined by the DDPM transition standard deviation at each time $t$ in the discrete-time process, but is instead scaled by $T^{1/2}$. This approach of converting the DDPM into a continuous-time diffusion model ensures that when the SDE is solved under the Euler-Maruyama scheme using a grid of $T$ timesteps (i.e., the original time grid used to define the DDPM), one samples from the original DDPM.

We perform sampling using $10^2$ timesteps and a non-uniform time grid: we sample the first 50 steps on the interval $[0, 0.05]$ and the remaining steps on the interval $[0.05, 1]$. We point out that the original Genie2 model was trained with $10^3$ denoising steps; we find that reducing the number of sampling steps dramatically decreases the runtime of CGM calibration. Our sampling scheme is possible since we redefined the base generative model to be a continuous-time diffusion process. We computed self-consistency metrics for the base Genie2 model sampled on the original time grid (with $10^3$ steps) and on our proposed grid (with $10^2$ steps), we did not observe any difference in sample quality.

For CGM-relax, we calibrate to $k$=99 constraints on the bivariate CDF of alpha helix and beta strand content. And for CGM-reward, we calibrate to $k$=15 constraints using $N = 2.5 \times 10^4$ samples from $p_{\theta_{\text{base}}}$. Since sampling from the Genie2 base model is time intensive, the sampling cost to compute $\widehat{\boldsymbol{\alpha}}_N$ (with small variance) is a downside to CGM-reward. Results reported in Figure 3 are averages over 3 trials, with two standard errors.

**Self-consistency RMSD and design failures.** To assess the quality of our generations, we compute the root mean-square deviation (RMSD) between $C_\alpha$ atoms resulting from (i) unfolding our generated structures into predicted amino sequences, (ii) refolding each of these predicted sequences into a protein structure, and (iii) aligning the predicted structures to the original structure. The self-consistency RMSD (scRMSD) is defined as the smallest RMSD between the given structure and one of the corresponding predictions. We use ProteinMPNN (Dauparas et al., 2022) for our inverse folding model and ESMFold (Lin et al., 2023) for our folding model; we compute scRMSD from 8 sequences. The pipeline we employ was developed by Lin et al. (2024b). Once we have determined the scRMSD of a generated structure, we classify it as a "design failure" if its scRMSD is greater than 2Å. Intuitively, designability is a binary measure of whether or not a structure could have been plausibly produced by folding an amino acid sequence.

**Secondary structure annotation.** As discussed in Section 4.1, we measure the diversity of a collection of protein structures by computing the proportion of residues that lie in each of the three protein secondary structure types. For the CATH domains and Genie2, we perform annotations using the Biotite package (Kunzmann & Hamacher, 2018), which considers only $C_\alpha$ backbone atoms. For the CATH proteins (Sillitoe et al., 2021), we obtain the secondary structure distribution by annotating the domains collected and published by Ingraham et al. (2019).

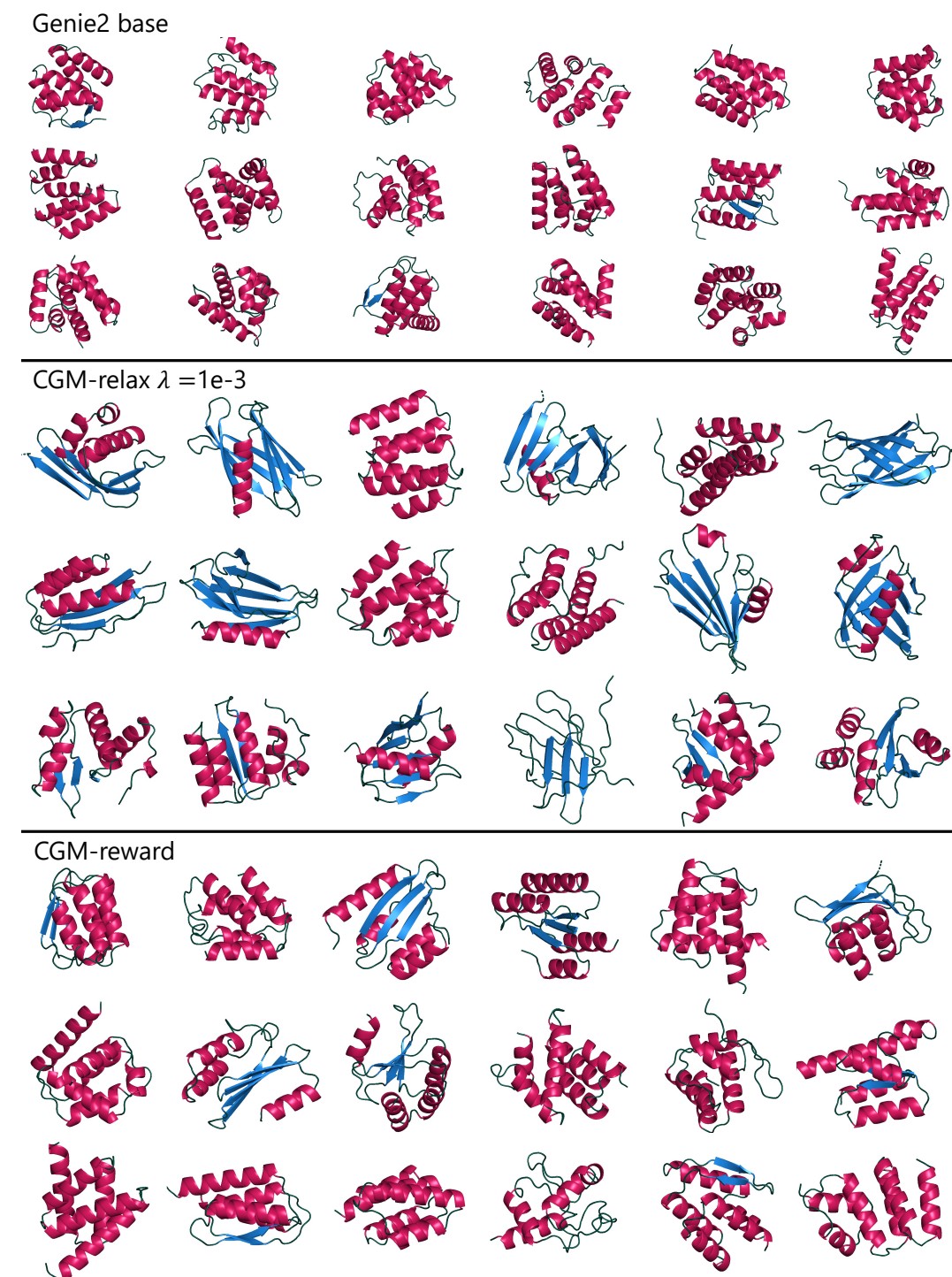

Figure 7: Random samples from the Genie2 model before calibration (top), after calibration using CGM-relax with 99 bivariate CDF constraints (middle), and after calibration using CGM-reward with 15 bivariate CDF constraints (bottom).

## E.2 Calibrating ESM3-open

In order to apply CGM to ESM3, we need to be able to sample from the model and to compute gradients of sample log-probabilities with respect to the model's parameters.

**Sampling method.** Following the method used by Hayes et al. (2025), sampling is achieved by treating the model as a discrete time Markov chain that starts at a sequence of mask tokens and ends at a sequence of fully unmasked structure tokens. Each step $i$ of the chain consists of three steps and transitions from state $\boldsymbol{x}(i-1)$ to $\boldsymbol{x}(i)$.

1. Pick token indices $U(i)$ to unmask uniformly at random without replacement from the masked tokens of $\boldsymbol{x}(i-1)$.

2. Use the model $\pi_\theta$ to predict a categorical distribution $\pi_\theta^{(j)}(\cdot \mid \boldsymbol{x}(i-1))$ for $j \in U(i)$ for each of the newly unmasked tokens given the previous partially masked state.

3. Sample the values of those tokens from the predicted categorical distributions, resulting in $|U(i)|$ more unmasked tokens.

As implemented by Hayes et al. (2025), we use $T = 50$ steps to sample 100-residue sequences and follow a cosine unmasking schedule. The cosine schedule determines the number of masked positions at each sampling step as

$$r(i) := \mathrm{round}\left(100 \times \cos\left(\frac{\pi}{2}\frac{i}{T}\right)\right), \quad i = 0, \ldots, T.$$

Early sampling steps unmask few tokens per step, while later ones sample many at once. Intuitively, this let's the model sample more tokens in parallel once it has more information to predict the final sequence. Note that the number of tokens unmasked at step $i > 0$ is $|U(i)| = r(i-1) - r(i)$.

**Transition probabilities.** A Markov chain can be characterized by its initial state distribution, $\boldsymbol{x}(0) \sim \pi_0(\boldsymbol{x}(0))$ and its transition probabilities for going from one state to the next. The ESM3 sampling method starts fully masked, so has initial distribution $\pi_0(\boldsymbol{x}(0)) = \mathbf{1}\{\boldsymbol{x}(0) \text{ is fully masked}\}$. The transition probabilities follow from the sampling procedure and are

$$p_\theta(\boldsymbol{x}(i) \mid \boldsymbol{x}(i-1)) = C(i) \prod_{j \in U(i)} \pi_\theta^{(j)}(\boldsymbol{x}(i)[j] \mid \boldsymbol{x}(i-1)), \tag{38}$$

where $C(i)$ is a constant that accounts for randomly choosing which tokens to unmask. $C(i)$ does not depend on $\theta$ or the sampling trajectory $(\boldsymbol{x}(0), \boldsymbol{x}(1), \ldots, \boldsymbol{x}(T))$, since every unmasking order is equally likely. Note $U(i)$ can be computed from $\boldsymbol{x}(i-1)$ and $\boldsymbol{x}(i)$ by finding which tokens are masked in $\boldsymbol{x}(i-1)$ and not in $\boldsymbol{x}(i)$.

**Trajectory log-probability.** As in the neural SDE setting (Appendix D.1), the marginal likelihood of $\boldsymbol{x}_T$ is intractable, so we treat samples $\boldsymbol{x}$ as entire trajectories, $\boldsymbol{x} = (\boldsymbol{x}(0), \boldsymbol{x}(1), \boldsymbol{x}(2), \ldots, \boldsymbol{x}(T))$. Using the Markov property, the log-probability of a trajectory is

$$\log p_\theta(\boldsymbol{x}) = \log\left(\pi_0(\boldsymbol{x}_0) \prod_{i=1}^{T} p_\theta(\boldsymbol{x}(i) \mid \boldsymbol{x}(i-1))\right)$$

$$= \log \pi_0(\boldsymbol{x}_0) + \sum_{i=1}^{T} \log\left(C(i) \prod_{j \in U(i)} \pi_\theta^{(j)}(\boldsymbol{x}(i)[j] \mid \boldsymbol{x}(i-1))\right) \quad \text{(by equation (38))}$$

$$= \sum_{i=1}^{T} \log\left(C(i) \prod_{j \in U(i)} \pi_\theta^{(j)}(\boldsymbol{x}(i)[j] \mid \boldsymbol{x}(i-1))\right) \quad (\pi_0(\boldsymbol{x}(0)) = 1 \text{ by construction})$$

$$= \sum_{i=1}^{T} C(i) + \sum_{i=1}^{T} \sum_{j \in U(i)} \log \pi_\theta^{(j)}(\boldsymbol{x}(i)[j] \mid \boldsymbol{x}(i-1)).$$

**Parameter gradients.** Now that we have defined the log-probability of $\boldsymbol{x}$, we can compute gradients with respect to $\theta$ as

$$\nabla_\theta \log p_\theta(\boldsymbol{x}) = \nabla_\theta \left( \sum_{i=1}^{T} C(i) + \sum_{i=1}^{T} \sum_{j \in U(i)} \log \pi_\theta^{(j)}(\boldsymbol{x}(i)[j] \mid \boldsymbol{x}(i-1)) \right)$$

$$= \sum_{i=1}^{T} \sum_{j \in U(i)} \nabla_\theta \log \pi_\theta^{(j)}(\boldsymbol{x}(i)[j] \mid \boldsymbol{x}(i-1)),$$

which conveniently is a sum over sampling steps. The decomposition of the gradient into a sum over sampling steps lets us compute parameter gradients using constant memory with respect to the number of sampling steps, which is critical for high-parameter-count models such as ESM3-open.

**Secondary structure annotation.** We use the ESM3 structure decoder and the ESM3 function `ProteinChain.infer_oxygen` to get heavy atom coordinates from sampled structure tokens. We then pass the coordinates to the Python package `PyDSSP` (Minami, 2023) to annotate secondary structure.

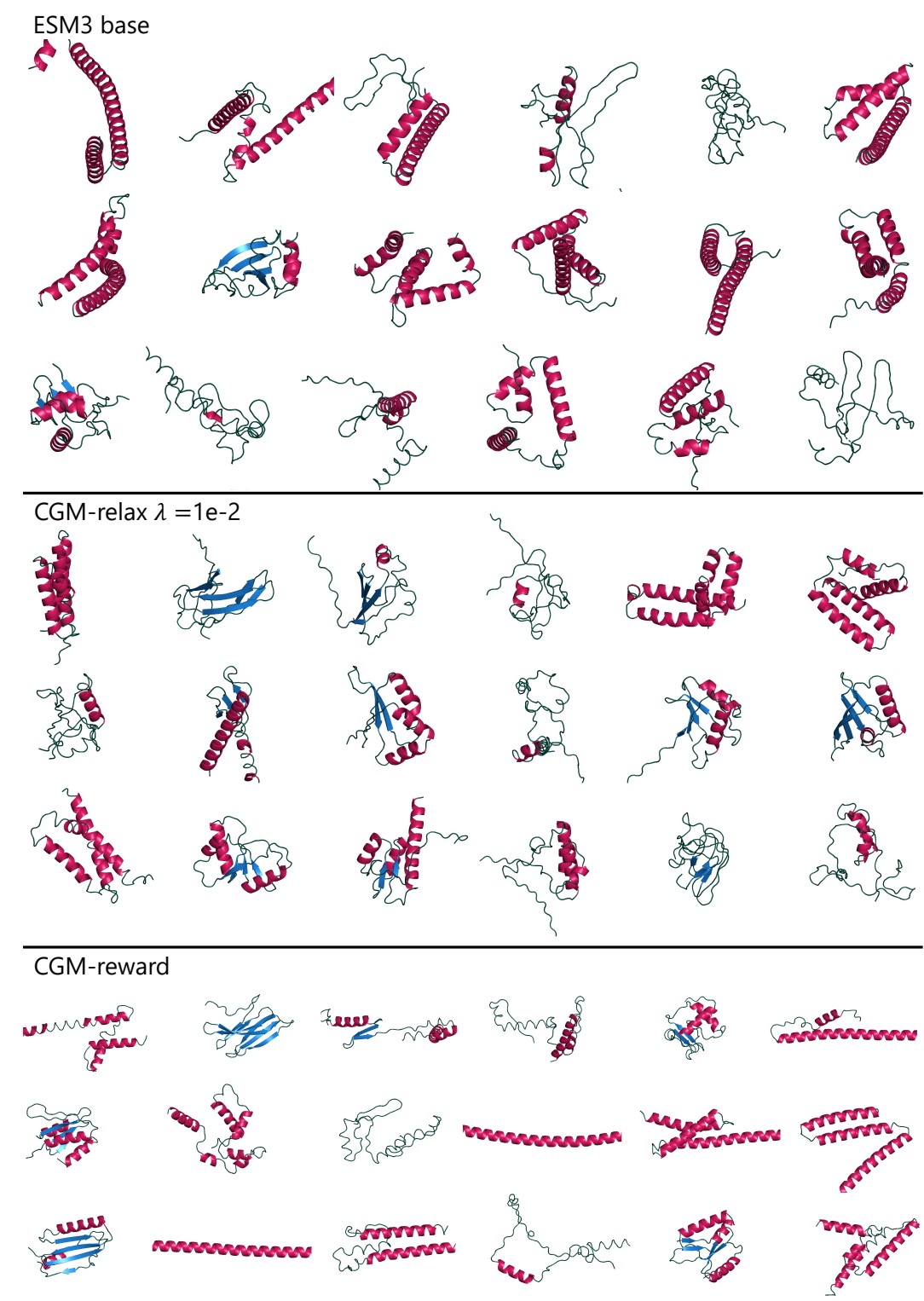

Figure 8: Random samples from the ESM3-open model before calibration (top), after calibration using CGM-relax with 99 bivariate CDF constraints (middle), and after calibration using CGM-reward with 15 bivariate CDF constraints (bottom).

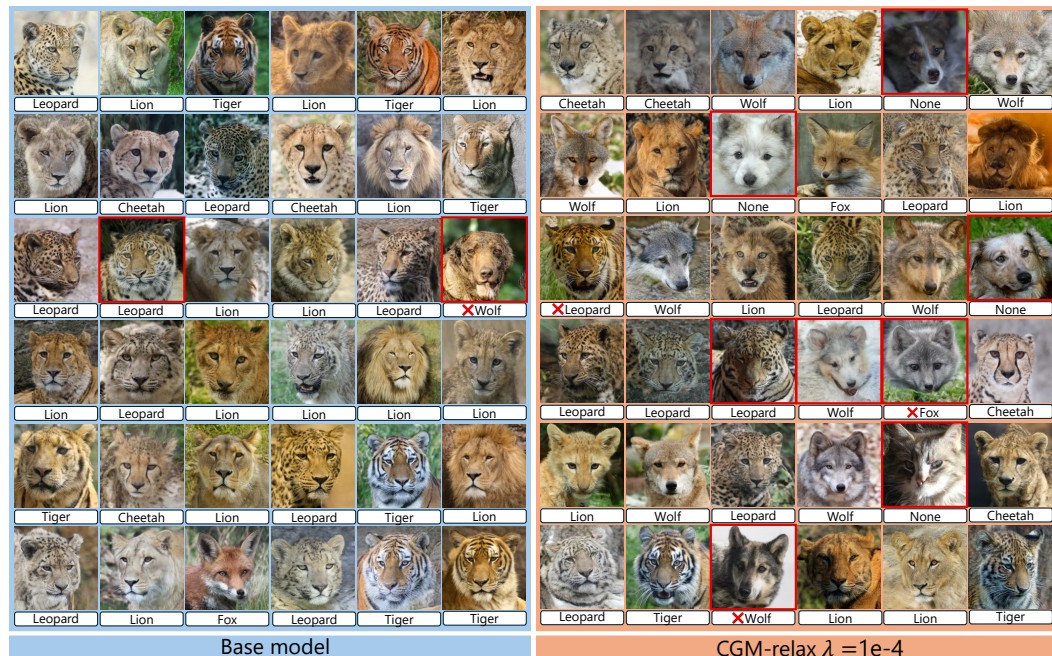

Figure 9: Random samples from the conditional TarFlow model trained on the AFHQ dataset (blue background) and the same model fine-tuned using CGM-relax ($\lambda=10^{-4}$) (orange background), with annotations by GPT o5-mini (white box). Red boxes denote poor-quality samples, and red crosses denote incorrect annotations. Although the model calibrated with CGM-relax produces animals with more balanced class proportions, it also produces fewer realistic samples.

### E.3 CALIBRATING TARFLOW

As we described in Section 4.2, our goal when calibrating the TarFlow model (Zhai et al., 2025), trained conditionally on the Animal Faces HQ (AFHQ) dataset (Choi et al., 2020), is to generate more diverse samples from the wildlife class. By directly examining the AFHQ dataset, we identify six animals: {lion, tiger, wolf, fox, leopard, cheetah}; we do not further distinguish among these animals e.g., leopard versus snow leopard. Within the AFHQ training dataset, these animals are represented in the wildlife class with proportions $\{0.2615, 0.2254, 0.0897, 0.0933, 0.2003, 0.1290\}$, as annotated by GPT o5-mini.

Our motivation for choosing this problem was twofold. First, the quality of images generated by the base TarFlow model is high, such that a pre-trained classifier could attain high accuracy without fine-tuning. Second, we observe that the wildlife images generated by the base TarFlow model contained predominantly lions and leopards (Figure 9), and rarely contained foxes or wolves. From $5 \times 10^3$ samples annotated by GPT o5-mini, we computed animal proportions $\{0.3590, 0.1260, 0.0404, 0.0256, 0.2704, 0.1752\}$.

For image classification, we queried GPT o5-mini to classify each image according to the following prompt:

```
You are labeling animal photos.
Return JSON only: {"label": <one of the options>, "confidence": <0..1>}.
Choose exactly one from: lion, tiger, wolf, fox, leopard, cheetah or none.
```

Although we require the model to state its confidence when labeling the images, we do not use these confidence scores for fine-tuning. The calibration function $h(x) \in \mathbb{R}^5$ is a one-hot encoding of the first five classes. Since nearly all of the samples from the base TarFlow model are labeled as one of the six animals, we observe that choosing a six-dimensional constraint (i.e., adding cheetah) results in a poorly conditioned dual problem (7), since then the components of $h$ are nearly linearly dependent. Our target is the uniform distribution over animals $h^* = [.167, .167, .167, .167, .167]$.

We perform calibration with regularization parameter $\lambda = 10^{-4}$, and sample size $N = 5 \times 10^3$. We assess the success of calibration using two metrics: the total variation (TV) distance of the distribution over animal proportions (using $5 \times 10^3$ annotated images) to the uniform distribution and the FID, computed using only the samples belonging to the wildlife class in the AFHQ training dataset. We use $5 \times 10^4$ samples from the generative model to compute FID. It is important to note that the FID is an imperfect metric for assessing the quality of generated images since it will be lower for models whose animal class makeup is similar to that of the training distribution. To account for this, we evaluate CGM-relax on the maximum entropy reweighting of the training dataset to the uniform distribution over animal classes. In other words, we up or down weighted images belonging to a particular animal class in order to sample the six animals belonging to wildlife class with equal probability.

Our best model, calibrated using CGM-relax with $\lambda = 10^{-4}$, obtains class proportions $\{0.2248, 0.0854, 0.1750, 0.1566, 0.1668, 0.1086\}$, evaluated using $5 \times 10^3$ samples from the model; 0.0828 of the samples were labeled as `None`. CGM-relax reduces the miscalibration error by nearly three times, from a TV distance of .306 to .108 (Figure 10). However, the FID score increases from 15.9 to 21.9. CGM-reward is unsuccessful at calibrating the base model; both miscalibration and FID is roughly the same as the base model. Since CGM-reward remains close to the base model, we evaluate FID on the original AFHQ training dataset.

In Figure 9, we provide random generations from both the pre-trained and the model calibrated with CGM-relax with $\lambda = 10^{-4}$. By examining samples from the calibrated model, we observe two axes along which sample quality worsens after calibration. First, some of the samples (those labeled as `None`) are dogs or cats, which lie outside the AFHQ wildlife class. Second, a greater proportion of samples depict blends of multiple animals.

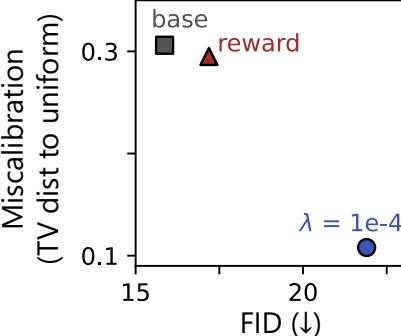

Figure 10: Calibrating TarFlow with CGM-relax reduces the TV distance of animal class labels to the uniform distribution by approximately three times. However, CGM-relax also produces fewer realistic samples, as measured by FID.

### E.4 CALIBRATING TINYSTORIES-33M

#### E.4.1 AUTOREGRESSIVE SAMPLING AND LOG-LIKELIHOODS

Following the setup used by Eldan & Li (2023), we sample in the standard autoregressive fashion with no temperature scaling. To compute sequence log-likelihoods, we consider the prompt as given and ignore tokens generated after the first end-of-sequence (EOS) token. Let $m$ be the length of the prompt and $n$ the index of the first EOS token. Then sequence $\boldsymbol{x}$ has log-probability

$$\log p_\theta(\boldsymbol{x}) = \sum_{i=m+1}^{n} \log p_\theta(\boldsymbol{x}(i) \mid \boldsymbol{x}(<i)).$$

For computational efficiency during training and evaluation, we set the maximum length of each story to be 200 tokens.

#### E.4.2 TINYSTORIES CONSTRAINT DEFINITION

To calibrate TinyStories-33M, we use a simple heuristic procedure to detect the gender of the story's character associated with the profession in the prompt. The procedure returns 1 for female, $-1$ for male, and 0 if the gender cannot be determined. Given a generated story, our procedure is as follows.

**(1) Pronoun at sentence two.** If the second sentence begins with a third-person singular pronoun, we assign gender based on that pronoun. This is common with our prompt templates, e.g., "Once upon a time there was a doctor named Sam. **She** was very kind...".

**(2) First-sentence scan.** If step (1) is inconclusive, we iterate through the words in the story's first sentence. If a title ("Mr.", "Mrs.", "Miss", "Ms.") appears, we assign the corresponding gender. Otherwise, we treat each word as a potential first name and query the `gender-guesser` package (Pérez et al., 2016). If the package classifies the token as "male", "mostly male", "female", or "mostly female", we assign the corresponding gender; otherwise we continue scanning.

**(3) No evidence.** If no gender is detected, we assign 0 (unknown).

We acknowledge the limitations of this simple approach but consider it sufficient for a proof of concept.

**Conditional constraint via sum-of-losses.** We wish to satisfy the conditional calibration constraints

$$\mathbb{E}[\boldsymbol{h}(\boldsymbol{x}) \mid \mathrm{prompt}_i] = 0, \quad \text{for } i = 1, \dots, k$$

which encodes that the male and female labels should be balanced *for each* of the $k$ professions. We implement this as a sum of CGM losses $\sum_{i=1}^{k} \widehat{\mathcal{L}}_i$, where $\widehat{\mathcal{L}}_i$ is the reward or relax loss for the conditional generative model $p_\theta(\boldsymbol{x} \mid \mathrm{prompt}_i)$. During every training batch, we sample 64 stories for each of the eight professions, resulting in a total batch size of 512.

#### E.4.3 TINYSTORIES-33M FIGURE DETAILS

Both panels of Figure 5 were created using 20 replicates per model with different Pytorch seeds, with points indicating the mean metric value across replicates. 2048 stories were sampled per profession for each replicate, resulting in $14 \times 2048 = 28672$ stories per model. Due to the high number of samples and replicates, two-times-standard-error-of-the-mean error bars are smaller than the markers, so are not shown.

**Khalifa et al. (2021) baseline.** Khalifa et al. (2021) use the same method as CGM-reward to define an approximate target distribution $p_{\widehat{\boldsymbol{\alpha}}_N}$. Unlike CGM-reward, they minimize the forward KL

$$
\mathrm{D}_{\mathrm{KL}}\left(p_{\widehat{\boldsymbol{\alpha}}_N} \parallel p_\theta\right) = \mathbb{E}_{p_{\widehat{\boldsymbol{\alpha}}_N}}\left[\log \frac{p_{\widehat{\boldsymbol{\alpha}}_N}(\boldsymbol{x})}{p_\theta(\boldsymbol{x})}\right]
$$

$$
= \mathbb{E}_{p_{\mathtt{stop\text{-}grad}(\theta)}}\left[\frac{p_{\widehat{\boldsymbol{\alpha}}_N}(\boldsymbol{x})}{p_{\mathtt{stop\text{-}grad}(\theta)}(\boldsymbol{x})} \log \frac{p_{\widehat{\boldsymbol{\alpha}}_N}(\boldsymbol{x})}{p_\theta(\boldsymbol{x})}\right] \quad \text{(change of measure)}
$$

$$
= \mathbb{E}_{p_{\mathtt{stop\text{-}grad}(\theta)}}\left[-\frac{p_{\widehat{\boldsymbol{\alpha}}_N}(\boldsymbol{x})}{p_{\mathtt{stop\text{-}grad}(\theta)}(\boldsymbol{x})} \log p_\theta(\boldsymbol{x})\right] + C
$$

$$
= \mathbb{E}_{p_{\mathtt{stop\text{-}grad}(\theta)}}\left[-\frac{p_{\theta_{\mathrm{base}}}(\boldsymbol{x}) \exp\left\{\widehat{\boldsymbol{\alpha}}_N^\top \boldsymbol{x} - A_{p_{\theta_{\mathrm{base}}}}(\widehat{\boldsymbol{\alpha}}_N)\right\}}{p_{\mathtt{stop\text{-}grad}(\theta)}(\boldsymbol{x})} \log p_\theta(\boldsymbol{x})\right] + C \quad \text{(definition of } p_{\widehat{\boldsymbol{\alpha}}_N})
$$

$$
= K\mathbb{E}_{p_{\mathtt{stop\text{-}grad}(\theta)}}\left[-\frac{p_{\theta_{\mathrm{base}}}(\boldsymbol{x}) \exp\left\{\widehat{\boldsymbol{\alpha}}_N^\top \boldsymbol{x}\right\}}{p_{\mathtt{stop\text{-}grad}(\theta)}(\boldsymbol{x})} \log p_\theta(\boldsymbol{x})\right] + C,
$$

where $C$ and $K$ are constants that do not depend on $\theta$. $C$ can be ignored as it has no affect on parameter gradients, and $K$ can be absorbed into the learning rate. Similar to CGM-reward, gradients of this KL-divergence are estimated using Monte Carlo. For a fair comparison, we use the same $\widehat{\boldsymbol{\alpha}}_N$ and batch size to train Khalifa et al. (2021), CGM-reward, and CGM-relax.

**Distance from base-model (symmetrized KL) definition.** For each fine-tuned model, we sample $N = 2048$ stories $\{\boldsymbol{x}_i\}_{i=1}^N$ per profession, and compute log-probabilities $\log p_\theta(\boldsymbol{x}_i)$ and base-model log-probabilities $\log p_{\theta_{\mathrm{base}}}(\boldsymbol{x}_i)$. We estimate the per-profession backward KL as

$$
\mathrm{D}_{\mathrm{KL}}\left(p_\theta \parallel p_{\theta_{\mathrm{base}}}\right) \approx \frac{1}{N} \sum_{i=1}^N \log \frac{p_\theta(\boldsymbol{x}_i)}{p_{\theta_{\mathrm{base}}}(\boldsymbol{x}_i)}.
$$

The forward KL uses importance sampling estimate

$$
\mathrm{D}_{\mathrm{KL}}\left(p_{\theta_{\mathrm{base}}} \parallel p_\theta\right) \approx \frac{1}{N} \sum_{i=1}^N \frac{p_{\theta_{\mathrm{base}}}(\boldsymbol{x}_i)}{p_\theta(\boldsymbol{x}_i)} \log \frac{p_{\theta_{\mathrm{base}}}(\boldsymbol{x}_i)}{p_\theta(\boldsymbol{x}_i)}.
$$

We add our estimates for the forward and backward KL for each profession to get the symmetrized KL, then report the average symmetrized KL across all eight professions.

**Gender imbalance definition.** For each model replicate, we compute the number of male (#male) and number of female (#female) characters in 2048 samples for each profession. The miscalibration for a single profession is defined as

$$
\left|\frac{\#\text{male} - \#\text{female}}{\#\text{male} + \#\text{female}}\right|,
$$

which takes maximum value 1 if all samples used the same gender and minimum value 0 if there are an equal number of each gender. The overall miscalibration values shown on the y-axis of Figure 5A were computed by taking the average miscalibration for the eight professions used for fine-tuning.

**Estimating KL for the max-entropy solution.** Using $N = 2048$ samples from the base-model for each profession, we compute an estimate $\widehat{\boldsymbol{\alpha}}$ of $\boldsymbol{\alpha}^*$ for each profession using the procedure outlined in Appendix C.2. We then compute importance weights for each sample $w_i = \exp(\widehat{\boldsymbol{\alpha}}^\top \boldsymbol{x}_i)$, which we use to compute normalized weights $\widetilde{w}_i = w_i / \sum_{j=1}^N w_j$. To estimate the forward and backward KL divergences, we treat base-model probabilities as $1/N$ and use

$$
\mathrm{D}_{\mathrm{KL}}\left(p_{\theta_{\mathrm{base}}} \parallel p_{\widehat{\boldsymbol{\alpha}}}\right) \approx \frac{1}{N} \sum_{i=1}^N \log \frac{1/N}{\widetilde{w}_i}, \qquad \mathrm{D}_{\mathrm{KL}}\left(p_{\widehat{\boldsymbol{\alpha}}} \parallel p_{\theta_{\mathrm{base}}}\right) \approx \frac{1}{N} \sum_{i=1}^N \frac{\widetilde{w}_i}{1/N} \log \frac{\widetilde{w}_i}{1/N}.
$$

This procedure was repeated 20 times for different sampling seeds to ensure variance from estimating $\boldsymbol{\alpha}$ had little effect on the outcome. The resulting error bars for the symmetrized KL are smaller than the marker size.

### E.4.4 EXAMPLE GENERATIONS

We provide example generations for four prompts before and after fine-tuning with CGM-relax with $\lambda = 0.1$.

### Listing 1: Samples from TinyStories-33M

Once upon a time there was a doctor named Jack. He was fit and strong, loved helping people. One day, Jack was working on an operation. He soon came across a little girl named Mary who was scared. Even though Jack told her not to worry, he said "hello" to her. Mary was still scared, and the doctor could tell her not to worry. He said that everything would be okay for her to do the operation. Mary felt a little bit better. Jack helped Mary and made sure she was fit. He even helped her little brother and made sure that she was always safe. At the end of the operation, Mary was smiling. She thanked Jack for his help and for helping her son. She had learned lessons about not being scared and having someone stronger than her.

Once upon a time there was a lawyer named Matt. He was very new and people stopped to talk to him. He came across a big building which said "Quiet down." Matt thought it was a bad idea but he was very curious. So he did as he was told and soon his house became quiet. All of a sudden he heard a voice and it said, "What are you doing?" It was the jail officer! Matt replied, "I thought it was a good idea to reverse the law in this house." The jail officer said, "Smart Jackie, you know a lot! This house is the law of the world." Matt smiled and said, "But it was fun to reverse the law." The jail officer nodded and said, "Yes it was. But the law hates them." Matt thanked the jail officer and went on his way. The moral of the story is that it pays to be creative, but also to be careful when you

Once upon a time there was a teacher named Miss Jane. She loved to answer any questions and her class was empty. One day, when Miss Jane was busy writing something on the top of her desk, she heard a voice coming from her kitchen floor. She looked down to see a small mouse squeezing a piece of cheese into the board. Miss Jane gasped and said, "mouse, that's naughty! Put that cheese down!" The mouse replied, "No, I won't. I'm making sure no one steals the cheese." Miss Jane was getting frustrated and said, "That's not right. You must ask before you litter into someone else's work. That's not nice." The mouse knew from the right place and quickly put the cheese back. Miss Jane saw what he had done and said, "That's the right thing to do!" Missy stared at him in amazement and said, "I suppose you won't tell everyone what to do with it again!"

Once upon a time there was a pilot named Bobby. He flew a plane on his travels. One day, Bobby was flying in his plane when he saw a tree fall. He crashed the plane right into the tree. Bobby felt so sorry. He called for help, but no one was around. Suddenly, he saw a 3−year−old child standing alone above the tree. Bobby flew to the child and offered to help. He told the little girl not to worry and helped him onto his plane. Together, they flew the plane back to its home in the tree. Bobby and the 3−year−old child became good friends after that until finally Bobby had to fly on time. The end.

### Listing 2: Samples from CGM-relax ($\lambda = 0.1$) model

Once upon a time there was a doctor named Susie. She was a very patient and kind lady who always supported others. Every day she took her to see lots of patients. She always tried to help them feel better. She was always so kind and always willing to lend a hand. One day, a patient called Sally came to see Susie. Sally was very sick and needed help. Susie was always so gentle and caring, not taking extra time to help others. When Sally asked her if she was okay, Susie just smiled and reassured her that she would be alright soon. She was so patient with Sally as she played and chased around the toy store. It was so nice to see how patient and caring Susie is between helping others. She continues to be sure to always lend a hand whenever she can and remember to be there for her when feeling ill.

Once upon a time there was a lawyer named Cass. Cass and Tim went to a big city. They decided to sit on the corner. It was a little dizzy when they sat because they were too dizzy. At the office, Mommy and Daddy talked to the boss. They asked the lawyer for help. They listened to his ideas and his ideas. The lawyer gave them big smiles. He showed them how to mix some ingredients together to make a cookie. The lawyer's ideas and his ideas fit the cookie dough. His ideas were so neat and the cookie dough was extra yummy! The lawyer thanked him and said, "Thanks for your help. You kept your ideas nice and happy." Cass and Tim smiled as they watched the lawyer leave the office. They waved goodbye as they walked away, still feeling happy each

Once upon a time there was a teacher named Mr. Jam and Mrs. Bunny. Mr. Bunny has very cool haircuts that everyone wanted. One day Mrs. Bunny said, "Tomorrow we will have a new surprise." The next morning, Mrs. Bunny came over to Rob's house. When they got to the house Mrs. Bunny said, "I have a new surprise for you!" She pulled out a big bicycle ticket. Mr. Bunny said, "This is for you my grandkids. Now settle down and take it for a ride." The little bunnies were so happy! They sat quietly and quietly, taking turns on the bicycle. Then Mrs. Bunny gave them a surprise, a cool drink. The little bunnies were so excited. They each had a cool drink! All the bunnies were so happy to be outside on such a cool day. And they thanked Mrs. Bunny for the special surprise.

Once upon a time there was a pilot named Judy. She loved to measure everything: trees, houses, farms, and anything else. She was really careful to measure each inch so that every time that she got a bit closer to the number. One day Judy was flying to measure a star in the sky. She drew a line through the line with her finger and measured it for a long time. She was almost done when something terrible happened. One of the letters flew too close and hit Judy in the face. It hurt a lot! Judy was very frightened and ran away from the window to escape from the bad hit. But unfortunately the bad letter kept rolling closer and closer until it was right at the edge of the world. Judy was so scared and upset she couldn't believe it! She had been measured and measured, but still got past the bad letter. The bad letter that Judy had measured was gone forever and Judy was left feeling very sad and lonely.

