# OpenReview forum: "Calibrating Generative Models"
_ICLR.cc/2026/Conference — Submitted to ICLR 2026_

### Official Review · Reviewer_HdMe · 2025-10-27

**Soundness:** 3
**Presentation:** 3
**Contribution:** 3
**Rating:** 6
**Confidence:** 2

**Summary:**

The authors propose two methods for performing generative modeling under moment constraints:

(1) *CGM-relax* encodes the constraint via a penalty on an unbiased estimate of the squared constraint deviation. (2) *CGM-reward* leverages an equivalence to the maximum-entropy principle, for which a solution of the considered problem can be computed in closed form, in the non-parametric setting and under complete knowledge of the underlying distribution. Thus, the idea behing CGM-reward is to minimize the Kullback-Leibler divergence of a generative model to an estimate of said closed-form solution.

For both (1) and (2), the authors propose gradient estimates that they show to be unbiased.

Both methods are evaluated on a range of models with tractable likelihood estimates (Gaussian mixture models, diffusion models, normalizing flows, language models) and tasks (protein design, image generation, 1D toy problem, natural language). Overall, CGM-relax performs favorably to CGM-reward at constraint satisfaction, especially if many constraints are present.

A central limitation of both methods lies in the fact that they rely on tractable estimates of the likelihood.

**Strengths:**

1. The manuscript is well-written (by and large).

2. The proposed methods are practically relevant.

3. The CGM-reward fine-tuning approach and its connection to the principle of maximum entropy is elegant and insightful.

4. Diverse experiments and rigorous evaluation are performed.

5. The authors highlight limitations openly and clearly (I value this highly, because it lies in contrast to most ICLR submissions).

I need to admit that I am not too familiar with the related work. Hence, it is difficult for me to judge novelty.

**Weaknesses:**

1. I am confused about the framing: The manuscript claims to be about calibration, but it really seems to be about generative modeling under moment constraints. Without going into proper mathematical definitions (like, e.g., [1]), calibration (typically defined for classifiers) means: "Model uncertainty is in line with true uncertainty". An extension of this concept to generative modeling would then be that the distribution is modeled well in the sense that there is no mode collapse. It seems the authors define "calibration" quite differently. My criticism is just regarding the wording, but I am afraid that many readers will be confused and that the "actual target audience" (those which are interested in generative modeling under constraints) will not discover this work. I therefore suggest modifying the title and the main body of the text accordingly.

2. The relax loss is somewhat naive and I have some concerns about it: In constrained optimization, such penalty formulations typically lead to either poor conditioning of the loss surface (for large $\lambda$) or the constraints are not satisfied (for small $\lambda$). I believe that this penalty method is a reasonable ablation, but I think it would be more interesting to replace it by the augmented Lagrangian method [2], for instance. One could just plug in the derived estimate for the constraint violation (equation 3) and end up with a more mature method that would very likely work better.

If the authors (i) implement my suggestion in weakness 1 or provide me a convincing argument that the term "calibration" is adequate and (ii) run additional experiments with a mature approach for constrained optimization (ideal) or at least add a critical discussion section about the CGM-relax method, I will raise my score.

[2] Magnus R. Hestenes. Multiplier and Gradient Methods. Journal of Optimization Theory and Applications, 4:303–320, 1969.

**Questions:**

* l. 10: *"wherein class probabilities and other statistics of the sampling distribution deviate from desired values"*. I suggest replacing *"desired"* with *"true"*.

* l.22: *"language models represent gender, race, religion, and age in ways that reinforce societal biases"*. While this is an important point, I believe this is not related to poor calibration, at least not from the definition I am familar with (e.g., [1]). If the generative model reflects societal biases as they are in the data, then it is well-calibrated. Please see my weakness 1 for more information.

* l.120: *"Theorem 2.1. Under assumptions, there exists a unique solution to (4) that has the form"* Would it be possible to re-write this by either (i) listing the assumptions explicitly; or (ii) write something like *"Theorem 2.1. Under the assumptions listed in Appx. ... , there exists a unique solution to (4) that has the form"* Otherwise, this is not a proper theorem.

* l.122: I highly recommend writing out what $\alpha^*$ is. I am afraid that without this information, the theorem statement is incomplete.

* l.136: *"which states that, under conditions,"* Similarly as in my previous comment, it would be helpful for the reader if the authors could write out the conditions or refer to the appendix to read up on them.

* l.213-215: Why is this unintuitive? And what do the authors mean by the score function being *"non-trivial in general"*?

* l.322: *"Consistent with our results in Section 3 we find that optimally-tuned CGM-relax outperforms CGM-reward, which falls short of meeting the calibration constraints."* Should CGM-relax not fall short of meeting the moment constraints, too? (see weakness 2)

[1] Wang, Cheng. "Calibration in deep learning: A survey of the state-of-the-art." arXiv preprint arXiv:2308.01222 (2023).

---

> ### Author Response · Authors · 2025-11-20
> **Response to Reviewer HdMe**
>
> We thank the reviewer for taking the time to provide feedback on our manuscript. We are glad they found the connection between generative model calibration and maximum entropy to be “elegant and insightful” and our experiments to be “diverse” and “rigorous”.
>
> > I need to admit that I am not too familiar with the related work. Hence, it is difficult for me to judge novelty.
>
> We appreciate the reviewer's honesty regarding their unfamiliarity with prior work. The literature on fine-tuning generative models under moment constraints is sparse and scattered across model classes.
>
> The core contribution of our work is to provide, to our knowledge, the first general-purpose framework for fine-tuning generative models with tractable likelihoods to satisfy expectation constraints. Despite applying broadly, our algorithm outperforms previous algorithms for narrower classes (e.g., [1]).
>
> In our response to Reviewer C8DD, we further solidify CGM as a practical algorithm by demonstrating that it effectively reduces constraint violation on the 9B Gemma 2 language model, while remaining within ~1 nat of the base model.
>
> # Weaknesses
> > The manuscript claims to be about calibration, but it really seems to be about generative modeling under moment constraints.
>
> We appreciate the reviewer pointing out that our definition of “calibration” may be misleading to some since the term has an alternate definition in supervised learning  (“on average, the model’s prediction is correct”). We agree with the suggestion and have renamed the manuscript to “Calibrating generative models to distributional constraints”. Additionally, we added a discussion of this alternate definition of calibration to the “Related work” section of the main text (see PDF upload).
>
> > The relax loss is somewhat naïve and I have some concerns about it. [...] I think it would be more interesting to replace it by the augmented Lagrangian method, for instance.
>
> We thank the reviewer for raising the point about the relax loss surface being ill-conditioned and for their suggestion to compare against the augmented Lagrangian (AL) method. We find empirically that choosing $\lambda$ via grid search (as in Section 3) mitigates poor conditioning of the loss landscape when $\lambda$ is too small, and the AL method does not demonstrate improvements over CGM-relax. In the appendix of the camera-ready version, we will provide a discussion of the AL method and experimental comparisons to CGM-relax.
>
> In Propositions C.6 and C.7, we show that for the measure-valued penalty problem,
> $$
> \text{argmin} \ \lambda D_{\text{KL}}(Q || P)+||E_Q[\boldsymbol{h}]-\boldsymbol{h}^\ast||^2,
> $$
> the solution $Q_{\lambda}$ converges to the solution of the calibration problem in the $\lambda \to 0$ limit. However, for small $\lambda$ a perturbation to the solution $Q_{\lambda}$ may not result in a large increase to the objective (i.e., the loss surface is ill-conditioned). This is why we advocate choosing $\lambda$ via grid search, balancing between constraint satisfaction and distance to the base model.
>
> Moreover, we agree with the reviewer that the AL method is a principled alternative to CGM-relax. In its textbook form, the AL method is intractable since the penalized Lagrangian does not admit a closed-form minimum with respect to the model parameters. Consequently, we jointly optimize the model parameters and Lagrange multipliers; this adds an additional hyperparameter: how frequently the Lagrange multipliers are updated.
>
> We compare CGM-relax to the AL method for fine-tuning a diffusion model to reweight the modes of a GMM (Figures 2A, 2B). We choose the penalty parameter according to a 10-point grid search on a log-linear scale and test two values for the Lagrange multiplier update frequency (20, 100 iterations). In both settings, the AL method performs no better than CGM-relax.
>
> # Questions
> > L.10, 22
>
> We hope our response to the reviewer’s previous question will address potential confusion between our definition of generative model calibration and the alternative definition used in supervised learning.
>
> > L.120, 122, 136
>
> We thank the reviewer for their suggestions to improve the mathematical clarity of Theorem 1 and the discussion surrounding the dual problem (eq. 6). We have implemented these changes in the revised manuscript we uploaded.
>
> > L.213-215
>
> We simply intend to convey that when the proposal distribution is equal to the current model, the importance weights are 1 and their gradient is the score function. Our revised manuscript makes this clearer.
>
> > L.322
>
> We show in Appendix C.3 that as $\lambda \to 0$, the constraint violation of the solution to the measure-valued penalty problem $Q_{\lambda}$ approaches zero at rate $\lambda$. Hence, we expect that when $\lambda$ is small and $p_{\theta}$ is expressive, the minimizer of (eq. 2) should approximately meet the calibration constraint.
>
> [1] Khalifa, M., et al. A distributional approach to controlled text generation. ICLR, 2021.

---

> > ### Author Response · Authors · 2025-11-20
> > **Response to Reviewer HdMe (Experimental Results)**
> >
> > ## Comparison of CGM-relax to Augmented Lagrangian (AL)
> >
> > ### Rare event (Figure 2A)
> > **KL to base model**
> > | Base prop.| Max ent | AL | CGM-relax |
> > |----------------------|----------------------|-----------------------|----------------------|
> > |0.8|0.|0.00±0.000|0.00±0.000|
> > |8e-2|1.50|1.89±0.012|1.68±0.017|
> > |9e-3|3.29|4.26±0.028|3.79±0.015|
> > |9e-4|5.07|9.81±1.160|6.10±0.058|
> > |9e-5|6.67|0.00±0.001|5.96±2.592|
> > |1e-5|7.57|0.00±0.000|4.69±3.109|
> >
> > **Constraint violation**
> > |Base prop.| Max ent | AL | CGM-relax |
> > |----------------------|----------------------|-----------------------|----------------------|
> > |0.8|0.|0.01±0.000|0.01±0.000|
> > |8e-2|0.|0.01±0.000|0.04±0.000|
> > |9e-3|0.|0.01±0.000|0.05±0.000|
> > |9e-4|0.|0.14±0.007|0.06±0.001|
> > |9e-5|0.|0.80±0.000|0.44±0.195|
> > |1e-5|0.|0.80±0.000|0.57±0.213|
> >
> > ### Increasing constraint dimension (Figure 2B)
> > **KL to base model**
> > | Constraint dim | Max ent | AL | CGM-relax |
> > |----------------------|----------------------|-----------------------|----------------------|
> > |1|0.2|0.23±0.001|0.17 ±0.001|
> > |200|40|62.14±0.053 |57.14±0.073|
> > |400|80|125.42±0.144|117.93±0.055|
> > |600|120|189.14±0.221|177.66±0.154|
> > |800|160|253.01±0.489 |237.92±0.192|
> > |1000|200|317.14±0.364 |298.98±0.166|
> >
> > **Constraint violation**
> > | Constraint dim | Max ent | AL | CGM-relax |
> > |----------------------|----------------------|-----------------------|----------------------|
> > |1|0.|0.01±0.000|0.03 ±0.000|
> > |200|0.|0.24±0.001 |0.37±0.001|
> > |400|0.|0.40±0.003|0.55±0.003|
> > |600|0.|0.50±0.004|0.68±0.005|
> > |800|0.|0.57±0.009|0.79±0.006|
> > |1000|0.|0.63±0.007|0.89±0.007|

---

> > > ### Comment · Reviewer_HdMe · 2025-11-21
> > >
> > > I thank the authors for addressing my points. Concern (i) is addressed well, but for (ii) I cannot find the results of these experiments in the updated manuscript. Also, after reading the other reviews, I have have some novelty concerns (raised by reviewer VapA).
> > >
> > > I will therefore keep my score and confidence.

---

> > > > ### Author Response · Authors · 2025-11-22
> > > > **Response to Reviewer HdMe**
> > > >
> > > > We appreciate the reviewer’s reply to our rebuttal and would like to further address their concerns regarding (i) the comparisons between the Augmented Lagrangian (AL) method and CGM and (ii) the novelty of CGM.
> > > >
> > > > To supplement the table we shared in our previous reply comparing CGM-relax to the AL algorithm, **we have added Appendix D.4, “Comparison to Augmented Lagrangian Method”**, which includes background on the AL algorithm, plots comparing the AL algorithm to CGM for two sets of synthetic data experiments, and pseudocode (Algorithm 3) for implementing the AL algorithm for the calibration problem. We have also explicitly referenced these comparisons in Section 3 of the main text:
> > > > *“We provide additional discussion of our simulation experiments in Appendix D, including an overview of diffusion models in Appendix D.1 and a comparison between CGM and the Augmented Lagrangian (AL) algorithm in Appendix D.4.”*
> > > >
> > > > As to the reviewer’s concern regarding novelty, we refer the reviewer to our response to reviewer VapA. **We are unaware of any prior method that addresses miscalibration of generative models in the general setting we consider.** While penalty methods and maximum entropy are classical tools in optimization and statistics, it is not clear that the relax or reward objectives admit unbiased gradient estimates with sufficiently low variance to be practical for high-dimensional latent-variable models, such as diffusion models. We hope the reviewer will reconsider the contributions of our manuscript and will not conclude a lack of novelty before VapA replies to the contrary.

---

> > > > > ### Comment · Reviewer_HdMe · 2025-11-23
> > > > >
> > > > > Thanks. I can find the AL experiments now, but I still have concerns regarding novelty from reading a bit into the literature. It seems that the approach that works (CGM-reward) among the two proposed ones is quite ad hoc. In case the paper is rejected, for a future submission, I would recommend investigating why the more interesting CGM-reward approach does not work and how one could find solutions for that.

---

> > > > > > ### Author Response · Authors · 2025-11-23
> > > > > > **Response to Reviewer HdMe**
> > > > > >
> > > > > > We thank the reviewer for their reply and for taking the time to assess the novelty of our work. Could the reviewer share a reference (or references) for the prior work that brings the novelty of CGM into question?
> > > > > >
> > > > > > We want to ensure that we are representing our contributions accurately and that we are not overlooking relevant literature.

---

> > > > > > > ### Comment · Reviewer_HdMe · 2025-11-27
> > > > > > >
> > > > > > > I looked at the work [1] that was brought up by reviewer VapA. While [1] is not about generative models, it still seems that the step of implementing this idea for generative models is not that great. Given that I am slightly leaning toward acceptance already, I will rather keep the score that I have. I wish the authors best of luck with this work.
> > > > > > >
> > > > > > > [1] Bertsekas, Dimitri P. "On penalty and multiplier methods for constrained minimization." SIAM Journal on Control and Optimization 14.2 (1976): 216-235.

---

### Official Review · Reviewer_VapA · 2025-10-27

**Soundness:** 2
**Presentation:** 3
**Contribution:** 2
**Rating:** 2
**Confidence:** 2

**Summary:**

The article introduces a calibration procedure for generative models. Calibration is understood in the article as matching the expected value of a nonlinear function to a desired target value. The article thus views calibration as a constrained optimization problem and introduces two heuristic approaches to solve these, one based on the penalty method, one called 'max entropy approach', which can, however, be interpreted as minimizing a corresponding Lagrange function.

The article presents numerical results spanning both toy examples and more advanced case studies, including protein folding and vision transformers.

**Strengths:**

The method has a low complexity and seems to work in practice. Although the degree to which it works successfully is not direclty visible from the experiments. The presentation of the ideas is adequate and the writing is clear.

**Weaknesses:**

There is very little originality and scientific contribution in the article. Essentially, the article suggests to view calibration through the lens of constrained optimization and applies a penalty method (approach 1) or minimizes a Lagrange dual (approach 2). I found the title and framing misleading, as I would expect statistical guarantees, which, however, cannot be delivered by the ad-hoc nature of the approach (or this would need substantial refinement).

Moreover, the approaches to constrained optimization are largely adhoc (penalty method cannot guarantee constraint satisfaction; approach 2 operates on a fixed multiplier) and cannot guarantee constraint satisfaction. Constraint satisfaction would be important for extracting meaningful statistical guarantees. In addition the problem formulation (minimization subject to expectation constraint) is widely studied in the stochastic optimization community, e.g., under the name of multistage stochastic program.

**Questions:**

Have the authors looked into knowledge distillation? I could imagine that approaches similar to the ones proposed in the article are frequently introduced as baselines.

In many practical situations one would like to use indicator functions for h(x). In these situations h(x) is no longer differentiable. Did the authors look carefully into this situation? The proposed, gradient-based optimization approaches do not seem to work well for this situation (essentially the gradient of the indicator is zero almost everywhere).

---

> ### Author Response · Authors · 2025-11-20
> **Response to Reviewer VapA**
>
> We thank the reviewer for taking the time to provide feedback on our manuscript. We believe we have compellingly addressed their concerns in our response, and we would be happy to answer any additional questions they may have.
>
> # Weaknesses
> > I found the title and framing misleading, as I would expect statistical guarantees, which, however, cannot be delivered by the ad hoc nature of the approach.
>
> We appreciate the reviewer pointing out that our definition of “calibration” may be misleading to some since the term has an alternate definition in supervised learning (informally stated as “on average, the model’s prediction is correct”). We agree with the suggestion and have renamed the manuscript to “Calibrating generative models to distributional constraints”.  Additionally, we added a discussion of this alternate definition of calibration to the “Related work” section of the main text (see PDF upload).
>
> > The degree to which it [CGM] works successfully is not directly visible from the experiments.
>
> We appreciate the reviewer pointing out the simplicity and practicality of CGM. However, we respectfully disagree with their comment that our experiments do not showcase its successes.
>
> Across diffusion models, normalizing flows, and autoregressive language models, CGM consistently reduces the majority of miscalibration error, even under hundreds of simultaneous constraints and in models with up to 1.4B parameters. In addition, we show in our response to Reviewer C8DD that our CGM fine-tuning algorithm continues to be effective on larger language models (9B parameters).
>
> These results represent substantial improvements upon prior distribution-level calibration methods such as [1], which apply only to a single model class and achieve weaker constraint satisfaction.
>
> > There is very little originality and scientific contribution in the article.
>
> We thank the reviewer for their perspective, and we would like to clarify that the primary contribution of our work lies in showing that distribution-level miscalibration in generative models can be cast as a constrained optimization problem that admits practical, unbiased gradient estimates. Moreover, these gradients have sufficiently low variance even for models with high-dimensional latent variables (e.g., diffusion models). To our knowledge, no prior work has demonstrated that such a practical and general approach is achievable.
>
> It is not our intention to present the use of a penalty method or a Lagrange dual for constrained optimization as novel.
>
> > Moreover, the approaches to constrained optimization are largely ad hoc and cannot guarantee constraint satisfaction.
>
> We thank the reviewer for raising this important point about constraint satisfaction. In our manuscript we show that, although the parametric problem we solve is non-convex, the measure-valued problem underlying CGM’s relax objective is not ad hoc and admits a precise theoretical characterization.
>
> In Appendix C.3, we relate the relax problem to a measure-valued penalty problem
> $$
> Q_{\lambda}=\text{argmin} \ \lambda D_{\text{KL}}(Q || P) + ||E_Q[\boldsymbol{h}] - \boldsymbol{h}^\ast ||^2.
> $$
> We explicitly characterize the solution $Q_{\lambda}$ as an exponential tilt of the base distribution $P$. Moreover, we characterize the suboptimality of $Q_{\lambda}$ in terms of its constraint satisfaction, both in the finite $\lambda$ (Proposition C.6) and $\lambda \to 0$ (Proposition C.7) regimes. As $\lambda \to 0$, $||E_{Q_{\lambda}}[\boldsymbol{h}]-\boldsymbol{h}^\ast||=O(\lambda)$.
>
> # Questions
>
> > Have the authors looked into knowledge distillation?
>
> We are not aware of a connection between knowledge distillation and distribution-level calibration, and we have been unable to identify relevant references in a preliminary literature search. If the reviewer has specific references in mind, we would be happy to address them.
>
> > In many practical situations one would like to use indicator functions for $\boldsymbol{h}(\boldsymbol{x})$. Did the authors look carefully into this situation?
>
> We agree with the reviewer that non-differentiable constraints are common in practice and important to consider. In fact, for all four experimental settings we describe in Section 4, the constraint function is an indicator function.
>
> Crucially, while the constraint function $\boldsymbol{h}(\boldsymbol{x})$ may not be differentiable in $\boldsymbol{x}$, the CGM relax and reward losses depend on the expectation of the constraint under the generative model $p_{\theta}$, which is differentiable in the model parameters $\theta$. The non-differentiability of $\boldsymbol{h}$ is not a concern when the gradient is approximated with the score function gradient estimate described in Section 2.3. This is a reason we prefer the score function gradient estimate over the reparameterization gradient, for which nondifferentiability would be an issue.
>
> [1] Khalifa, M., et al. A distributional approach to controlled text generation. ICLR, 2021.

---

> > ### Comment · Reviewer_VapA · 2025-11-27
> > **Rebuttal acknowledgement**
> >
> > Dear authors,
> >
> > I would like to thank you for the time and effort spent in engaging with my feedback and in revising the manuscript. I also appreciate the fact that the authors substantiated the dependence of constraint violation on the parameter \lambda, this clearly strengthens the manuscript.
> >
> > However, I still don't quite understand the depth of the contributions. The authors' theoretical results substantiate the issues of using penalty-based approaches (which are known for a long time). Here is why:
> >
> > 1) From an optimization point of view, some of the results in the appendix, such as Prop. C.7 seem to be well known. A classic paper on the subject is D. Bertsekas, "On Penalty and Multiplier Methods for Constrained Minimization", SIAM J. of Control and Optimization, 1976.  I thus fail to see somewhat the concrete technical contributions of the paper that are non-incremental extensions from prior works.
> >
> > 2) The fact that the constraint violation scales with O(\lambda) or O(\sqrt{lambda}) is actually worrying (and at the same time typical for penalty methods), as it means that the penalty parameter needs to become arbitrarily small to mitigate constraint violations. This affects the conditioning of the optimization, hence convergence may be very slow for \lambda small.
> >
> > I understand that the numerical examples are well executed, well presented in the paper, and highlight that the penalty method works somewhat. However, the level of numerical experiments seems standard for today's top-tier conferences, as it involves small adjustments to existing models/training setups. In summary, I am still missing the novel scientific insight (either theoretical or empirical) that is beyond the status quo. Feel free to correct me if my assessment is not appropriate.

---

> ### Author Response · Authors · 2025-11-27
> **Response to Reviewer VapA**
>
> We appreciate the reviewer taking the time to critically evaluate our rebuttal, and we would like to respond to their concerns regarding (i) constraint satisfaction for penalty methods and (ii) the novelty of our work.
>
> ## Constraint Satisfaction
> We agree with the reviewer that the $\mathcal{O}(\lambda)$ constraint violation is standard for penalty methods with squared Euclidean norm penalty (e.g., [1], [2]). Our intention is not to claim a new rate, but rather to show how this classical behavior manifests in the measure-valued maximum entropy problem (eq. 4) and to identify when a quadratic penalty is expected to perform well in practice.
>
> Namely, the proof of Proposition C.7 demonstrates that **the constraint can be satisfied to high accuracy with finite $0 < \lambda < 1$ whenever the Fisher information matrix of the exponential family defined by the maximum entropy solution is well-conditioned**. See Appendix C.3, (eq. 23). Conversely, when the constraints (i.e., components of $\boldsymbol{h}$) are nearly linearly dependent, then the Fisher information matrix is poorly conditioned and we would expect approximate constraint satisfaction to be attainable only by choosing $\lambda$ to be very small.
>
> We hypothesize that this is why we observe CGM-relax to be successful in our case studies, and largely insensitive to the choice of $\lambda$: so long as there is sufficient curvature in the exponential family about $\boldsymbol{\alpha}^\ast$, approximate constraint satisfaction is attainable with finite $\lambda$.
>
> ## Novelty
> Although penalty and Augmented Lagrangian methods have been well explored theoretically and empirically, their adoption within the generative modeling community is highly nonstandard. For example, in computational protein design, structure diffusion models rely on increased noise during inference time to ensure the structural diversity of sampled proteins matches that of naturally occurring proteins [3].
>
> To the best of our knowledge, **our work is the first general method to go beyond such heuristics** and show that computation of unbiased, low-variance gradient estimates enables stochastic optimization of classical convex-analytic objectives for constraint satisfaction in generative models. This step—from minimizing a convex, measure-valued objective to stochastic optimization of a non-convex objective (in the model parameters)—is nontrivial since modern generative models are high dimensional, and it is not obvious that the gradients of these objectives can be estimated with low variance. The success of CGM across numerous generative model classes with high-dimensional latent variables and parameter spaces provides definitive evidence that such an approach is possible.
>
> [1] Hestenes, Magnus R. Multiplier and gradient methods. Journal of Optimization Theory and Applications, 1969.
>
> [2] Powell, Michael JD. A method for nonlinear constraints in minimization problems. Optimization, 1969.
>
> [3] Yim, J., et al. SE(3) diffusion model with application to protein backbone generation. ICLR, 2023.

---

### Official Review · Reviewer_YGqy · 2025-10-29

**Soundness:** 3
**Presentation:** 3
**Contribution:** 3
**Rating:** 6
**Confidence:** 3

**Summary:**

This paper introduces a framework to correct distribution-level miscalibration in generative models. Authors formalize calibration as a constrained optimization problem. Because the exact constraint is intractable, they propose two surrogate fine-tuning methods, one uses soft quadratic plus KL regularization (CGM-relax) and one approximates the maximum-entropy projection of the base model toward a target exponential-family distribution (CGM-Reward). Then they tested these to proteins, images, and language domains.

**Strengths:**

- The writing is cohesive and clear. Problem definition on calibration is sound and clear, and their methods, especially connection with maximum entropy problem is interesting.
- They also provide a nice practical way to realize theory.

**Weaknesses:**

- I think the main concern of this paper is about end-point only calibration. CGM adjust only the terminal marginal distribution (e.g. final diffusion sample) rather than the entire probability flow. In other words, we lose access on how the probability path changes by this finetuning process. Someone can say this is not a problem if we can sample from the target distribution (with desired constraints) anyways, but in terms of theory I am not sure if this is the direction we really want. For example FK steering tries to control the pathwise dynamics.

**Questions:**

- Regarding weakness 1, what is the benefit of this method compared to recent reward-guided finetuning methods that controls pathwise dynamics? Is there any specific problem setup where CGM is the only remedy? I am happy to discuss this more, and raise score.

---

> ### Author Response · Authors · 2025-11-20
> **Response to Reviewer YGqy**
>
> We thank the reviewer for their thoughtful feedback on our manuscript. We are glad they found the connection between the generative model calibration and the maximum entropy problems “interesting” and that CGM “provides a nice practical way to realize theory.”
>
> # Weaknesses
> > I think the main concern of this paper is about end-point only calibration. CGM adjusts only the terminal marginal distribution (e.g., final diffusion sample) rather than the entire probability flow. In other words, we lose access to how the probability path changes by this fine-tuning process.
>
> We appreciate the reviewer’s insightful question and are eager to share our understanding of how CGM adjusts the distribution of the full path when fine-tuning a diffusion generative model. In this setting, the base model is
> $$p_{\theta_{\text{base}}}: d\boldsymbol{x}(t) = b_{\theta_{\text{base}}}(\boldsymbol{x}(t), t)dt + \sigma(t)d\boldsymbol{w}(t), \quad \boldsymbol{x}(0) = p_{\text{init}}, \quad 0 \leq t \leq 1,$$
> and the constraint is defined only on the terminal distribution $\mathbb{E}[\boldsymbol{h}(\boldsymbol{x}(1))] = \boldsymbol{h}^*$.
>
> As the reviewer pointed out, there are infinitely many diffusion processes that satisfy the constraint on the terminal distribution. However, the calibration problem explicitly seeks the one closest in KL divergence (on the full path measures) to the base model $p_{\theta_{\text{base}}}$.  Moreover, an implication of the equivalence of calibration and reward fine-tuning (our Theorem 1) is that under regularity conditions, the solution to the calibration problem $p^\ast(\boldsymbol{x})$ satisfies a strong structural property
> $$p^\ast(\cdot | \boldsymbol{x}(1)) = p_{\theta_{\text{base}}}(\cdot | \boldsymbol{x}(1)).$$
> Domingo-Enrich et al., 2025 [1] (Theorem 1) provides a complete statement and proof.
>
> Consequently, **although the terminal marginal changes to satisfy the constraint, the conditional path distribution given the endpoint is preserved**. Thus, calibration does not arbitrarily alter the entire probability flow; it changes the endpoint distribution while maintaining the same conditional dynamics “backwards from the endpoint.”
> We appreciate the reviewer highlighting this point. Our initial submission omitted these details because they relate only to the application of CGM to diffusion models; in light of renewed interest we have added a dedicated discussion in Appendix D.1, "Solution to the maximum entropy problem" (see PDF upload).
>
> # Questions
> > What is the benefit of this method compared to recent reward-guided fine-tuning methods that control pathwise dynamics? Is there any specific problem setup where CGM is the only remedy?
>
> CGM addresses settings for which we believe it was not previously clear how to apply reward fine-tuning methods. Consider, for example, the protein application from Section 4.1. Here, we observe that statistics of model samples deviate systematically from those of naturally occurring proteins. With CGM, we express this deviation as a disagreement in expected value of a high-dimensional statistic, and we seek to minimize that disagreement by fine-tuning.
>
> From the perspective of reward fine-tuning, it is not clear how one would define a scalar reward function $r(\boldsymbol{x})$ to accomplish the same goal. One of the key conceptual contributions of our work is the observation (via the maximum entropy formulation) that such calibration problems can be re-expressed as reward fine-tuning problems that control pathwise dynamics [1,2].
>
> However, leveraging this equivalence algorithmically requires first approximately computing the parameter $\boldsymbol{\alpha}^\ast$ defining the reward for which the equivalence holds. Once the form of the reward has been computed, many options for inference may be possible.  Our score-based gradient estimation approach is just one way, which we choose for its simplicity and because it allows for non-differentiable rewards. By contrast, other reward fine-tuning methods (e.g., adjoint matching [1]) require differentiability of the constraint function and thus cannot be applied.
>
> [1] Domingo-Enrich, C., et al. Adjoint matching: fine-tuning flow and diffusion generative models with memoryless stochastic optimal control. ICLR, 2025.
>
> [2] Uehara, M., et al. Fine-tuning of continuous time diffusion models as entropy-regularized control. arXiv preprint, 2024.

---

### Official Review · Reviewer_C8DD · 2025-11-03

**Soundness:** 3
**Presentation:** 3
**Contribution:** 3
**Rating:** 6
**Confidence:** 3

**Summary:**

This work formulates calibration of generative models as finding the closest distribution to a base model that matches specified expectation constraints. This paper introduced two fine-tuning losses, CGM-relax (penalized constraint violation + KL to base) and CGM-reward (match exponential-tilt max-entropy target) with unbiased loss/gradient estimators and leave-one-out baselines. The authors tested with protein/image/language generative models and show large reductions in miscalibration under constraints, with a slight degradation in quality compared with the base model.

**Strengths:**

- The paper includes experiments that showed robust empirical coverage across different modalities (proteins/images/text), with different model architectures, making the approach very usable
- Both algorithms only need sampling, log-density, and score, which most diffusion/flow/LM codebases already support, lowering adoption costs across modalities. The paper also shows how to realize these for continuous-time diffusion and masked LMs.

**Weaknesses:**

- The CGM-relax method relies on the $\lambda$ parameter, which balances the constraint violation and the KL penalty. The paper currently uses grid search to find optimal values, and an analysis of the sensitivity to $\lambda$ and heuristics for setting it would make the work more practical.
- Scale-up to larger LMs is untested. TinyStories-33M and ESM3 1.4B are helpful, but applying CGM to popular 7-30B LMs would stress the need to compute long-sequence log-probs and scores efficiently
- The method requires tractable likelihoods and scores, making it challenging to extend to VAEs/GANs

**Questions:**

- For images, did you compute class-conditional FID/FJD to decouple class-mix shifts from visual quality?
- How does $\lambda$ interact with temperature, classifier-free guidance, or noise schedules?
- How sensitive is CGM-relax to the choice of $\lambda$? For a new problem, do you have any heuristics or intuitions for setting a good initial raneg for $\lambda$ to avoid a costly search?

---

> ### Author Response · Authors · 2025-11-20
> **Response to Reviewer C8DD**
>
> We thank the reviewer for their feedback and are glad they recognize the “robust empirical coverage” of our experiments. We found their suggestions helpful and believe the manuscript has improved by addressing them.
>
> # Weaknesses
> >  The paper currently uses grid search to find optimal values (of $\lambda$), and an analysis of the sensitivity to $\lambda$ and heuristics for setting it would make the work more practical.
>
> We agree with the reviewer that the choice of $\lambda$ is an important consideration and that providing a clear heuristic to choose it along with further sensitivity analyses will strengthen the paper. In our response, we prescribe our heuristic for selecting $\lambda$ that we will include in the camera-ready version, and we describe additional sensitivity experiments we intend to run. We provide additional sensitivity results for the Genie2 and ESM3 case studies.
>
> **Heuristic for $\lambda$.** We observe miscalibration decreases as $\lambda$ decreases (Figure 1B) and $\log(\lambda)$ in $[-3, 0]$ performs well across all case studies. We therefore suggest performing a 10-point grid search on $\log(\lambda)$ in this range and choosing the largest value for which the constraint violation is reduced relative to the base model by 90%.  We adopted this heuristic for our experiments in Section 3. In the camera-ready version, we will recommend this procedure as part of the CGM-relax method in a new paragraph of Section 2.1 titled “Selecting $\lambda$”.
>
> In compute-limited regimes, one could use fewer grid points or perform a logarithmic scale binary search. In practice, fine-tuning is fast and inexpensive (< 2 days on at most 2 H100 GPUs), so this is rarely needed.
>
> **Sensitivity to $\lambda$.** We find that in most settings CGM-relax is not sensitive to $\lambda$.  In our submission, we provided sensitivity analyses in Figures 1 and 5. In the camera-ready version, we will include experimental results for a grid of $\lambda$ in all case studies.
>
> > Applying CGM to popular 7-30B LMs would stress the need to compute long-sequence log-probs and scores efficiently.
>
> We thank the reviewer for this suggestion. We ran additional experiments demonstrating that CGM scales favorably to a large-scale language model; we will include the results in our camera-ready version.
>
> In a scaled-up version of our TinyStories setup, we applied CGM-relax to the 9B-parameter Gemma 2 instruction-tuned model to calibrate the gender of main characters in children’s stories conditioned on profession. We used a TinyStories-style prompt asking for 3-5 paragraph stories in simple language, and we explicitly requested “roughly equal probabilities” of male and female characters.
>
> Despite this prompt, the base Gemma 2 model exhibits strong gender imbalance (doctor: 12% female vs 87% male; nurse: 96% vs 3%). We then fine-tuned Gemma 2 with CGM-relax on eight target professions on two H100 GPUs for ~8 hours. After CGM-relax, all calibrated professions are close to the desired 50/50 marginal. Additionally, the fine-tuned model only has KL divergence 1 nat with respect to the base model, indicating the sample distribution is very similar.
>
> Interestingly, the calibrated Gemma 2 model also generalizes better than our TinyStories-33M models to professions not included in the CGM objective (accountant improves from 88% miscalibration to 8%).
>
> > The method requires tractable likelihoods and scores, making it challenging to extend to VAEs/GANs.
>
> We agree with the reviewer that this requirement is a limitation.  We suspect extensions using reparameterization gradients or alternative regularizers (e.g., MMDs) might apply to VAEs and GANs; these are beyond our scope.
>
> # Questions
> > For images, did you compute class-conditional FID to decouple class-mix shifts from visual quality?
>
> Yes, when evaluating the FID of CGM on the AFHQ wildlife class, we annotated the animal type of each real data sample and reweighted the samples to reflect the target distribution. From this reweighted distribution, we sampled images (with replacement) to form a new balanced dataset.
>
> > How does it interact with temperature, classifier-free guidance, or noise schedules?
>
> Any sampling choices, such as temperature or noise schedule, determine a distribution over samples to which CGM can be applied. For example, in our Genie2 diffusion experiments we chose a noise schedule that yielded high sample diversity before fine-tuning. However, changing sampling parameters after fine-tuning will not, in general, preserve satisfaction of the moment constraints. If the practitioner is interested in calibrating the model’s conditional distributions (which might be the case for a diffusion model trained with classifier-free guidance), we propose minimizing a sum of losses (Appendix E.4.2), each of which targets the miscalibration for a single conditional distribution.
>
> Overall, CGM is not tied to any particular sampling scheme and can be used as a tool for conditional calibration.

---

> > ### Author Response · Authors · 2025-11-20
> > **Response to Reviewer C8DD (Experimental Results)**
> >
> > ## Sensitivity Experiments for Genie2 and ESM3
> >
> > ### Genie2
> > | $\lambda$| Prop. design failure | Sym. KL |
> > |----------------------|----------------------|-----------------------|
> > |base|0.016|3.301|
> > |1e-1|0.127±0.007|0.692±0.052|
> > |1e-3|0.128±0.002|0.845±0.046|
> > |1e-4|0.138±0.013|0.845±0.046|
> >
> > ### ESM3
> > | $\lambda$| Prop. design failure | Sym. KL |
> > |----------------------|----------------------|-----------------------|
> > |base|0.752|0.683|
> > |1e0|0.771±0.006|0.236±0.016|
> > |1e-1|0.778±0.013|0.283±0.051|
> > |1e-2|0.855±0.022|0.281±0.049|
> >
> > ## Results for Gemma 2 9B-parameter Model
> >
> > | Profession | Pre-CGM miscalibration | Post-CGM miscalibration |
> > |  --  |  --  |  --  |
> > | doctor | 75% | 1% |
> > | lawyer | 74% | 9% |
> > | teacher | 69% | 7% |
> > | pilot | 89% | 5% |
> > | chef | 88% | 3% |
> > | scientist | 62% | 12% |
> > | nurse | 93% | 10% |
> > | artist | 49% | 3% |
> > | **Held-out professions** |  |  |
> > | sheriff | 93% | 13% |
> > | judge | 14% | 52% |
> > | accountant | 88% | 8% |
> > | dancer | 18% | 28% |
> > | athlete | 86% | 16% |
> > | baker | 93% | 16% |

---

### Author Response · Authors · 2025-11-26

We again thank the reviewers for the time and thought put into evaluating our manuscript. We have posted our rebuttals and addressed all raised concerns in detail. In particular, we added experiments demonstrating that CGM scales to the Gemma 2 9B model (Reviewer C8DD), provided a comparison to the Augmented Lagrangian algorithm (Reviewer HdMe), and offered additional theoretical insight into how CGM relates to reward fine-tuning algorithms for diffusion models (Reviewer YGqy). We have also sought to elucidate the contributions of our work (Reviewers VapA, HdMe). These additions will be incorporated into the final manuscript.

As reviewer discussions remain open until December 2, we wanted to let the reviewers know that we are happy to provide any additional clarifications, experiments, or references that would help resolve remaining questions.

---

### Meta-Review · Area_Chair_ydW2 · 2026-01-07

**Summary:**

This work calibrates generative models by finding the closest distribution to a base model that matches specified expectation constraints via fine tuning. Reviewers and I all agree this is an interesting idea. Meanwhile, multiple reviewers had concerns that the constrained optimization was not done in a sophsicated enough fashion. Unfortunately, these concerns seem to remain largely unresolved after the rebuttal discussion. In addition, if the assumption is such calibration improves the performance of the generative model, then evaluation could place more emphasis on generation quality instead of constraint satisfaction. Recognizing the potential of the idea, I encourage the authors to take the discussions into consideration and re-submit a revised version.

**Reviewer Concerns:**

I feel the concern about experiments is relatively well addressed, but not those on constrained optimization.

**Reviewer Scores:**

Just a guess: they probably won't increase by too much.

---

### Decision · Program_Chairs · 2026-01-26

Reject